# Adaptive Quasi-Newton and Anderson Acceleration Framework with Explicit Global Convergence Rates

## Abstract

Despite the impressive numerical performance of quasi-Newton and Anderson/nonlinear acceleration methods, their global convergence rates have remained elusive for over 50 years. This paper addresses this long-standing question by introducing a framework that derives novel and adaptive quasi-Newton or nonlinear/Anderson acceleration schemes. Under mild assumptions, the proposed iterative methods exhibit explicit, non-asymptotic convergence rates that blend those of gradient descent and Cubic Regularized Newton's method. Notably, these rates are achieved adaptively, as the method autonomously determines the optimal step size using a simple backtracking strategy. The proposed approach also includes an accelerated version that improves the convergence rate on convex functions. Numerical experiments demonstrate the efficiency of the proposed framework, even compared to a fine-tuned BFGS algorithm with line search.

## 1 Introduction

Consider the problem of finding the minimizer $x^\star$ of the unconstrained minimization problem

$$f(x^\star) = f^\star = \min_{x \in \mathbb{R}^d} f(x),$$

where $d$ is the problem's dimension, and the function $f$ has a Lipschitz continuous Hessian.

**Assumption 1.** *The function $f(x)$ has a Lipschitz continuous Hessian with a constant $L$,*

$$\forall \ y, z \in \mathbb{R}^d, \quad \|\nabla^2 f(z) - \nabla^2 f(y)\| \le L\|z - y\|. \tag{1}$$

In this paper, $\|.\|$ stands for the maximal singular value of a matrix and for the $\ell_2$ norm for a vector. Many twice-differentiable problems like logistic or least-squares regression satisfy Assumption 1.

The Lipschitz continuity of the Hessian is crucial when analyzing second-order algorithms, as it extends the concept of smoothness to the second order. The groundbreaking work by Nesterov et al. [45] has sparked a renewed interest in second-order methods, revealing the remarkable convergence rate improvement of Newton's method on problems satisfying Assumption 1 when augmented with cubic regularization. For instance, if the problem is also convex, accelerated gradient descent typically achieves $O(\frac{1}{t^2})$, while accelerated second-order methods achieve $O(\frac{1}{t^3})$. Recent advancements have further pushed the boundaries, achieving even faster convergence rates of up to $\mathcal{O}(\frac{1}{t^{7/2}})$ through the utilization of hybrid methods [42, 14] or direct acceleration of second-order methods [43, 27, 39].

Unfortunately, second-order methods may not always be feasible, particularly in high-dimensional problems common in machine learning. The limitation is that exact second-order methods require solving a linear system that involves the Hessian of the function $f$. This main limitation motivated alternative approaches that balance the efficiency of second-order methods and the scalability of first-order methods, such as *inexact/subspace/stochastic techniques*, *nonlinear/Anderson acceleration*, and *quasi-Newton* methods.

Submitted to 37th Conference on Neural Information Processing Systems (NeurIPS 2023). Do not distribute.

## 1.1 Contributions

Despite the impressive numerical performance of quasi-Newton methods and nonlinear acceleration schemes, there is currently no knowledge about their global explicit convergence rates. In fact, global convergence cannot be guaranteed without using either exact or Wolfe-line search techniques. This raises the following long-standing question **that has remained unanswered for over 50 years**:

*What are the non-asymptotic global convergence rates of quasi-Newton*
*and Anderson/nonlinear acceleration methods?*

This paper provides a partial answer by introducing generic updates (see algorithms 1 to 3) that can be viewed as cubic-regularized quasi-Newton methods or regularized nonlinear acceleration schemes.

Under mild assumptions, the iterative methods constructed within the proposed framework (see algorithms 3 and 6) exhibit *explicit, global and non-asymptotic* convergence rates that interpolate the one of first order and second order methods (more details in appendix A):

- Convergence rate on non-convex problems (Theorem 4): $\min_i \|\nabla f(x_i)\| \leq O(t^{-\frac{2}{3}} + t^{-\frac{1}{3}})$,

- Convergence rate on (star-)convex problems (Theorems 5 and 6): $f(x_t) - f^\star \leq O(t^{-2} + t^{-1})$,

- Accelerated rate on convex problems (Theorem 8): $f(x_t) - f^\star \leq O(t^{-3} + t^{-2})$.

## 1.2 Related work

**Inexact, subspace, and stochastic methods.** Instead of explicitly computing the Hessian matrix and Newton's step, these methods compute an approximation using sampling [2], inexact Hessian computation [29, 19], or random subspaces [20, 31, 34]. By adopting a low-rank approximation for the Hessian, these approaches substantially reduce per-iteration costs without significantly compromising the convergence rate. The convergence speed in such cases often represents an interpolation between the rates observed in gradient descent methods and (cubic) Newton's method.

**Nonlinear/Anderson acceleration.** Nonlinear acceleration techniques, including Anderson acceleration [1], have a long standing history [3, 4, 28]. Driven by their promising empirical performance, they recently gained interest in their convergence analysis [61, 26, 60, 37, 66, 64, 69, 68, 53, 62, 63, 6, 57, 8, 54]. In essence, Anderson acceleration is an optimization technique that enhances convergence by extrapolating a sequence of iterates using a combination of previous gradients and corresponding iterates. Comprehensive reviews and analyses of these techniques can be found in notable sources such as [37, 7, 36, 35, 5, 17]. However, these methods do not generalize well outside quadratic minimization and their convergence rate can only be guaranteed asymptotically when using a line-search or regularization techniques [59, 65, 53].

**Quasi-Newton methods.** Quasi-Newton schemes are renowned for their exceptional efficiency in continuous optimization. These methods replace the exact Hessian matrix (or its inverse) in Newton's step with an approximation that is updated iteratively during the method's execution. The most widely used algorithms in this category include DFP [18, 25] and BFGS [58, 30, 24, 10, 9]. Most of the existing convergence results predominantly focus on the asymptotic super-linear rate of convergence [67, 32, 12, 11, 15, 22, 72, 70, 71]. However, recent research on quasi-Newton updates has unveiled explicit and non-asymptotic rates of convergence [49, 51, 50, 40, 41]. Nonetheless, these analyses suffer from several significant drawbacks, such as assuming an infinite memory size and/or requiring access to the Hessian matrix. These limitations fundamentally undermine the essence of quasi-Newton methods, which are typically designed to be Hessian-free and maintain low per-iteration cost through their low-memory requirement and low-rank structure.

Recently, Kamzolov et al. [38] introduced an adaptive regularization technique combined with cubic regularization, with global, explicit (accelerated) convergence rates for any quasi-Newton method. The method incorporates a backtracking line search on the secant inexactness inequality that introduces a quadratic regularization. However, this algorithm relies on prior knowledge of the Lipschitz constant specified in Assumption 1. Unfortunately, the paper does not provide an adaptive method to find jointly the Lipschitz constant as well, as it is *a priory* too costly to know which parameter to update. This aspect makes the method impractical in real-world scenarios.

**Paper Organization** **Section 2** introduces the proposed novel generic updates and some essential theoretical results. **Section 3** presents the convergence analysis of the iterative algorithm, which uses one of the proposed updates. **Section 4** is dedicated to the accelerated version of the proposed framework. **Section 5** presents examples of methods generated by the proposed framework.

## 2 Type-I and Type-II Step

This section first examines a remarkable property shared by quasi-Newton and Anderson acceleration: the sequence of iterates of these methods can be expressed as a combination of *directions* formed by previous iterates and the current gradient. Building upon this observation, section 2.1 investigates how to obtain second-order information without directly computing the Hessian of the function $f$ by *approximating* the Hessian within the subspace formed by these directions. Subsequently, section 2.2 demonstrates how to utilize this approximation to establish an *upper bound* for the function $f$ and its gradient norm $\|\nabla f(x)\|$. Minimizing these upper bounds, respectively, leads to a type-I and type-II method.

**Motivation: what quasi-Newton and nonlinear acceleration schemes actually do?** The BFGS update is a widely used quasi-Newton method for unconstrained optimization. It approximates the inverse Hessian matrix using updates based on previous gradients and iterates. The update reads

$$x_{t+1} = x_t - h_t H_t \nabla f(x_t), \quad H_t = H_{t-1} \left( I - \frac{g_t d_t^T}{g_t^T d_t} \right) + d_t \left( d_t^T \frac{d_t^T g_t + g_t^T H_{t-1} d_t}{(g_t^T d_t)^2} - \frac{g_t^T H_{t-1}}{g_t^T d_t} \right)$$

where $H_t$ is the approximation of the inverse Hessian at iteration $t$, $h_t$ is the step size, $d_t = x_t - x_{t-1}$ is the step direction, $g_t = \nabla f(x_t) - \nabla f(x_{t-1})$ is the gradient difference. After unfolding the equation, the BFGS update can be seen as a combination of the $d_i$'s and $\nabla f(x_t)$,

$$x_{t+1} - x_t = H_0 P_0 \dots P_t \nabla f(x_t) + \sum_{i=1}^{t} \alpha_i d_i, \tag{2}$$

where $P_i$ are projection matrices in $\mathbb{R}^{d \times d}$ and $\alpha_i$ are coefficients. Similar reasoning can be applied to other quasi-Newton formulas (see appendix B for more details).

This observation aligns with the principles of Anderson acceleration methods. Considering the same vectors $d_t$ and $g_t$, Anderson acceleration updates $x_{t+1}$ as:

$$\alpha^\star = \min_\alpha \left\| \nabla f(x_t) + \sum_{i=0}^{t-1} \alpha_i r_i \right\|, \quad x_{t+1} - x_t = \sum_{i=0}^{t} \alpha_i^\star (d_i - h_t g_i),$$

where $h_t$ is the relaxation parameter, which can be seen as the step size of the method. As all $x_i$'s belong to the span of previous gradients, the update is similar to (2), see appendix B for more details. This is not surprising, as it has been shown that Anderson acceleration can be viewed as a quasi-Newton method [23]. Some studies have explored the relationship between these two classes of optimization techniques and established strong connections in terms of their algorithmic behavior [23, 73, 56, 13].

Hence, quasi-Newton algorithms and nonlinear/Anderson acceleration methods utilize previous directions $d_i$ and the current gradient $\nabla f(x_t)$ in subsequent iterations. However, their convergence is guaranteed only if a line search is used, and their convergence speed is heavily dependent on $H_0$ (quasi-Newton) or $h_t$ (Anderson acceleration) [48].

### 2.1 Error Bounds on the Hessian-Vector Product Approximation by a Difference of Gradients

Consider the following $d \times N$ matrices that represent the *algorithm's memory*,

$$Y = [y_1, \dots, y_N], \quad Z = [z_1, \dots, z_N], \quad D = Y - Z, \quad G = [\dots, \nabla f(y_i) - \nabla f(z_i), \dots]. \tag{3}$$

For example, to mimic quasi-Newton techniques, the matrices $Y$ and $Z$ can be defined such that,

$$D = [\dots, x_{t-i+1} - x_{t-i}, \dots], \quad G = [\dots, \nabla f(x_{t-i+1}) - \nabla f(x_{t-i}), \dots], \quad i = 1 \dots N.$$

Motivated by (2), this paper studies the following update, defined as a linear combination of the previous directions $d_i$,

$$x_+ - x = D\alpha \quad \text{where} \quad \alpha \in \mathbb{R}^N. \tag{4}$$

The objective is to determine the optimal coefficients $\alpha$ based on the information contained in the matrices defined in (3). Notably, the absence of the gradient in the update (4) distinguishes this

approach from (2), allowing for the development of an adaptive method that eliminates the need for an initial matrix $H_0$ (quasi-Newton methods) or a mixing parameter $h_t$ (Anderson acceleration).

Under assumption (1), the following bounds hold for all $x, y, z, x_+ \in \mathbb{R}^d$ [45],

$$\|\nabla f(y) - \nabla f(z) - \nabla^2 f(z)(y - z)\| \leq \tfrac{L}{2}\|y - z\|^2, \tag{5}$$

$$\left| f(x_+) - f(x) - \nabla f(x)(x_+ - x) - \tfrac{1}{2}(x_+ - x)^T \nabla^2 f(x)(x_+ - x) \right| \leq \tfrac{L}{6}\|x_+ - x\|^3. \tag{6}$$

The accuracy of the estimation of the matrix $\nabla^2 f(x)$, depends on the *error vector* $\varepsilon$,

$$\varepsilon \overset{\text{def}}{=} [\varepsilon_1, \ldots, \varepsilon_N], \quad \text{and} \quad \varepsilon_i \overset{\text{def}}{=} \|d_i\| \left( \|d_i\| + 2\|z_i - x\| \right). \tag{7}$$

The following Theorem 1 explicitly bounds the error of approximating $\nabla^2 f(x)D$ by $G$.

**Theorem 1.** *Let the function $f$ satisfy Assumption 1. Let $x_+$ be defined as in (4) and the matrices $D$, $G$ be defined as in (3) and vector $\varepsilon$ as in (7). Then, for all $w \in \mathbb{R}^d$ and $\alpha \in \mathbb{R}^N$,*

$$-\tfrac{L\|w\|}{2} \sum_{i=1}^{N} |\alpha_i|\varepsilon_i \leq w^T(\nabla^2 f(x)D - G)\alpha \leq \tfrac{L\|w\|}{2} \sum_{i=1}^{N} |\alpha_i|\varepsilon_i, \tag{8}$$

$$\|w^T(\nabla^2 f(x)D - G)\| \leq \tfrac{L\|w\|}{2}\|\varepsilon\|. \tag{9}$$

**Proof sketch and interpretation.**   The theorem states that the Hessian-vector product $\nabla^2 f(x)(y-z)$ can be approximated by the difference of gradients $\nabla f(y) - \nabla f(z)$, providing a cost-effective approach to estimate $\nabla^2 f$ without computing it. This property is the basis of quasi-Newton methods. The detailed proof can be found in appendix F. The main idea of the proof is as follows. From (5) with $y = y_i$ and $z = z_i$, writing $d_i = y_i - z_i$, and Assumption 1,

$$\|\nabla f(y_i) - \nabla f(z_i) - \nabla^2 f(x)(y_i - z_i)\| \leq \frac{L}{2}\|d_i\|^2 + \|\nabla^2 f(x) - \nabla^2 f(z)\|\|d_i\| \leq \frac{L}{2}\varepsilon_i.$$

The *first* term in $\varepsilon_i$ bounds the error of (5), while the *second* comes from the distance between (5) and the current point $x$ where the Hessian is estimated. Then, it suffices to combine the inequalities with coefficients $\alpha$ to obtain Theorem 1.

## 2.2   Type I and Type II Inequalities and Methods

In the literature, Type-I methods often refer to algorithms that aim to minimize the function value $f(x)$, while type-II methods minimize the gradient norm $\|\nabla f(x)\|$ [23, 73, 13]. Applying the bounds (6) and (5) to the update in (4) yields the following Type-I and Type-II upper bounds, respectively.

**Theorem 2.** *Let the function $f$ satisfy Assumption 1. Let $x_+$ be defined as in (4), the matrices $D$, $G$ be defined as in (3) and $\varepsilon$ be defined as in (7). Then, for all $\alpha \in \mathbb{R}^N$,*

$$f(x_+) \leq f(x) + \nabla f(x)^T D\alpha + \tfrac{\alpha^T H \alpha}{2} + \tfrac{L\|D\alpha\|^3}{6}, \quad H \overset{\text{def}}{=} \tfrac{G^T D + D^T G + IL\|D\|\|\varepsilon\|}{2} \tag{10}$$

$$\|\nabla f(x_+)\| \leq \|\nabla f(x) + G\alpha\| + \tfrac{L}{2}\left( \sum_{i=1}^{N} |\alpha_i|\varepsilon_i + \|D\alpha\|^2 \right), \tag{11}$$

The proof can be found in appendix F. Minimizing eqs. (10) and (11) leads to algorithms 1 and 2, respectively, whose constant $L$ is replaced by a parameter $M$, found by backtracking line-search. A study of the (strong) link between these proposed algorithms and nonlinear/Anderson acceleration and quasi-Newton methods can be found in appendix B.

**Solving the sub-problems**   In algorithms 1 and 2, the coefficients $\alpha$ are computed by solving a minimization sub-problem in $O(N^3 + Nd)$ (see appendix C for more details). Usually, $N$ is rather small (e.g. between 5 and 100); hence solving the subproblem is negligible compared to computing a new gradient $\nabla f(x)$. Here is the summary:

- **In algorithm 1**, the subproblem can be solved easily by a convex problem in two variables, which involves an eigenvalue decomposition of the matrix $H \in \mathbb{R}^{N \times N}$ [45].

- **In algorithm 2**, the subproblem can be cast into a linear-quadratic problem of $O(N)$ variables and constraints that can be solved efficiently with SDP solvers (e.g., SDPT3).

---

**Algorithm 1** Type-I Subroutine with Backtracking Line-search

---
**Require:** First-order oracle for $f$, matrices $G$, $D$, vector $\varepsilon$, iterate $x$, initial smoothness $M_0$.
1: Initialize $M \leftarrow \frac{M_0}{2}$
2: **do**
3:     $M \leftarrow 2M$  **and**  $H \leftarrow \frac{G^T D + D^T G}{2} + \mathrm{I}_N \frac{M\|D\|\|\varepsilon\|}{2}$
4:     $\alpha^\star \leftarrow \arg\min_\alpha f(x) + \nabla f(x)^T D\alpha + \frac{1}{2}\alpha^T H\alpha + \frac{M\|D\alpha\|^3}{6}$
5:     $x_+ \leftarrow x + D\alpha$
6: **while**  $f(x_+) \geq f(x) + \nabla f(x)^T D\alpha^\star + \frac{1}{2}[\alpha^\star]^T H\alpha^\star + \frac{M\|D\alpha^\star\|^3}{6}$
7: **return** $x_+$, $M$

---

---

**Algorithm 2** Type-II Subroutine with Backtracking Line-search

---
    Same as algorithm 1, but minimize and check the upper bound (11) instead of (10) on lines 4 and 6.

---

## 3 Iterative Type-I Method: Framework and Rates of Convergences

The rest of the paper analyzes the convergence rate of methods that use algorithm 1 as a subroutine; see algorithm 3. The analysis of methods that uses algorithm 2 is left for future work.

### 3.1 Main Assumptions and Design Requirements

This section lists the important assumptions on the function $f$. Some subsequent results require an upper bound on the radius of the sub-level set of $f$ at $f(x_0)$.

**Assumption 2.** *The radius of the sub-level set $\{x : f(x) \leq f(x_0)\}$ is bounded by $\mathrm{R} < \infty$.*

To ensure the convergence toward $f(x^\star)$, some results require $f$ to be star-convex or convex.

**Assumption 3.** *The function $f$ is star convex if, for all $x \in \mathbb{R}^d$ and $\forall \tau \in [0, 1]$,*
$$f((1 - \tau)x + \tau x^\star) \leq (1 - \tau)f(x) + \tau f(x^\star).$$

**Assumption 4.** *The function $f$ is convex if, for all $y$, $z \in \mathbb{R}^d$, $f(y) \geq f(z) + \nabla f(z)(y - z)$.*

The matrices $Y$, $Z$, $D$ must meet some conditions listed below as "requirements" (see section 5 for details). All convergence results rely on *one* of these conditions on the projector onto **span**$(D)$,
$$P_t \overset{\text{def}}{=} D_t(D_t^T D_t)^{-1}D_t^T. \tag{12}$$

**Requirement 1a.** *For all $t$, the projector $P_t$ of the stochastic matrix $D_t$ satisfies $\mathbb{E}[P_t] = \frac{N}{d}\boldsymbol{I}$.*

**Requirement 1b.** *For all $t$, the projector $P_t$ satisfies $P_t\nabla f(x_t) = \nabla f(x_t)$.*

The first condition guarantees that, in expectation, the matrix $D_t$ spans partially the gradient $\nabla f(x_t)$, since $\mathbb{E}[P_t\nabla f(x_t)] = \frac{N}{d}\nabla f(x_t)$. The second condition simply requires the possibility to move towards the current gradient when taking the step $x + D\alpha$. This condition resonates with the idea presented in (2), where the step $x_+ - x$ combines previous directions and the current gradient $\nabla f(x_t)$.

In addition, it is required that the norm of $\|\varepsilon\|$ does not grow too quickly, hence the next assumption.

**Requirement 2.** *For all $t$, the relative error $\frac{\|\varepsilon_t\|}{\|D_t\|}$ is bounded by $\delta$.*

The Requirement 2 is also non-restrictive, as it simply prevents taking secant equations at $y_i - z_i$ and $z_i - x_i$ too far apart. Most of the time, $\delta$ satisfies $\delta \leq O(R)$.

Finally, the condition number of the matrix $D$ also has to be bounded.

**Requirement 3.** *For all $t$, the matrix $D_t$ is full-column rank, which implies that $D_t^T D_t$ is invertible. In addition, its condition number $\kappa_{D_t} \overset{\text{def}}{=} \sqrt{\|D_t^T D_t\|\|(D_t^T D_t)^{-1}\|}$ is bounded by $\kappa$.*

The condition on the rank of $D$ is not overly restrictive. In most practical scenarios, this condition is typically satisfied without issue. However, the second condition might be hard to meet, but section 5 studies strategies that prevent $\kappa_D$ from exploding by taking orthogonal directions or pruning $D$.

---

**Algorithm 3** Generic Iterative Type-I Methods

---

**Require:** First-order oracle $f$, initial iterate and smoothness $x_0$, $M_0$, number of iterations $T$.
  **for** $t = 0, \ldots, T-1$ **do**
    Update $G_t$, $D_t$, $\varepsilon_t$ (see section 5).
    $x_{t+1}, M_{t+1} \leftarrow$ [algorithm 1]$(f, G_t, D_t, \varepsilon_t, x_t, (M_t/2))$
  **end for**
  **return** $x_T$

---

## 3.2 Rates of Convergence

When $f$ satisfies Assumption 1, algorithm 3 ensures a minimal function decrease at each step.

**Theorem 3.** *Let $f$ satisfy Assumption 1. Then, at each iteration $t \geq 0$, algorithm 3 achieves*

$$f(x_{t+1}) \leq f(x_t) - \tfrac{M_{t+1}}{12}\|x_{t+1} - x_t\|^3, \quad M_{t+1} < \max\left\{2L \; ; \; \tfrac{M_0}{2^t}\right\}. \tag{13}$$

Under some mild assumptions, algorithm 3 converges to a critical point for non-convex functions.

**Theorem 4.** *Let $f$ satisfy Assumption 1, and assume that $f$ is bounded below by $f^*$. Let Requirements 1b to 3 hold, and $M_t \geq M_{\min}$. Then, algorithm 3 starting at $x_0$ with $M_0$ achieves*

$$\min_{i=1,\ldots,t}\|\nabla f(x_i)\| \leq \max\left\{ \frac{3L}{t^{2/3}}\left(12\frac{f(x_0)-f^\star}{M_{\min}}\right)^{2/3} \; ; \; \left(\frac{C_1}{t^{1/3}}\right)\left(12\frac{f(x_0)-f^\star}{M_{\min}}\right)^{1/3} \right\},$$

*where* $C_1 = \delta L\left(\frac{\kappa + 2\kappa^2}{2}\right) + \max_{i \in [0,t]}\|(I - P_i)\nabla^2 f(x_i)P_i\|$.

Going further, algorithm 3 converges to an optimum when the function is star-convex.

**Theorem 5.** *Assume $f$ satisfy Assumptions 1 to 3. Let Requirements 1b to 3 hold. Then, algorithm 3 starting at $x_0$ with $M_0$ achieves, for $t \geq 1$,*

$$(f(x_t) - f^\star) \leq 6\frac{f(x_t)-f^\star}{t(t+1)(t+2)} + \frac{1}{(t+1)(t+2)}\frac{L(3R)^3}{2} + \frac{1}{t+2}\frac{C_2(3R)^2}{4},$$

*where* $C_2 \overset{def}{=} \delta L\frac{\kappa+2\kappa^2}{2} + \max_{i\in[0,t]}\|\nabla^2 f(x_i) - P_i\nabla^2 f(x_i)P_i\|$.

Finally, the next theorem shows that when algorithm 3 uses a stochastic $D$ that satisfies Requirement 1a, then $f(x_t)$ also converges in expectation to $f(x^\star)$ when $f$ is convex.

**Theorem 6.** *Assume $f$ satisfy Assumptions 1, 2 and 4. Let Requirements 1a, 2 and 3 hold. Then, in expectation over the matrices $D_i$, algorithm 3 starting at $x_0$ with $M_0$ achieves, for $t \geq 1$,*

$$\mathbb{E}_{D_t}[f(x_t) - f^\star] \leq \frac{1}{1 + \frac{1}{4}\left[\frac{N}{d}t\right]^3}(f(x_0)-f^\star) + \frac{1}{\left[\frac{N}{d}t\right]^2}\frac{L(3R)^3}{2} + \frac{1}{\left[\frac{N}{d}t\right]}\frac{C_3(3R)^2}{2},$$

*where* $C_3 \overset{def}{=} \delta L\frac{\kappa+2\kappa^2}{2} + \frac{(d-N)}{d}\max_{i\in[0,t]}\|\nabla^2 f(x_i)\|$.

**Interpretation** The rates presented in Theorems 4 to 6 combine the ones of cubic regularized Newton's method and gradient descent (or coordinate descent, as in Theorem 6) for functions with Lipschitz-continuous Hessian. As $C_1$, $C_2$, and $C_3$ decrease, the rates approach those of cubic Newton.

The constants $C_1$, $C_2$, and $C_3$ quantify the error of approximating $D\nabla^2 f(x)D$ by $H$ in (10) into two terms. The first represents the error made by approximating $\nabla^2 f(x)D$ by $G$, while the second describes the low-rank approximation of $\nabla^2 f(x)$ in the subspace spanned by the columns of $D$. The approximation is more explicit in $C_3$, where increasing $N$ reduces the constant up to $N = d$.

To retrieve the convergence rate of Newton's method with cubic regularization, the approximation needs to satisfy three properties: **1)** the points contained in $Y_t$ and $Z_t$ must be close to each other, and to $x_t$ to reduce $\delta$ and $\|\varepsilon\|$; **2)** the condition number of $D$ should be close to 1 to reduce $\kappa$; **3)** $D$ should span a maximum dimension in $\mathbb{R}^d$ to improve the approximation of $\nabla^2 f(x)$ by $P\nabla^2 f(x)P$.

For example, $Z_t = x_t\mathbf{1}_N^T$, $D_t = hI_N$ with $h$ small, and $Y_t = Z_t + D_t$ achieve these conditions. This (naive) strategy estimates all directional second derivatives with a finite difference for all coordinates and is equivalent to performing a Newton's step in terms of complexity.

---

**Algorithm 4** Type-I subroutine with backtracking for the accelerated method

---

**Require:** First-order oracle $f$, matrices $G$, $D$, vector $\varepsilon$, iterate $x$, smoothness $M_0$, minimal norm $\Delta$

Initialize $M \leftarrow \frac{M_0}{2}$, $\gamma \leftarrow \frac{1}{4}\frac{\|\varepsilon\|}{\|D\|}\left(1+\kappa_D^2\right)$, $\texttt{ExitFlag} \leftarrow \texttt{False}$

**while** $\texttt{ExitFlag}$ is False **do**

   Update $M$   and   $H \leftarrow \frac{G^T D + D^T G}{2} + \mathrm{I}_N \frac{M\|D\|\|\varepsilon\|}{2}$

   $\alpha^* \leftarrow \arg\min_\alpha f(x) + \nabla f(x)^T D\alpha + \frac{1}{2}\alpha^T H\alpha + \frac{M\|D\alpha\|^3}{6}$

   $x_+ \leftarrow x + D\alpha$

   **If** $-\nabla f(x_+)^T D\alpha \geq \frac{\|\nabla f(x_+)\|^{3/2}}{\sqrt{\frac{3M}{4}}}$   **and**   $\|D\alpha\| \geq \Delta$ **then** $\texttt{ExitFlag} \leftarrow \texttt{LargeStep}$

   **If** $-f(x_+)^T D\alpha \geq \frac{\|\nabla f(x_+)\|^2}{M\left(\gamma+\frac{\|D\alpha\|}{2}\right)}$ **then** $\texttt{ExitFlag} \leftarrow \texttt{SmallStep}$

**end while**

**return** $x_+, \alpha, M, \gamma, \texttt{ExitFlag}$

---

---

**Algorithm 5** Adaptive Accelerated Type-I Algorithm (Sketch, see appendix D for the full version)

---

**Require:** First-order oracle $f$, initial iterate and smoothness $x_0$, $M_0$, number of iterations $T$.

Initialize $G_0, D_0, \varepsilon_0, \lambda_0^{(1)}, \lambda_0^{(2)}, \Delta, x_1, M_1, (M_0)_1$.

**for** $t = 1, \ldots, T-1$ **do**

   Update $G_t, D_t, \varepsilon_t$.

   **do**

      Compute $v_t \leftarrow \arg\min \Phi_t$, set $y_t = \frac{t}{t+3}x_t + \frac{3}{t+3}v_t$, and update $(M_0)_t$

      $\{x_{t+1}, \texttt{ExitFlag}\} \leftarrow [\texttt{algorithm 4}](f, G_t, D_t, \varepsilon_t, y_t, (M_0)_t, \Delta)$

      **if** $\Phi_{t+1}(v_{t+1}) \leq f(x_{t+1})$ **then** %% Parameters adjustment if needed

         $\texttt{ValidBound} \leftarrow \texttt{False}$

         **if** $\texttt{ExitFlag}$ is $\texttt{SmallStep}$ **then** $\lambda_t^{(1)} \leftarrow 2\lambda_t^{(1)}$, **otherwise** $\lambda_t^{(2)} \leftarrow 2\lambda_t^{(2)}$

      **else**

         $\texttt{ValidBound} \leftarrow \texttt{True}$ %% Successful iteration

      **end if**

   **while** $\texttt{ValidBound}$ is $\texttt{False}$

**end for**

**return** $x_T$

---

## 4 Accelerated Algorithm for Convex Functions

This section introduces algorithm 5, an accelerated variant of algorithm 3 for convex functions, designed using the estimate sequence technique from [43]. It consists in iteratively building a function $\Phi_t(x)$, a regularized lower bound on $f$, that reads

$$\Phi_t(x) = \frac{1}{\sum_{i=0}^t b_i}\left(\sum_{i=0}^t b_i\left(f(x_i) + \nabla f(x_i)(x-x_i)\right) + \lambda_t^{(1)}\frac{\|x-x_0\|^2}{2} + \lambda_t^{(2)}\frac{\|x-x_0\|^3}{6}\right),$$

where $\lambda_t^{(1,2)}$ are non-decreasing. The key aspects of acceleration are as follows (see section 4 for more details): **1)** The accelerated algorithm makes a step at a linear combination between $v_t$, the optimum of $\Phi_t$, and the previous iterate $x_t$. **2)** It uses a modified version of algorithm 1, see algorithm 4. **3)** Under some conditions, the step size can be considered as "large", i.e., similar to a cubic-Newton step. The $\Delta > 0$ ensures the step is sufficiently large to ensure theoretical convergence - but setting $\Delta = 0$ does not seem to impact the numerical convergence. The presence of both small and large steps is crucial to obtain the theoretical rate of convergence.

**Theorem 7.** *Assume $f$ satisfy Assumptions 1, 2 and 4. Let Requirements 1b to 3 hold. Then, algorithm 5 starting at $x_0$ with $M_0$ achieves, for all $\Delta > 0$ and for $t \geq 1$,*

$$f(x_t) - f^\star \leq \frac{(M_0)_{\max}^2}{L}\left(\frac{3R}{t+3}\right)^2 + \frac{4(M_0)_{\max}}{3\sqrt{3}}\max\left\{1\ ;\ \frac{2}{\Delta}\right\}\left(\frac{3R}{t+3}\right)^3 + \frac{\frac{\tilde\lambda^{(1)}R^2}{2} + \frac{\tilde\lambda^{(2)}R^3}{6}}{(t+1)^3}.$$

*where* $\tilde\lambda^{(1)} = 0.5 \cdot \delta\left(L\kappa + M_1\kappa^2\right) + \|\nabla f(x_0) - P_0\nabla f(x_0)P_0\|,$   $\tilde\lambda^{(2)} = M_1 + L,$

$(M_0)_{\max} = \frac{L}{2}(2\Delta + (2\kappa^2+\kappa)\delta) + (2\sqrt{3}-1)\max_{0\leq i\leq t}\|(I-P_i)\nabla^2 f(x_i)P_i\|.$

**Interpretation**  The interpretation is similar to the one from Section 3. Ignoring $\tilde{\lambda}^{(1,2)}$, the rate of Theorem 7 combines the one of accelerated gradient and accelerated cubic Newton [44, 43]. The constant $M_0$ blends the Lipschitz constant of the Hessian $L$ with its approximation errors $(2\kappa^2 + \kappa)\delta$ and $\|(I - P)\nabla^2 f(x)\|$. The better the Hessian is approximated, the smaller the constant.

## 5  Some update strategies for matrices $Y$, $Z$, $D$, $G$

The framework presented in this paper is characterized by its generality, requiring only minimal assumptions on the matrix $D$ and vector $\varepsilon$. This section explores different strategies for updating the matrices from (3), which can be classified into two categories: *online* and *batch techniques*.

**Recommended method.**  Among all the methods presented in this section, the most promising technique seems to be the *Orthogonal Forward Estimates Only*, as it ensures that the condition number $\kappa_D = 1$ and the norm of the error vector $\|\varepsilon\|$ is small.

### 5.1  Online Techniques

The online technique updates the matrix $D$ while algorithms 3 and 5 are running. To achieve Requirement 1b, the method employs either a steepest or orthogonal forward estimate, defined as

$$x_{t+\frac{1}{2}} = x_t - h\nabla f(x_t) \quad \text{(steepest)} \quad \text{or} \quad x_{t+\frac{1}{2}} = x_t - h(I - P_{t-1})\frac{\nabla f(x_t)}{\|\nabla f(x_t)\|} \quad \text{(orthogonal)}.$$

Then, it include $x_{t+\frac{1}{2}} - x_t$ in the matrix $D_t$. The projector $P_{t-1}$ is defined in (12), and parameter $h$ can be a fixed small value (e.g., $h = 10^{-9}$). This section investigates three different strategies for storing past information: *Iterates only*, *Forward Estimates Only*, and *Greedy*, listed below.

$$Y_t = [x_{t+\frac{1}{2}}, x_t, x_{t-1}, \ldots, x_{t-N+1}], \quad Z_t = [x_t, x_{t-1}, \ldots, x_{t-N}] \qquad \text{(Iterates only)}$$

$$Y_t = [x_{t+\frac{1}{2}}, x_{t-\frac{1}{2}}, \ldots, x_{t-N+\frac{1}{2}}], \quad Z_t = [x_t, x_{t-1}, \ldots, x_{t-N}] \qquad \text{(Forward Estimates Only)}$$

$$Y_t = [x_{t+\frac{1}{2}}, x_t, x_{t-\frac{1}{2}}, \ldots, x_{t-\frac{N+1}{2}}], \quad Z_t = [x_t, x_{t-\frac{1}{2}}, \ldots, x_{t-\frac{N}{2}}] \qquad \text{(Greedy)}$$

**Iterates only:**  In the case of quasi-Newton updates and Nonlinear/Anderson acceleration, the iterates are constructed using the equation $x_{t+1} - x_t \in \nabla f(x_t) + \mathbf{span}\{x_{t-i+1} - x_{t-i}\}_{i=1\ldots N}$. The update draws inspiration from this observation. However, it does not provide control over the condition number of $D_t$ or the norm $\|\varepsilon\|$. To address this, one can either accept a potentially high condition number or remove the oldest points in $D$ and $G$ until the condition number is bounded (e.g., $\kappa = 10^9$).

**Forward Estimates Only:**  This method provides more control over the iterates added to $Y$ and $Z$. When using the *orthogonal* technique to compute $x_{i+\frac{1}{2}}$ reduces the constants in Theorems 4, 5 and 7: the condition number of $D$ is equal to 1 as $D^T D = h^2 I$, and the norm of $\varepsilon$ is small ($\|\varepsilon\| \leq O(h)$).

**Greedy:**  The greedy approach involves storing both the iterates and the forward approximations. It shares the same drawback as the *Iterates only* strategy but retains at least the most recent information about the Hessian-vector product approximation, thereby reducing the $\|z_i - x_i\|$ term in $\varepsilon$ (7).

### 5.2  Batch Techniques

Instead of making individual updates, an alternative approach is to compute them collectively, centered on $x_t$. This technique generates a matrix $D_t$ consisting of $N$ orthogonal directions $d_1, \cdots, d_N$ of norm $h$. The corresponding $Y_t, Z_t, G_t$ matrices are then defined as follows:

$$Y_t = [x_t + d_1, \ldots, x_t + d_n], \quad Z_t = [x_t, \ldots, x_t], \quad G_t = [\ldots, \nabla f(x_t + d_i) - \nabla f(x_t), \ldots].$$

This section explores two batch techniques that generate orthogonal directions: *Orthogonalization* and *Random Subspace*. Both lead to $\delta = 3h$ and $\kappa = 1$ in Requirements 2 and 3. However, they require $N$ additional gradient computations at each iteration (instead of one for the online techniques). For clarity, in the experiments, only the Greedy version is considered.

**Orthogonalization:**  This technique involves using any online technique discussed in the previous section and storing the directions in a matrix $\tilde{D}_t$. Then, it constructs the matrices $D_t$ by performing an orthogonalization procedure on $\tilde{D}_t$, such as the QR algorithm. This approach provides Hessian estimates in relevant directions, which can be more beneficial than random ones.

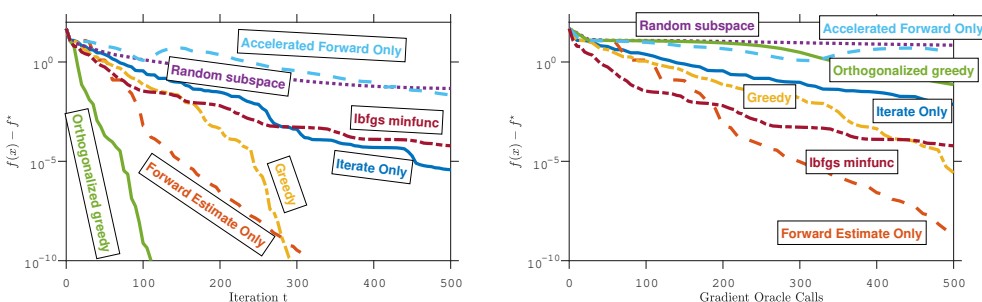

Figure 1: Comparison between the type-1 methods proposed in this paper and the optimized implementation of $\ell$-BFGS from `minFunc` [52] with default parameters, except for the memory size. All methods use a memory size of $N = 25$.

**Random Subspace:** Inspired by [34], this technique randomly generates $D_t$ at each iteration by either taking $D_t$ to be $N$ random (rescaled) canonical vectors or by using the $Q$ matrix from the QR decomposition of a random $N \times D$ matrix. This ensures that $D_t$ satisfies Requirement 1a. For clarity, in the experiments, only the QR version is considered.

## 6 Numerical Experiments

This section compares the methods generated by this paper's framework to the fine-tuned $\ell$-BFGS algorithm from `minFunc` [52]. More experiments are conducted in appendix E. The tested methods are the Type-I iterative algorithms (algorithm 3 with the techniques from section 5). The step size of the forward estimation was set to $h = 10^{-9}$, and the condition number $\kappa_{D_t}$ is maintained below $\kappa = 10^9$ with the iterates only and Greedy techniques. The accelerated algorithm 6 is used only with the *Forward Estimates Only* technique. The compared methods are evaluated on a logistic regression problem with no regularization on the Madelon UCI dataset [33]. The results are shown in fig. 1.

Regarding the number of iterations, the greedy orthogonalized version outperforms the others due to the orthogonality of directions (resulting in a condition number of one) and the meaningfulness of previous gradients/iterates. However, in terms of gradient oracle calls, the recommended method, *orthogonal forward iterates only*, achieves the best performance by striking a balance between the cost per iteration (only two gradients per iteration) and efficiency (small and orthogonal directions, reducing theoretical constants). Surprisingly, the accelerated method's performance is suboptimal, possibly because it tightens the theoretical analysis, diminishing its inherent adaptivity.

## 7 Conclusion, Limitation, and Future work

This paper introduces a generic framework for developing novel quasi-Newton and Anderson/Nonlinear acceleration schemes, offering a global convergence rate in various scenarios, including accelerated convergence on convex functions, with minimal assumptions and design requirements.

One limitation of the current approach is requiring an additional gradient step for the *forward estimate*, as discussed in Section 5. However, this forward estimate is crucial in enabling the algorithm's adaptivity, eliminating the need to initialize a matrix $H_0$ (quasi-Newton) or employ a mixing parameter $h_0$ (Anderson acceleration).

In future research, although unsuitable for large-scale problems, the method presented in this paper can achieve super-linear convergence rates, as with infinite memory, they would be as fast as cubic Newton methods. Utilizing the average-case analysis framework from existing literature, such as [47, 55, 21, 16, 46], could also improve the constants in Theorems 4 and 5 to match those in Theorem 6. Furthermore, exploring convergence rates for type-2 methods, which are believed to be effective for variational inequalities, is a worthwhile direction.

Ultimately, the results presented in this paper open new avenues for researchs. It may also provide a potential foundation for investigating additional properties of existing quasi-Newton methods and may even lead to the discovery of convergence rates for an adaptive, cubic-regularized BFGS variant.

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
