**Supplementary Materials**

 **Preconditioner for the Type 1 Step**

470 This section presents a simple diagonal preconditionner that helps in reducing the theoretical constants
471 that involves the error vector $\varepsilon$. This simple preconditionner impacts the efficiency of the methods
472 presented in this paper, in particular, the accelerated Type-1 step.

473 The type-1 step (10) in Theorem 2 from section 2 is actually a simplified, looser upper bound.
474 Looking at the last steps of proof of Theorem 2, the upper bound on $f$ actually reads

$$f(x_+) \leq f(x) + \nabla f(x)D\alpha + \frac{1}{2}\left((D\alpha)^T G\alpha + \frac{L\|D\alpha\|}{2}\sum_{i=1}^{N}|\alpha_i|\varepsilon_i\right) + \frac{L}{6}\|D\alpha\|^3.$$

475 However, the minimization of the upper may be intractable as it is a non smooth, potentially non
476 convex problem. Therefore, it uses the bounds

$$\sum_{i=1}^{N}|\alpha_i|\varepsilon_i = \alpha^T(\text{sign}(\alpha) \odot \varepsilon) \leq \|\alpha\|\|\varepsilon\|,$$

$$\|D\alpha\| \leq \|D\|\|\alpha\|.$$

477 **Diagonal preconditioner** Introducing a diagonal preconditioner $\mathcal{D}$ leads to those alternatives
478 bounds,

$$\sum_{i=1}^{N}|\alpha_i|\varepsilon_i \leq \|\mathcal{D}\alpha\|\|\mathcal{D}^{-1}\varepsilon\|,$$

$$\|D\alpha\| \leq \|D\mathcal{D}^{-1}\|\|\mathcal{D}\alpha\|.$$

479 which gives the following type-1 upper bound on the function values,

$$f(x_+) \leq f(x) + \nabla f(x)D\alpha + \frac{\alpha^T \tilde{H}\alpha}{2} + \frac{L}{6}\|D\alpha\|^3,$$

480 where

$$\tilde{H} = \frac{R^T D + D^T R + L\|D\mathcal{D}^{-1}\|\|\mathcal{D}^{-1}\varepsilon\|\mathcal{D}^2}{2}.$$

481 The diagonal preconditioner can be set, for instance, to $\texttt{ddiag}(D^T D)$, where $\texttt{ddiag}$ is the operator
482 that extract the diagonal of a matrix. There are two important benefits to use the diagonal precondi-
483 tioner, as it **1)** diminishes the condition number of the matrix $D$, **2)** diminishes the constant $\delta$. The
484 effect of this preconditioner is more important when there is a big difference between the norm of
485 the direction $d_i$, in particular for the *Greedy* strategies and memorize the difference between iterates
486 $x_i - x_{i-1}$ (that can be large) and the forward estimates $x_{i+\frac{1}{2}} - x_i$ (that can be small).

## A  Known rates of convergence

This section explores the known rates of convergence for different optimization methods. Specifically, it focuses on two scenarios: functions with Lipschitz continuous gradient and functions with Lipschitz-continuous Hessian. For smooth functions, the rates of plain gradient descent and its accelerated version are examined. On the other hand, for functions with a Lipschitz-continuous Hessian, the rates of the cubic regularized Newton method and its accelerated variant are investigated.

When the function is smooth, i.e., has Lipschitz continuous gradients,

$$f(y) \leq f(x) + \nabla f(x)(y - x) + \frac{\mathcal{L}}{2}\|y - x\|^2,$$

the rates of plain gradient descent and its accelerated version read [45]

$$\min_{0 \leq i \leq t} \|\nabla f(x_i)\| \leq \sqrt{\frac{\mathcal{L}f(x_0) - f^\star}{t + 1}}, \qquad \text{(plain, non-convex)} \tag{14}$$

$$f(x_t) - f(x^\star) \leq \mathcal{L}\frac{2}{t + 4}\|x_0 - x^\star\|^2, \qquad \text{(plain, convex)} \tag{15}$$

$$f(x_t) - f(x^\star) \leq \mathcal{L}\frac{4}{(t + 2)^2}\|x_0 - x^\star\|^2. \qquad \text{(accelerated)} \tag{16}$$

However, the class of functions considered in this paper is *not* the class of smooth functions. However, if the sequence $\{x_t\}$ is monotone, the constant $\mathcal{L}$ can be estimated as

$$\mathcal{L} \leq LR.$$

On the other hand, when the function has a Lipschitz-continuous Hessian, the cubic regularized Newton method and its accelerated version converge with the following rates [46, 44, 35]:

$$\min_{0 \leq i \leq t} \|\nabla f(x_i)\| \leq \frac{16L}{9}\left(\frac{3(f(x_0) - f^\star)}{2tM_{\min}}\right)^{2/3}, \qquad \text{(plain, non-convex)} \tag{17}$$

$$f(x_t) - f(x^\star) \leq \quad 9L\frac{R^3}{(t + 4)^2}, \qquad \text{(plain, convex)} \tag{18}$$

$$\mathbb{E}[f(x_t)] - f(x^\star) \leq \left(\frac{d - N}{N}\right)\frac{\mathcal{L}(3R)^2}{2t} + \left(\frac{d}{N}\right)^2\frac{L(3R)^3}{3t^2} + O\left(\frac{1}{t^3}\right), \quad \text{(stochastic, convex)} \tag{19}$$

$$f(x_t) - f(x^\star) \leq L\frac{14\|x_0 - x^\star\|}{t(t + 1)(t + 2)}. \qquad \text{(accelerated)} \tag{20}$$

Overall, the rates are faster than first order methods.

## B  Linking with Existing Methods

This section presents the fundamentals of Anderson/nonlinear acceleration (appendix B.1), quasi-Newton schemes (appendix B.2), and their relationship with the proposed method in this paper (appendix B.3).

### B.1  Anderson Acceleration and Nonlinear Acceleration

Anderson acceleration, also known as nonlinear acceleration, is a powerful technique that enhances the convergence speed of fixed point iterations and optimization algorithms. Initially developed for solving linear systems, Anderson acceleration has gained popularity due to its effectiveness in accelerating iterative methods. The method leverages previous iterations to construct an improved estimate of the objective function's minimizer.

The Anderson acceleration algorithm employs the following approximation to compute weights:

$$\nabla f \left( \sum_{i=0}^{N} \beta_i x_i \right) \approx \sum_{i=0}^{N} \beta_i \nabla f(x_i), \ \ \sum_{i=0}^{N} \beta_i = 1.$$

When the function $f$ is quadratic, this approximation becomes an equality. The underlying idea is as follows: since the optimum satisfies $\nabla f(x^\star) = 0$,

$$\sum_{i=0}^{N} \beta_i \nabla f(x_i) \approx 0 \ \ \Rightarrow \nabla f \left( \sum_{i=0}^{N} \beta_i x_i \right) \approx 0 \ \ \Rightarrow \sum_{i=0}^{N} \beta_i x_i \approx x^\star.$$

The Anderson acceleration steps is thus given by

$$x_{t+1} = \sum_{i=0}^{N} \beta_i^\star x_{t-i+1}, \quad \beta^\star = \arg\min_\beta \| \sum_{i=0}^{N} \beta_i \nabla f(x_{t-i+1}) \|^2$$

Over the past decades, the ideas behind Anderson acceleration have been refined. For example, the constraint can be eliminated by considering the step $x_{t+1} - x_t$ instead:

$$x_{t+1} - x_t = \sum_{i=0}^{N} \beta_i x_{t-i+1} - x_t$$

$$= \sum_{i=0}^{N} \tilde{\beta}_i x_{t-i+1}.$$

The vector $\tilde{\beta}_i$ has the property that its sum equals zero. Hence, it can be rewritten as

$$x_{t+1} - x_t = \sum_{i=1}^{N} \alpha_i (x_{t-i+1} - x_{t-i})$$

$$\alpha = \arg\min_\alpha \left\| \nabla f(x_t) + \sum_{i=1}^{N} \alpha_i (\nabla f(x_{t-i+1}) - \nabla f(x_{t-i})) \right\|$$

where $\alpha \in \mathbb{R}^N$ has no constraint. By writing $d_i = x_{t-i+1} - x_{t-i}$, $g_i = \nabla f(x_{t-i+1}) - \nabla f(x_{t-i})$, and $D = [d_t, \ldots, d_{t-N+1}]$, $G = [g_t, \ldots, g_{t-N+1}]$, the step becomes

$$x_{t+1} - x_t = D_t \alpha, \quad \alpha = \arg\min_\alpha \| \nabla f(x_t) + G_t \alpha \|.$$

However, this version of Anderson acceleration is non-convergent because there is no contribution from $\nabla f(x_t)$ in the step $x_{t+1} - x_t$. The most popular solution to this problem is introducing a *mixing parameter* that combines gradient steps, resulting in the following expression:

$$x_{t+1} = x_t - h\nabla f(x_t) + (D - hG)\alpha, \quad \alpha = \arg\min_\alpha \| \nabla f(x_t) + G\alpha \|. \qquad \text{(AA Type II)}$$

Following a similar idea, recent works have introduced a type I variant of the algorithm [23, 72, 76, 13] that minimizes the function value instead of the gradient norm:

$$x_{t+1} = x_t - h\nabla f(x_t) + (D - hG)\alpha, \quad \alpha = \arg\min f(x_t) + \nabla f(x_t)D_t\alpha + \frac{1}{2}\alpha^T D_t^T G_t\alpha,$$
(AA Type I)

By incorporating regularization [56, 13], globalization techniques [76], or performing a line search on the parameter $h$, the algorithm converges towards $x^\star$.

## B.2 Single-secant and Multisecant Quasi-Newton Methods

Quasi-Newton methods, such as the Broyden-Fletcher-Goldfarb-Shanno (BFGS) method, approximate the Hessian matrix in order to efficiently solve unconstrained optimization problems. These methods avoid the expensive computation of the exact Hessian by using iterative updates based on previous iterates and gradients of the objective function.

While the BFGS method has been discussed previously (see section 2), this section focuses on other updates commonly used in quasi-Newton methods: the Davidon-Fletcher-Powell (DFP) formula, the Symmetric Rank-One (SR1) formula, and the Broyden type-1 and type-2 updates.

### B.2.1 The Ideas Behind Single-Secant and Multisecant Hessian Approximation

In quasi-Newton methods, the Hessian approximation is updated using the *secant equation*, which relates the gradients and Hessian at two different points. For a twice continuously differentiable function, the secant equation is given by:

$$\nabla f(y) - \nabla f(x) = \nabla^2 f(\xi)(y - x),$$

where $\xi$ is a point on the line segment connecting $x$ and $y$. This equation serves as the basis for updating the Hessian approximation.

Based on this remarkable identity, quasi-Newton methods update an approximation of the Hessian $B_t$ or its inverse $H_t$ such that the approximation satisfies

$$\nabla f(x_t) - \nabla f(x_{t-1}) = B_t(x_t - x_{t-1}), \quad H_t\left(\nabla f(x_t) - \nabla f(x_{t-1})\right) = x_t - x_{t-1}.$$

What distinguishes the different updates is how to fix the remaining degrees of freedom. For instance, the simple SR-1 method updates $H_t$ such that

$$\min_H \|H - H_{t-1}\|_F \quad : H = H^T, \quad H\left(\nabla f(x_t) - \nabla f(x_{t-1})\right) = x_t - x_{t-1}. \tag{21}$$

Those methods are called *single-secant* as they update $H_t$ only one secant equation at a time. Hence, in general, $H_t$ only satisfies the latest secant equation.

Multisecant updates, on the other hand, approximate the Hessian using a batch of secant equations. By introducing matrices $D = [d_{t-N+1}, \ldots, d_t]$ and $G_t = [g_{t-N+1}, \ldots, g_t]$, the multisecant updates satisfy

$$G_t = B_t D_t, \quad H_t G_t = D_t.$$

Unfortunately, when imposing symmetry, it is impossible satisfy multiple secants at a time [54], although there are some works trying to enforce symmetry while approximating the secant equation in a least square sense [55, 59].

When symmetry is not imposed, the solution for $B_t$ and $H_t$ can be obtained as:

$$B_t = G_t[D_t]^\dagger + B_0(I - D_t D_t^\dagger), \quad H_t = D_t[G_t]^\dagger + H_0(I - G_t G_t^\dagger), \tag{22}$$

where $B_0$ and $H_0$ are the initial approximations, and $[A]^\dagger$ denotes the pseudo-inverse of matrix $A$. Different choices of pseudo-inverse lead to different methods.

The inversion of $B_t$ can be computed using the Woodbury matrix identity, which provides an efficient way to compute the inverse. The update for $B_t^{-1}$ is given by:

$$B_t^{-1} = B_0^{-1}\left(I - G_t\left(D_t^\dagger B_0^{-1} G_t\right)^{-1} D_t^\dagger B_0^{-1}\right) + D_t\left(D_t^\dagger B_0^{-1} G_t\right)^{-1} D_t^\dagger B_0^{-1}.$$

557 This update is equivalent to the update for $H_t$, given that

$$B_0^{-1} = H_0, \quad \text{and} \quad G_t^\dagger = \left(D_t^\dagger B_0^{-1} G_t\right)^{-1} D_t^\dagger B_0^{-1}. \tag{23}$$

558 In summary, quasi-Newton methods use the secant equation to update the Hessian approximation.
559 Single-secant methods update the approximation one secant equation at a time, while multisecant
560 methods use a batch of secant equations. The choice of updating strategy and pseudo-inverse affects
561 the behavior of the method.

### B.2.2 Davidon-Fletcher-Powell (DFP) Formula

563 The DFP formula is a Quasi-Newton update rule used to iteratively refine an approximation of the
564 inverse Hessian matrix. It is defined as follows:

$$H_t = H_{t-1} + \frac{d_t d_t^T}{d_t^T g_t} - \frac{H_{t-1} g_t g_t^T H_{t-1}}{g_t^T H_{t-1} g_t}, \tag{24}$$

565 In the above equation, $g_t = \nabla f(x_t) - \nabla f(x_{t-1})$ represents the difference in gradients, and $d_t = $
566 $x_t - x_{t-1}$ denotes the difference in parameter values. The DFP formula updates the matrix $H_t$ using
567 a rank-two matrix such that it remains symmetric and positive definite.

### B.2.3 Symmetric Rank-One (SR1) Formula

569 The Symmetric Rank-One (SR1) formula is another Quasi-Newton update rule used to estimate the
570 inverse Hessian matrix. It is defined as:

$$H_t = H_{t-1} + \frac{(d_t - H_{t-1} g_t)(d_t - H_{t-1} g_t)^T}{(d_t - H_{t-1} g_t)^T g_t}, \tag{25}$$

571 Here, $g_t = \nabla f(x_t) - \nabla f(x_{t-1})$ and $d_t = x_t - x_{t-1}$. The SR1 formula updates $H_t$ at each iteration
572 to approximate the inverse Hessian matrix, ensuring that the resulting matrix $H_t$ remains symmetric.

### B.2.4 Multisecant Broyden Methods

574 The multisecant Broyden methods utilize the update equation from (22), where $A^\dagger$ is chosen as the
575 Moore-Penrose pseudo-inverse of $A$, given by $A^\dagger = (A^T A)^{-1} A$. In this equation, $B_0$ and $H_0$ are
576 scaled identity matrices. After simplification, the two types of updates can be expressed as follows:

$$B_t^{-1} = D_t \left(D_t^\dagger G_t\right)^{-1} D_t^\dagger + B_0^{-1} \left(I - G_t \left(D_t^\dagger G_t\right)^{-1} D_t^\dagger\right), \tag{26}$$

$$H_t = D_t (G_t^T G_t)^{-1} G_t^T + H_0 \left(I - G_t \left(G_t^T G_t\right)^{-1} G_t^T\right). \tag{27}$$

577 Both updates are quite similar, differing mainly in the choice of the pseudo-inverse of the matrix $G$.

### B.2.5 Link with Anderson Acceleration

579 The connection between quasi-Newton methods and Anderson Acceleration is strong, as for instance,
580 there exists an equivalence between Broyden methods and Anderson acceleration. To illustrate this,
581 let's closely examine the update of $\alpha$ in (AA Type I):

$$x_{t+1} = x_t - h\nabla f(x_t) + (D_t - hG_t)\alpha, \quad \alpha = \arg\min f(x_t) + \nabla f(x_t) D_t \alpha + \frac{1}{2}\alpha^T D_t^T G_t \alpha$$

$$\Leftrightarrow x_{t+1} = x_t - h\nabla f(x_t) + (D_t - hG_t)\alpha, \quad \alpha : D_t^T \nabla f(x_t) + D_t^T G_t \alpha = 0$$

$$\Leftrightarrow x_{t+1} = x_t - h\nabla f(x_t) + (D_t - hG_t)\alpha, \quad \alpha : \alpha = -(D_t^T G_t)^{-1} D_t^T \nabla f(x_t)$$

$$\Leftrightarrow x_{t+1} = x_t - h\nabla f(x_t) - (D_t - hG_t)(D_t^T G_t)^{-1} D_t^T \nabla f(x_t).$$

$$\Leftrightarrow x_{t+1} = x_t - \left(D_t (D_t^T G_t)^{-1} D_t^T + h\left(I - G_t (D_t^T G_t)^{-1} D_t^T\right)\right) \nabla f(x_t)$$

The above step is precisely the quasi-Newton step $x_{t+1} = x_t - B_t^{-1}\nabla f(x_t)$, where $B_t^{-1}$ corresponds to the Broyden update given by Equation 26, with $B_0^{-1} = hI$. A similar reasoning can be applied to Equation 27.

When considering the single secant updates, following the same reasoning as in Section 3 leads to the same conclusion for the SR-1 and DFP updates.

This result is expected since the approximations $H_t$ or $B_t^{-1}$ satisfy the single or multisecant equation:
$$H_t G_t = D_t,$$
indicating that the matrix $H_t$ maps vectors from the span of previous gradients to the span of previous directions. This observation justifies the construction in (4).

## B.3   Links with Algorithms 1 and 2

Both Algorithms 1 and 2 can be viewed as quasi-Newton and Anderson/nonlinear acceleration schemes. The update formulas are

$$\min_\alpha f(x_t) + \nabla f(x_t)^T D_t\alpha + \frac{\alpha^T H_t\alpha}{2} + \frac{M\|D_t\alpha\|^3}{6}, \quad H_t \overset{\text{def}}{=} \frac{G_t^T D_t + D_t^T G_t + IM\|D_t\|\|\varepsilon_t\|}{2}.$$

(Type I)

$$\min_\alpha \|\nabla f(x_t) + G_t\alpha\| + \frac{M}{2}\Big(\sum_{i=1}^N |\alpha_i|[\varepsilon_t]_i + \|D_t\alpha\|^2\Big),$$

(Type II)

The resemblance with Anderson/nonlinear acceleration is strong, as the objective function are similar. In fact, if the function is quadratic, $L = 0$ and therefore $M$ can be set to $0$ as well. In this case, the coefficients $\alpha$ are *exactly* the type I and type II Anderson steps eqs. (AA Type I) and (AA Type II).

The same idea holds when comparing to quasi-Newton methods. In both cases, the optimal solution $\alpha^\star$ can be written implicitly:

$$\alpha^\star = -\Big(H_t + \frac{MD_t^T D_t\|D_t\alpha^\star\|}{6}\Big)^{-1} D_t^T\nabla f(x_t),$$

(Type I - solution)

$$\alpha^\star = -\Big(G_t^T G_t + \tilde{M}D_t^T D_t\Big)^{-1}\Big(G_t^T\nabla f(x) + \frac{\tilde{M}\|\varepsilon_t\|}{2}\partial(|\alpha^\star|)\Big),$$

(Type II - solution)

where $\tilde{M} \overset{\text{def}}{=} \|\nabla f(x_t) + G_t\alpha\|M$ and $\partial(|\alpha^\star|)$ is a subgradient of $|\alpha^*|$. The step then reads
$$x_{t+1} = x_t + D\alpha^\star$$

(Generic step)

$$x_{t+1} = x_t - D_t\Big(H_t + \frac{MD_t^T D_t\|D_t\alpha^\star\|}{6}\Big)^{-1} D_t^T\nabla f(x_t),$$

(Type I - step)

$$x_{t+1} = x_t - D_t\Big(G_t^T G_t + \tilde{M}D_t^T D_t\Big)^{-1}\Big(G_t^T\nabla f(x) + \frac{\tilde{M}\|\varepsilon_t\|}{2}\partial(|\alpha^\star|)\Big),$$

(Type II - step)

The type I is a quasi-Newton step with a symetrization of $G^T D$, along with a regularization, while the type II step can be seen as a quasi-Newton method with a regularization on $R^\dagger$, with a correction term on the gradient. The Hessian approximation therefore reads

$$B_t^{-1} = D_t\Big(H_t + \frac{MD_t^T D_t\|D_t\alpha^\star\|}{6}\Big)^{-1} D^T, \quad H_t = D_t\Big(G_t^T G_t + \tilde{M}D_t^T D_t\Big)^{-1} G_t^T.$$

Again, when the objective function is quadratic, $L = 0$ and therefore $M = 0$. Moreover, when $f$ is quadratic, the matrix multiplication $D^T G$ satisfies $D^T G + G^T D = 2D^T G$ as $D^T G$ becomes symmetric. Hence,

$$x_{t+1} = x_t - D_t\big(D_t^T G_t\big)^{-1} D_t^T\nabla f(x_t),$$

(Type I - quadratic)

$$x_{t+1} = x_t - D_t\big(G_t^T G_t\big)^{-1} G_t^T\nabla f(x_t),$$

(Type II quadratic)

The steps are *exactly* the type I and type II multisecant Broyden methods from eqs. (26) and (27), with the only difference that there is no initialization $H_0$ or $B_0$. Again, this is expected by construction of the method, where the initialization is estimated with a forward estimate (see section 5).

# C    Solving the sub-problems

**Solving the Type 1 Subproblem**    The Type 1 subproblem is a well-studied problem that involves minimizing a specific objective function. A method proposed by[46] has proven to be efficient for solving this problem. The method utilizes eigenvalue decomposition on a matrix to find the optimal solution. In this paper, the matrix involved in this problem is relatively small, therefore eigenvalue decomposition is not a concern even for large-scale problems. The subproblem aims to determine the norm of the solution, and this can be achieved through solving a system of nonlinear equations using bisection or secant method.

**Solving the Type 2 Subproblem**    The Type 2 subproblem can be formulated as a Second-Order Cone Program (SOCP). The objective function of this subproblem consists of three terms: a norm term, a sum of absolute values term, and a quadratic term. The norm term can be transformed using singular value decomposition, and the sum of absolute values term can be expressed as linear programming. The quadratic term can be simplified using a rotated quadratic cone. By utilizing these techniques, the Type 2 subproblem can be effectively solved using existing SOCP solvers.

## C.1    Solving the Type 1 Subproblem

The Type 1 subproblem can be expressed as follows:

$$\min_{\alpha} \nabla f(x)D\alpha + \frac{1}{2}\alpha^T H\alpha + \frac{M}{6}\|D\alpha\|^3,$$

where $H$ is symmetric but not necessarily positive definite. This problem has been well-studied, and [46] proposed an efficient method to solve it using eigenvalue decomposition on the matrix $H$. Although eigenvalue decomposition may be challenging for large-scale problems, it is not a concern here since $H \in \mathbb{R}^{N \times N}$, with a relatively small $N$ (e.g., $N = 25$ in the experiments).

In essence, the subproblem involves determining the norm of the solution $r = \|\alpha\|$. This can be accomplished through a simple bisection on the following system of nonlinear equations:

$$\left(H + \frac{MD^T Dr}{2}I\right)\alpha = -D^t\nabla f(x), \quad \|\alpha\| = r, \quad r \geq -\lambda_{\min}(H). \tag{28}$$

Interestingly, this problem is equivalent to the following formulation, as shown in Proposition 1:

$$\left(\Lambda + \frac{Mr}{2}I\right)\tilde{\alpha} = -V^T(D^T D)^{-1/2}D^t\nabla f(x), \ \|\alpha\| = r, \ r \geq -\lambda_{\min}(H), \ \tilde{\alpha} = V^T(D^T D)^{1/2}\alpha, \tag{29}$$

which involves the eigenvalue decomposition $(D^T D)^{-1/2}H(D^T D)^{-1/2} = V\Lambda V^T$.

**Proposition 1.** *Problems (28) and (29) are equivalent.*

*Proof.* The first step is to split $D^T D = (D^T D)^{1/2}(D^T D)^{1/2}$ and then employ an eigenvalue decomposition on $(D^T D)^{-1/2}H(D^T D)^{-1/2} = V\Lambda V^T$ (where $V$ is orthonormal due to the symmetry of the matrix):

$$\left(H + \frac{MD^T Dr}{2}I\right)\alpha = -D^t\nabla f(x)$$

$$\Leftrightarrow (D^T D)^{1/2}\left((D^T D)^{-1/2}H(D^T D)^{-1/2} + \frac{Mr}{2}I\right)(D^T D)^{1/2}\alpha = -D^t\nabla f(x)$$

$$\Leftrightarrow (D^T D)^{1/2}V\left(\Lambda + \frac{Mr}{2}I\right)V^T(D^T D)^{1/2}\alpha = -D^t\nabla f(x)$$

$$\Leftrightarrow \left(\Lambda + \frac{Mr}{2}I\right)V^T(D^T D)^{1/2}\alpha = -V^T(D^T D)^{-1/2}D^t\nabla f(x)$$

$$\Leftrightarrow \left(\Lambda + \frac{Mr}{2}I\right)\tilde{\alpha} = -V^T(D^T D)^{-1/2}D^t\nabla f(x).$$

$\square$

Once the eigenvalue decomposition is performed, the subproblem (29) becomes relatively simple since it involves solving a diagonal system of equations for a fixed value of $r$. The main objective is to find an interval $[r_{\min}, r_{\max}]$ that encompasses the optimal value $r = \|\alpha\|$. Once this interval is identified, a straightforward bisection or secant method can be employed to obtain the optimal solution.

**Finding initial bounds**  Starting with $r_{\min} = \max\{0, -\lambda_{\min(H)}\}$ and $r_{\max} = \max\{2r_{\min}, 1\}$,

$$\text{do } r_{\max} \leftarrow 2r_{\max} \quad \text{while } \|\tilde{\alpha}\| \geq r_{\max}.$$

where $\tilde{\alpha} = -\left(\Lambda + \frac{Mr_{\max}}{2}I\right)^{-1} V^T (D^T D)^{-1/2} D^t \nabla f(x)$. Increasing $r_{\max}$ increases the regularization, hence reduces the norm of $\tilde{\alpha}$.

**Finding $\alpha$**  After $r^\star$ has been found such that $|r^\star - \|\tilde{\alpha}\||$ is sufficiently small, the best $\alpha$ is simply

$$\alpha = (D^T D)^{-1/2} V \tilde{\alpha} = -(D^T D)^{-1/2} V \left(\Lambda + \frac{Mr^\star}{2}I\right)^{-1} V^T (D^T D)^{-1/2} D^t \nabla f(x).$$

In the case where the diagonal matrix is not invertible, which happens when $r^\star = r_{\min}$, it suffices to use the pseudo-inverse instead.

## C.2  Solving the Type 2 Subproblem

The Type 2 subproblem is given by:

$$\min_{\alpha} \underbrace{\|\nabla f(x) + G\alpha\|}_{\textbf{(a)}} + \frac{L}{2}\bigg(\underbrace{\sum_{i=1}^{N} |\alpha_i|\varepsilon_i + \underbrace{\|D\alpha\|^2}_{\textbf{(c)}}}_{\textbf{(b)}}\bigg). \tag{30}$$

Although it may not be immediately apparent, this subproblem can be formulated as a Second-Order Cone Program (SOCP) with $O(N)$ variables and constraints.

### C.2.1  Fundamentals of SOCP

SOCP solvers handle the following conic problems:

$$\min_{x, t_i, \omega_i} c_0 x + \sum_i c_i [t_i; \omega_i] \quad \text{subject to}$$

$$A_0 x + \sum_{i=1}^{k} A_i [t_i; \omega_i] = b \qquad \text{(SOCP Standard Matrix Form)}$$

$$x \geq 0$$

$$(t_i, \omega_i) \in \mathcal{K}_i \quad \Leftrightarrow t_i \geq \|\omega_i\|, \quad t \geq 0.$$

Here, $k$ represents the number of cones, and the cone $\mathcal{K}$ refers to the second-order cone, also known as the *Lorenz* cone.

A useful transformation is the *rotated quadratic cone*, defined as follows:

$$[a, b, c] \in \mathcal{K}_q \quad \Leftrightarrow \quad 2ab \geq \|c\|^2.$$

The rotated quadratic cone can be reformulated as a second-order cone using a linear transformation:

$$\text{if} \quad \begin{bmatrix} a \\ b \\ c \end{bmatrix} = \begin{bmatrix} \frac{1}{\sqrt{2}} & \frac{1}{\sqrt{2}} & 0 \\ \frac{1}{\sqrt{2}} & -\frac{1}{\sqrt{2}} & 0 \\ 0 & 0 & I_K \end{bmatrix} \begin{bmatrix} t \\ \omega^{(0)} \\ \omega \end{bmatrix} \quad \text{then} \quad (t, [\omega^{(0)}; \omega]) \in \mathcal{K} \quad \Leftrightarrow \quad [a, b, c] \in \mathcal{K}_q.$$

Thanks to this transformation, the rotated quadratic cone can be included in SOCP solvers.

### C.2.2 SOCP Formulation of the Type 2 Subproblem

The SOCP of (30) is composed of the three terms **a**, **b**, and **c**.

**Term (a)** Let $U_G \Sigma_G V_G^T$ be the singular value decomposition of $G$. Write $P_G = U_G U_G^T$ as the projector onto the columns of $G$. Then,

$$
\begin{aligned}
\|\nabla f(x) + R\alpha\| &= \|P_G \nabla f(x) + P_G G\alpha + (I - P_G)\nabla f(x)\| \\
&= \sqrt{\|P_G \nabla f(x) + R\alpha\|^2 + \|(I - P_G)\nabla f(x)\|^2} \\
&= \sqrt{\left\|U_G \left(U_G^T \nabla f(x) + \Sigma_G V_G^T \alpha\right)\right\|^2 + \|(I - P_G)\nabla f(x)\|^2} \\
&= \sqrt{\left\|U_G^T \nabla f(x) + \Sigma_G V_G^T \alpha\right\|^2 + \|(I - P_G)\nabla f(x)\|^2}
\end{aligned}
$$

Let the vector $\omega_1 = \left[U_G^T \nabla f(x) + \Sigma_G V\alpha;\ \|(I - P_G)\nabla f(x)\|\right]$. Hence,

$$
\|\nabla f(x) + G\alpha\| = \min_{t_1, \alpha, \omega_1} t_1 : (t_1, \omega_1) \in \mathcal{K}_L, \quad \omega_1 = \left[U_G^T \nabla f(x) + \Sigma_G V\alpha;\ \|(I - P_G)\nabla f(x)\|\right].
$$

**Term (b)** This term is standard in linear programming. Let $\alpha = \alpha_+ - \alpha_-$, with $\alpha_+, \alpha_- \geq 0$,

$$
\sum_{i=1}^N |\alpha_i| \varepsilon_i = \sum_{i=1}^N (\alpha_+ + \alpha_-)\varepsilon_i.
$$

**Term (c)** Let $U_D \Sigma_D V_D^T$ be the singular value decomposition of $D$. Using the rotated cone, the constraint can be written as

$$
2 t_3 b \geq \|U_D \Sigma_D V_D \alpha\|^2 = \|\Sigma_D V_D \alpha\|^2, \quad b = \frac{1}{2}.
$$

Using the transformation into a Lorenz cone, this is equivalent to

$$
\begin{bmatrix} 1 & 0 & 0 \\ 0 & 1 & 0 \\ 0 & 0 & \Sigma_D V_D^T \end{bmatrix} \begin{bmatrix} t_3 \\ b \\ \alpha \end{bmatrix} = \begin{bmatrix} \frac{1}{\sqrt{2}} & \frac{1}{\sqrt{2}} & 0 \\ \frac{1}{\sqrt{2}} & -\frac{1}{\sqrt{2}} & 0 \\ 0 & 0 & I_k \end{bmatrix} \begin{bmatrix} t_2 \\ \omega_2^{(0)} \\ \omega_2 \end{bmatrix}, \quad b = \frac{1}{2}, \quad (t_2, [\omega_2^{(0)}, \omega_2]) \in \mathcal{K}.
$$

**Simplification.** Note that, since $b = \frac{1}{2}$, the value can be immediately replaced. Same idea with $t_3$: the constraint is written as

$$
t_3 = \frac{t_2 + \omega_2^{(0)}}{\sqrt{2}}, \quad t_3 \geq 0.
$$

Since, by construction, $t_2 \geq \omega_2^{(0)}$ and $t_2 \geq 0$, $t_3$ always satisfies the condition, which means both $t_3$ and its constraint can be removed. The constraints thus simplify into

$$
\begin{bmatrix} \frac{1}{2} \\ 0 \end{bmatrix} + \begin{bmatrix} 0 \\ \Sigma_D V_D^T \end{bmatrix} [\alpha] = \begin{bmatrix} \frac{1}{\sqrt{2}} & -\frac{1}{\sqrt{2}} & 0 \\ 0 & 0 & I_k \end{bmatrix} \begin{bmatrix} t_2 \\ \omega_2^{(0)} \\ \omega_2 \end{bmatrix}, \quad (t_2, [\omega_2^{(0)}, \omega_2]) \in \mathcal{K}.
$$

**Final formulation** Gathering all terms, the final SOCP formulation reads

$$\text{minimize} \quad t_1 + \frac{L}{2}\left((\alpha_+ + \alpha_-)^T \varepsilon + t_2\right)$$

$$\text{subject to} \quad \omega_1 = \left[U_G^T \nabla f(x) + \Sigma_G V_G^T \alpha \; ; \; \|(I - P_G)\nabla f(x)\|\right],$$

$$\alpha_+, \alpha_- \geq 0$$

$$\alpha = \alpha_+ - \alpha_-$$

$$\begin{bmatrix} \mathbf{0}_{1 \times N} & -\frac{1}{\sqrt{2}} & \frac{1}{\sqrt{2}} & 0 \\ \Sigma_D V_D^T & \mathbf{0}_{N \times 1} & \mathbf{0}_{N \times 1} & -I_N \end{bmatrix} \begin{bmatrix} \alpha \\ t_2 \\ \omega_2^{(0)} \\ \omega_2 \end{bmatrix} = \begin{bmatrix} 0 \\ -\frac{1}{2} \\ \mathbf{0}_{N \times 1} \end{bmatrix}$$

$$(t_1, \omega_1) \in \mathcal{K}, \quad (t_2, [\omega_2^{(0)}; \omega_2]) \in \mathcal{K}_L, \quad t_2 \geq 0.$$

673 **Standard matrix formulation**  The SOCP can be written under the standard matrix form (SOCP
674 Standard Matrix Form). Let the variables

$$\alpha_+, \alpha_- \geq 0, \quad (t_1, \omega_1) \in \mathcal{K}_1, \quad (t_2, [\omega_2^{(0)} \omega_2]) \in \mathcal{K}_2,$$

675 where $t_1$, $t_2$, and $\omega_2^{(0)}$ are scalars, $\omega_2$, $\alpha_+$, and $\alpha_-$ are vectors of size $N$, and $\omega_1$ is a vector of size
676 $N + 1$. The SOCP matrices read

$$c_0 = \begin{bmatrix} \frac{L\varepsilon^T}{2} & \frac{L\varepsilon^T}{2} \end{bmatrix} \quad c_1 = \begin{bmatrix} 1 & \mathbf{0}_{1 \times N+1} \end{bmatrix} \quad c_2 = \begin{bmatrix} \frac{L}{2\sqrt{2}} & \frac{L}{2\sqrt{2}} & \mathbf{0}_{1 \times N} \end{bmatrix}$$

$$A_0 = \begin{bmatrix} -\Sigma_G V_G^T & \Sigma_G V_G^T \\ \mathbf{0}_{2 \times N} & \mathbf{0}_{2 \times N} \\ \Sigma_D V_D^T & -\Sigma_D V_D^T \end{bmatrix}$$

$$A_1 = \begin{bmatrix} \mathbf{0}_{N+1 \times 1} & I_{N+1 \times N+1} \\ \mathbf{0}_{N+1 \times 1} & \mathbf{0}_{N+1 \times N+1} \end{bmatrix}$$

$$A_2 = \begin{bmatrix} \mathbf{0}_{N+1 \times 1} & \mathbf{0}_{N+1 \times 1} & \mathbf{0}_{N+1 \times N} \\ -\frac{1}{\sqrt{2}} & \frac{1}{\sqrt{2}} & \mathbf{0}_{1 \times N} \\ \mathbf{0}_{N \times 1} & \mathbf{0}_{N \times 1} & -I_{N \times N} \end{bmatrix}$$

$$b = \begin{bmatrix} \nabla f(x)^T U_G & \|(I - P_R)\nabla f(x)\| & -\frac{1}{2} & \mathbf{0}_{N \times 1} \end{bmatrix}^T.$$

677 This completes the SOCP formulation of the type 2 subproblem.

 # D  Accelerated Algorithm

---

**Algorithm 6** Adaptive Accelerated Type-I Iterative Algorithm

---

**Require:** First-order oracle $f$, initial iterate and smoothness $x_0$, $M_0$, number of iterations $T$.

    $\lambda_0^{(1)} \leftarrow 0$, $\lambda_0^{(2)} \leftarrow 0$, $\Delta \leftarrow -\infty$, $(M_0)_t \leftarrow M_0$

    Initialize $G_0$, $D_0$, $\varepsilon_0$ (see section 5)

    $x_1, M_1 \leftarrow$ [algorithm 1]$(f, G_0, D_0, \varepsilon_0, x_0, (M_0)_0)$

    Initialize $\ell_0^{(0)} = f(x_1)$,   $\ell_0^{(1)} = 0$, $\Delta = \|x_1 - x_0\|$

    **for** $t = 1, \ldots, T-1$ **do**

        Update $G_t$, $D_t$, $\varepsilon_t$ (see section 5)

        Set $b_t \leftarrow \frac{(t+1)(t+2)}{2}$, $B_t \leftarrow \frac{t(t+1)(t+2)}{6}$, $\beta_t \leftarrow \frac{3}{t+3}$.

        Update $\ell_t^{(0)} \leftarrow \ell_{t-1}^{(0)} + b_{t-1}[f(x_t) - \nabla f(x_t)^T x_t]$,   $\ell_t^{(1)} \leftarrow \ell_{t-1}^{(1)} + b_{t-1}\nabla f(x_t)$

        **do**

            ValidBound $\leftarrow$ True

            Set $v_t \leftarrow \arg\min_v \phi_t(v)$ (See proposition 2).

            Let $y_t \leftarrow \frac{3}{t+3}v_t + \frac{k}{t+3}x_t$

            $\{x_{t+1}, \alpha_t\, M_{t+1}, \gamma_t, \text{ExitFlag}\} \leftarrow$ [algorithm 4]$(f, G_t, D_t, \varepsilon_t, y_t, (M_0)_t, \Delta)$

            %% Check if the next $\phi$ is still a lower bound for $f(x_{t+1})$

            Define $\phi_+ = \phi_t + b_t[f(x_{t+1} + \nabla f(x_{t+1})(x - x_{t+1})]$.

            Set $v_+ \leftarrow \arg\min_v \phi_+(v)$ (See proposition 2).

            **if** $\Phi_+(v_+) \leq f(x_{t+1})$ **then**    %% Parameters adjustment if needed

                ValidBound $\leftarrow$ False    %% Unsuccessful iteration: $\phi_{t+1}(v_{t+1}) \geq f(x_{t+1})$.

                **if** ExitFlag is LargeStep **then**

                    **If** $\lambda_t^{(2)} = 0$ **then** $\lambda_t^{(2)} \leftarrow \dfrac{16}{9}\dfrac{(b_t\|\nabla f(x_{t+1})\|)^3}{(B_{t+1}\nabla f(x_{t+1})^T D_t\alpha_t)^2}$. **Else,** $\lambda_t^{(2)} \leftarrow 2\lambda_t^{(2)}$.

                **else** %% Exitflag is SmallStep

                    **If** $\lambda_t^{(1)} = 0$ **then** $\lambda_t^{(1)} \leftarrow \dfrac{-b_t^2\|\nabla f(x_{t+1})\|^2}{2B_{t+1}\nabla f(x_{t+1})^T D_t\alpha_t}$. **Else,** $\lambda_t^{(1)} \leftarrow 2\lambda_t^{(1)}$.

                **end if**

                **if** $(M_0)_{t+1} < M_{t+1}$ **then** $(M_0)_{t+1} \leftarrow M_{t+1}\left(\frac{\|\varepsilon_t\|}{\|D_t\|} + \frac{\|D_t\alpha_t\|}{2}\right)$ %% Rescaling

                **end if**

            **else**

                $\{\lambda_{t+1}^{(1)}, \lambda_{t+1}^{(2)}\} \leftarrow \{\lambda_t^{(1)}, \lambda_t^{(2)}\}$,  $(M_0)_{t+1} \leftarrow \frac{M_{t+1}}{2}$   %% Successful iteration

            **end if**

         **while** ValidBound is False

    **end for**

    **return** $x_T$

---

**Proposition 2.** *Let $v_t$ be the the minimizer of*

$$\phi_t(v) = \ell_t^{(0)} + \left[\ell_t^{(1)}\right]^T v + \frac{\lambda_t^{(1)}}{2}\|v - x_0\|^2 + \frac{\lambda_t^{(2)}}{6}\|v - x_0\|^3.$$

*where $\lambda_t^{(1,2)} \geq 0$. Let $r_t = \|v_t - x_0\|$. Then,*

$$r_t = \|v_t - x_0\| = \begin{cases} 0 & \text{if } \lambda_t^{(1)} = \lambda_t^{(2)} = 0 \\ \frac{\|\ell_t^{(1)}\|}{\lambda_t^{(1)}} & \text{if } \lambda_t^{(1)} > 0 \text{ and } \lambda_t^{(2)} = 0 \\ \frac{-\lambda_t^{(1)} + \sqrt{[\lambda_t^{(1)}]^2 + 2\lambda_t^{(2)}\|\ell_k\|}}{\lambda_2^{(2)}} & \text{if } \lambda_t^{(2)} > 0 \end{cases}$$

$$v_t = \arg\min \Phi_t(x) = x_0 - r_t\frac{\ell_t^{(1)}}{\|\ell_t^{(1)}\|}$$

# E    Additional Numerical Experiments

This section presents additional numerical experiments.

**Methods**    The methods compared are the type 1 and type 2 steps with the following strategies: *Iterate only*, *Forward estimate only*, *Greedy* (refer to section 5), and the accelerated type 1 method with the strategy *forward estimate only*. The batch methods are not included as they perform poorly in terms of the number of oracle calls. The baseline is the L-BFGS method from `minFunc` [53].

**Method parameters**    In all experiments, the memory of the methods is set to $N = 25$. The parameters of the L-BFGS are left untouched except for the memory. The initial smoothness parameter is set to 1 for the type 1 and type 2 methods. The initial point is randomly generated by the function `randn()` in Matlab, with a seed of $0$.

**Functions**    The minimized problems are: square loss with cubic regularization, logistic loss with small quadratic regularization, and the generalized Rosenbrock function. The regularization parameter of the square loss is set to $1e - 3$ times the norm of the Hessian, and the regularization of the logistic loss is set to $1e - 10$ times the square norm of the feature matrix.

**Dataset**    The datasets for the square loss and the logistic loss are Madelon [33], Sido0 [34], and Marti2 [34] datasets.

**Post-processing**    The dataset matrix is normalized by its norm, and a feature vector of ones is added to the data matrix.

## E.1    Nonconvex optimization

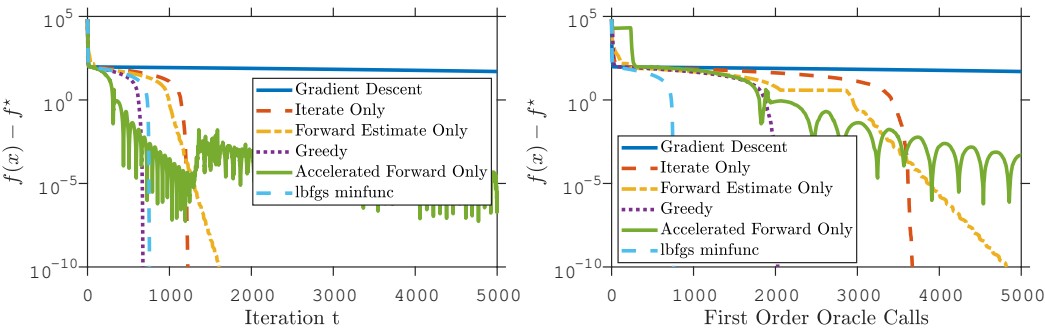

Figure 2: Comparison of type 1 methods on the Generalized Rosenbrock function in $\mathbb{R}^{100}$

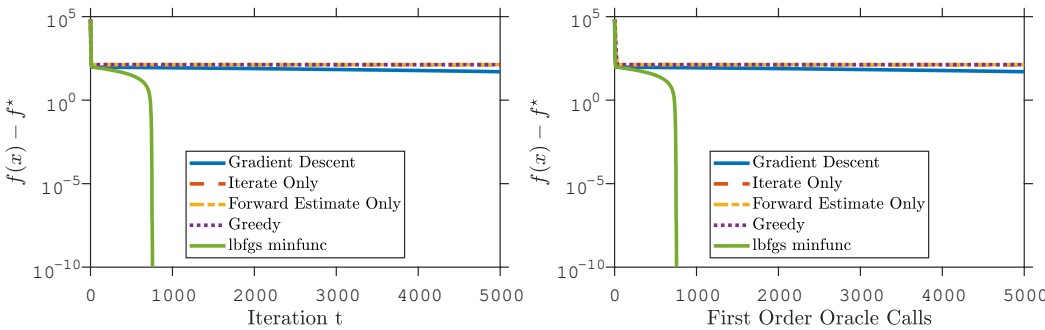

Figure 3: Comparison of type 2 methods on the Generalized Rosenbrock function in $\mathbb{R}^{100}$

 ## E.2 Comparison of Type 1 Methods on Convex Problems

 ### E.2.1 Square loss and cubic regularization

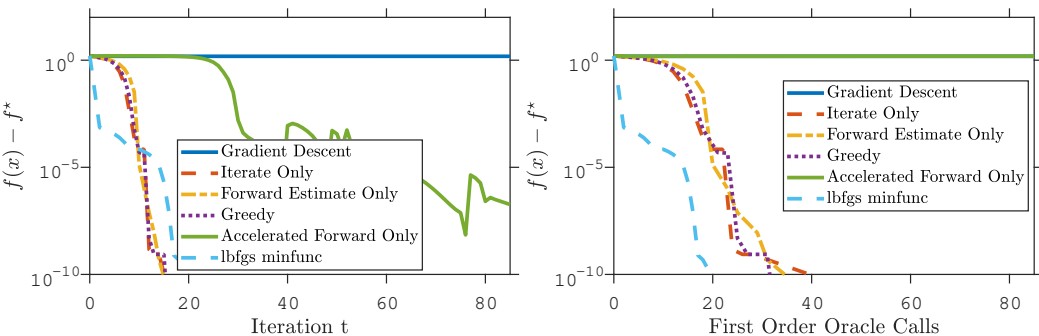

Figure 4: Comparison of type 1 methods: Square loss and cubic regularization on Madelon dataset

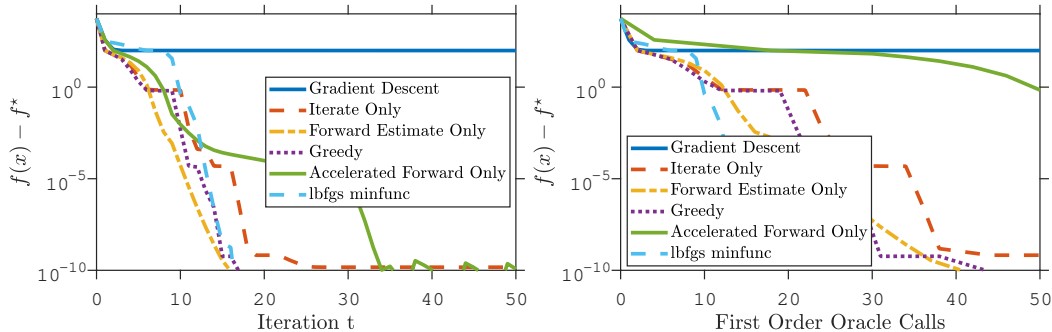

Figure 5: Comparison of type 1 methods: Square loss and cubic regularization on sido0 dataset

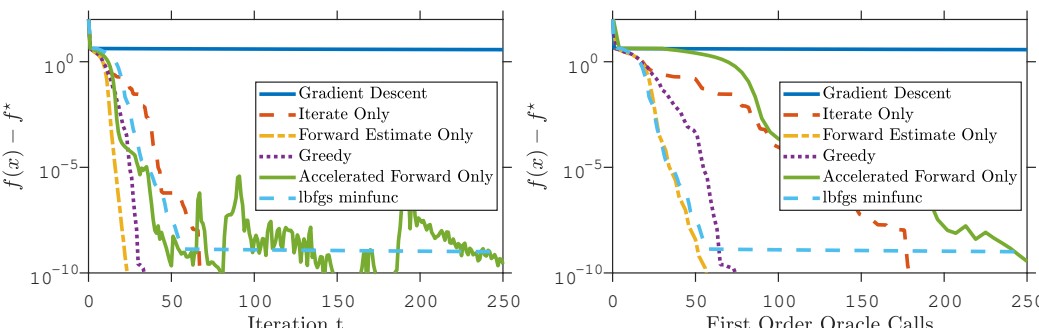

Figure 6: Comparison of type 1 methods: Square loss and cubic regularization on marti2 dataset

 **E.2.2 Logistic regression**

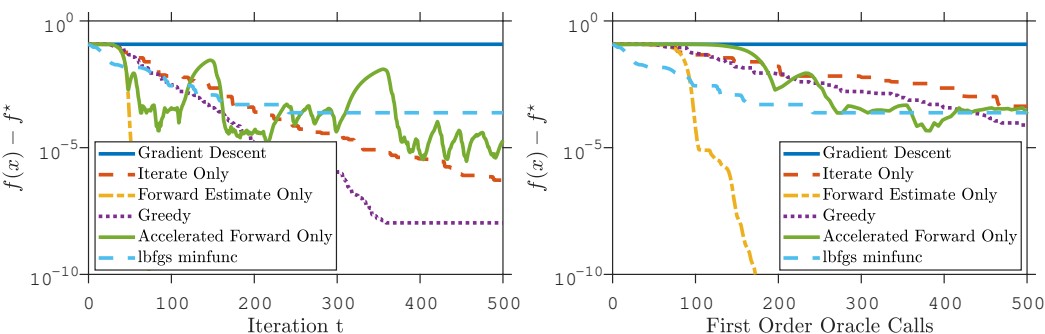

Figure 7: Comparison of type 1 methods: Logistic loss and cubic regularization on Madelon dataset

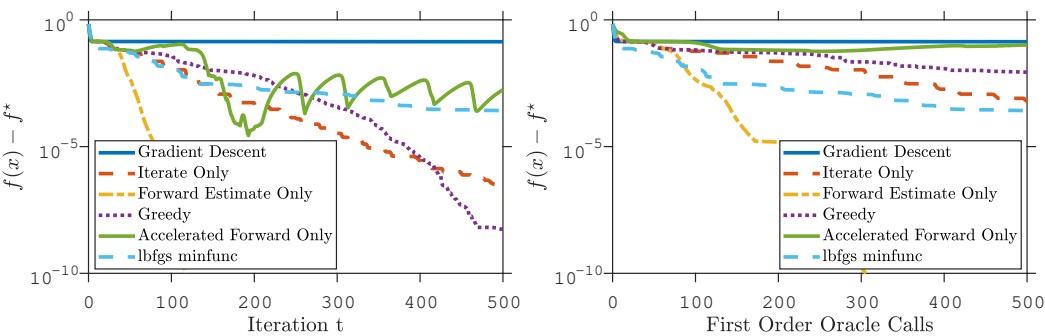

Figure 8: Comparison of type 1 methods: Logistic loss and cubic regularization on sido0 dataset

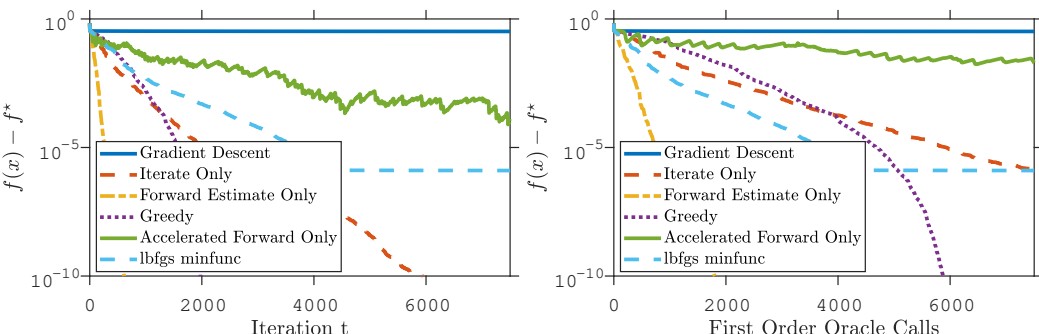

Figure 9: Comparison of type 1 methods: Logistic loss and cubic regularization on marti2 dataset

 ### E.3 Comparison of Type 2 Methods on Convex Problems

 ### E.3.1 Square loss and cubic regularization

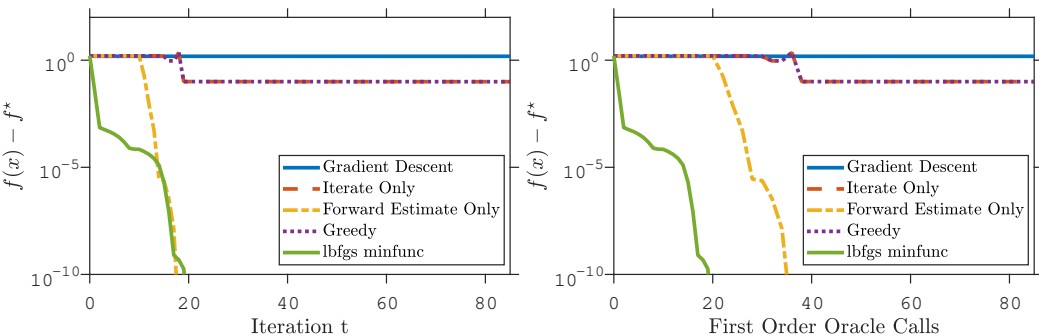

Figure 10: Comparison of type 2 methods: Square loss and cubic regularization on Madelon dataset

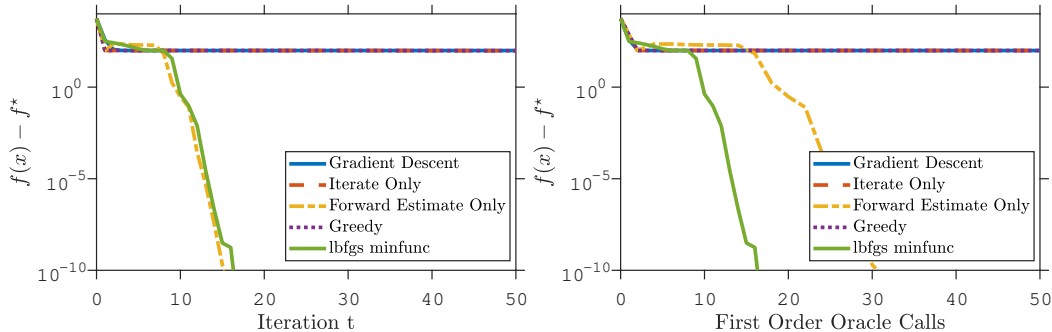

Figure 11: Comparison of type 2 methods: Square loss and cubic regularization on sido0 dataset

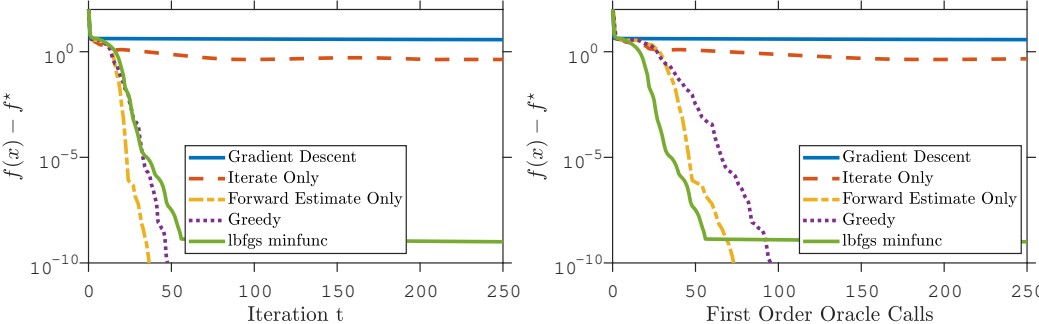

Figure 12: Comparison of type 2 methods: Square loss and cubic regularization on marti2 dataset

 ## E.3.2 Logistic regression

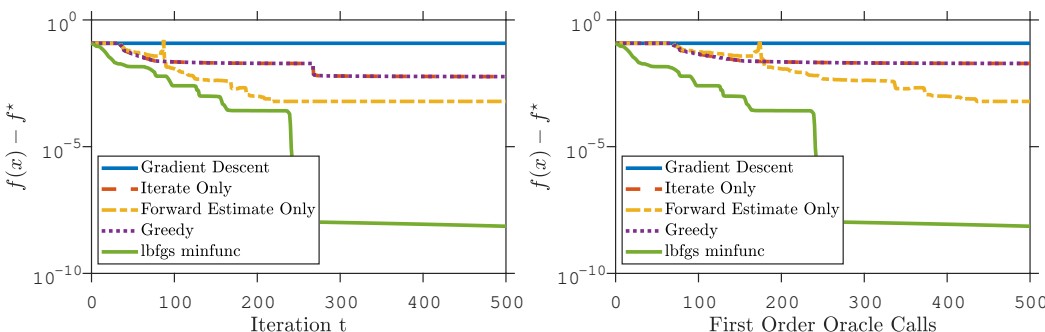

Figure 13: Comparison of type 2 methods: Logistic loss and cubic regularization on Madelon dataset

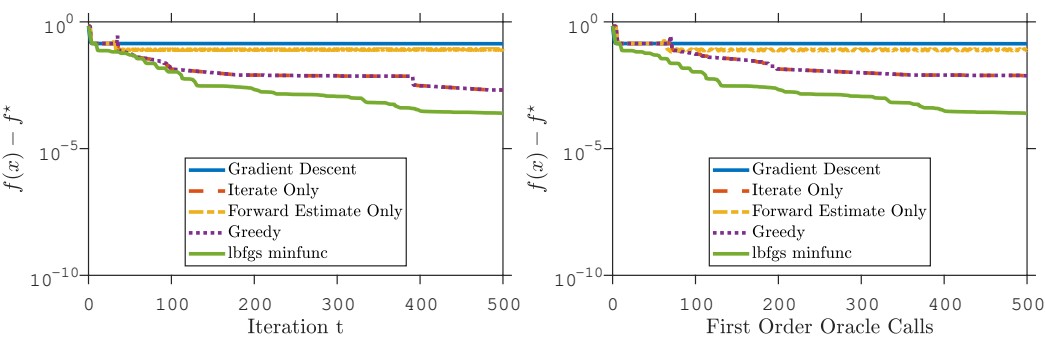

Figure 14: Comparison of type 2 methods: Logistic loss and cubic regularization on sido0 dataset

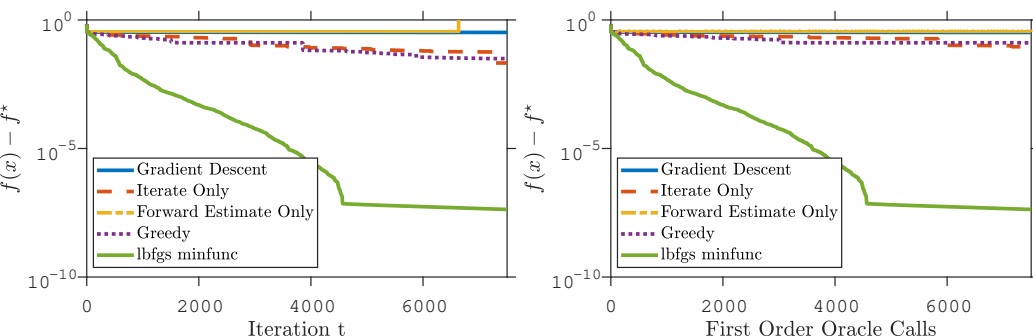

Figure 15: Comparison of type 2 methods: Logistic loss and cubic regularization on marti2 dataset

 # F   Missing proofs

 ## F.1   Technical results

 In this section, the following definitions simplify the notations:

$$D_\dagger = (D^T D)^{-1} D^T, \tag{31}$$

$$D_\dagger^T = D(D^T D)^{-1}, \tag{32}$$

$$\kappa_D = \|D_\dagger\| \|D\|, \tag{33}$$

$$\tilde{H} = D_\dagger^T H D_\dagger \qquad \text{where } H \text{ is defined in (10).} \tag{34}$$

 Note that the pseudo inverse $D_\dagger$ exists under Requirement 3.

 **Proposition 3.** *The first-order and second-order conditions of the subproblem in algorithm 1 read*

$$D^T \nabla f(x) + H\alpha + \frac{M}{2} D^T D\alpha \|D\alpha\| = 0, \tag{35}$$

$$H + \frac{M}{2} D^T D \|D\alpha\| \succeq 0. \tag{36}$$

 *Proof.* See [44], equation (3.3), and [46], equation (2.7). ☐

 **Proposition 4.** *Let $f$ satisfies Assumption 1 and $B \in \mathbb{R}^{d \times d}$ be any matrix. Then,*

$$\|\nabla f(x) + BD\alpha - \nabla f(x_+)\| \le \frac{L}{2} \|D\alpha\|^2 + \|[B - \nabla^2 f(x)]D\alpha\|.$$

 *Proof.* The result follows directly from (5),

$$\|\nabla f(x) + BD\alpha - \nabla f(x_+)\| \le \|\nabla f(x) + \nabla^2 f(x)D\alpha - \nabla f(x_+)\| + \|BD\alpha - \nabla^2 f(x)D\alpha\|$$

$$\le \frac{L}{2} \|D\alpha\|^2 + \|[B - \nabla^2 f(x)]D\alpha\|.$$

 ☐

 **Proposition 5.** *Assume the matrix $D$ satisfies Requirement 1b, and $\alpha$ satisfies the first-order condi-*
 *tion (35). Let $\tilde{H}$ be defined in (34). Then,*

$$\|\nabla f(x) + BD\alpha - \nabla f(x_+)\| = \|(\tilde{H} - B + \frac{M\|D\alpha\|}{2})D\alpha + \nabla f(x_+)\|$$

 *Proof.* The following equation follows from the optimality condition multiplied by $D(D^T D)^{-1}$,
 writing $P = DD_\dagger = D_\dagger^T D^T$, assuming $P\nabla f(x) = \nabla f(x)$,

$$\nabla f(x) + (\tilde{H} + \frac{M\|D\alpha\|}{2})D\alpha = 0.$$

 It suffices to replace $\nabla f(x)$. ☐

 **Proposition 6.** *Assume $D$ satisfies Requirement 1b. Let $\tilde{H}$ be defined in (34). Then, if $B = \tilde{H} - M\gamma$*
 *in proposition 4, the following holds:*

$$\|[B - \nabla^2 f(x)]D\alpha\| \le \|D\alpha\| \left( \frac{L}{2} \|D_\dagger\| \|\varepsilon\| + \|(I - P)\nabla^2 f(x)P\| + M \left\| D_\dagger^T D_\dagger \frac{\|D\| \|\varepsilon\|}{2} - \gamma P \right\| \right)$$

 *Proof.* Since

$$\nabla^2 f(x)D\alpha = P\nabla^2 f(x)PD\alpha + (I - P)\nabla^2 f(x)PD\alpha,$$

 where $P = D(D^T D)^{-1} D^T$, and because $PD = D$ and

$$\tilde{H} = D_\dagger^T \left( \frac{D^T G + G^T D}{2} + \frac{M\|D\| \|\varepsilon\|}{2} \right) D_\dagger = \frac{PGD_\dagger + D_\dagger^T G^T P}{2} + D_\dagger^T D_\dagger \frac{M\|D\| \|\varepsilon\|}{2},$$

the inequality becomes

$$\|[B - \nabla^2 f(x)]D\alpha\| \leq \left\|\left(\frac{PGD_\dagger + D_\dagger^T G^T P}{2} - P\nabla^2 f(x)P\right)D\alpha\right\| \tag{37}$$

$$+ \left\|\left(D_\dagger^T D_\dagger \frac{M\|D\|\|\varepsilon\|}{2} - M\gamma P - (I - P)\nabla^2 f(x)P\right)D\alpha\right\| \tag{38}$$

The term (38) can be decomposed into

$$\left\|\left(D_\dagger^T D_\dagger \frac{M\|D\|\|\varepsilon\|}{2} - M\gamma P - (I - P)\nabla^2 f(x)P\right)D\alpha\right\|$$
$$= \left\|P\left(\left(D_\dagger^T D_\dagger \frac{M\|D\|\|\varepsilon\|}{2} - M\gamma\right)D\alpha - (I - P)\nabla^2 f(x)PD\alpha\right)\right\|$$

Since $P$ satisfies $\|Pv_1 + (I - P)v_2\| = \|Pv_1\| + \|(I - P)v_2\|$,

$$\|[B - \nabla^2 f(x)]D\alpha\| \leq \left\|\left[\frac{PGD_\dagger + D_\dagger^T G^T P}{2} - P\nabla^2 f(x)P\right]D\alpha\right\|$$
$$+ M\|D\alpha\| \left\|D_\dagger^T D_\dagger \frac{\|D\|\|\varepsilon\|}{2} - P\gamma\right\| \tag{39}$$
$$+ \|D\alpha\|\|(I - P)\nabla^2 f(x)P\|.$$

It remains to bound the first from (37). Since $D^T D_\dagger = D_\dagger^T D^T = P$, $D_\dagger D = I$, $PD = D$, and $\|P\| = 1$,

$$\left\|\left[\frac{PGD_\dagger + D_\dagger^T G^T P}{2} - P\nabla^2 f(x)P\right]D\alpha\right\|$$
$$\leq \frac{1}{2}\left(\|(PGD_\dagger - P\nabla^2 f(x)P)D\alpha\| + \|(D_\dagger^T G^T P - P\nabla^2 f(x)P)D\alpha\|\right)$$
$$\leq \frac{1}{2}\left(\|G\alpha - \nabla^2 f(x)D\alpha\| + \|D_\dagger\|\|(G^T - D^T\nabla^2 f(x))D\alpha\|\right)$$

Using inequality (8) for the first term and (9) for second gives

$$\left\|\left[\frac{PGD_\dagger + D_\dagger^T G^T P}{2} - P\nabla^2 f(x)P\right]D\alpha\right\| \leq \frac{1}{2}\left(\frac{L}{2}\sum_{i=1}^{N}|\alpha_i|\varepsilon_i + \|D_\dagger\|\frac{L\|D\alpha\|}{2}\|\varepsilon\|\right)$$

Because $\sum_{i=1}^{N}|\alpha_i|\varepsilon_i \leq \|\alpha\|\|\varepsilon\| \leq \|D_\dagger\|\|D\alpha\|$,

$$\left\|\left[\frac{PGD_\dagger + D_\dagger^T G^T P}{2} - P\nabla^2 f(x)P\right]D\alpha\right\| \leq \frac{L}{2}\|D_\dagger\|\|\varepsilon\|\|D\alpha\|.$$

Injecting this result back in (39) gives the desired result,

$$\|[B - \nabla^2 f(x)]D\alpha\| \leq \|D\alpha\|\left(\frac{L\|D_\dagger\|\|\varepsilon\|}{2} + \|(I - P)\nabla^2 f(x)P\|\right)$$
$$+ M\|D\alpha\|\left\|D_\dagger^T D_\dagger \frac{\|D\|\|\varepsilon\|}{2} - I\gamma\right\|.$$

$\square$

**Proposition 7.** *Under the assumptions of propositions 4 to 6, setting $\gamma = 0$ gives*

$$\left\|\frac{M\|D\alpha\|}{2}D\alpha + \nabla f(x_+)\right\|$$
$$\leq \frac{L}{2}\|D\alpha\|^2 + \|D\alpha\|\left(\frac{\|\varepsilon\|}{\|D\|}\left(\frac{L + M\kappa_D}{2}\right)\kappa_D + \|(I - P)\nabla^2 f(x)P\|\right). \tag{40}$$

*Proof.* Using propositions 4 and 5, setting $B = \tilde{H} - M\gamma P$ and $\gamma = 0$ gives

$$\left\| \frac{M\|D\alpha\|}{2} D\alpha + \nabla f(x_+) \right\|$$

$$\leq \frac{L\|D\alpha\|^2}{2} + \|D\alpha\| \left( \frac{L}{2}\|D_\dagger\|\|\varepsilon\| + \|(I-P)\nabla^2 f(x)P\| + M \left\| D_\dagger^T D_\dagger \frac{\|D\|\|\varepsilon\|}{2} \right\| \right)$$

Moreover,

$$\left\| D_\dagger^T D_\dagger \frac{\|D\|\|\varepsilon\|}{2} \right\| \leq \frac{\|D_\dagger\|^2\|D\|\|\varepsilon\|}{2}$$

All together, and by definition of $\kappa_D$ (33),

$$\left\| \frac{M\|D\alpha\|}{2} D\alpha + \nabla f(x_+) \right\|$$

$$\leq \frac{L}{2}\|D\alpha\|^2 + \|D\alpha\| \left( \frac{\|\varepsilon\|}{\|D\|} \left( \frac{L + M\kappa_D}{2} \right) \kappa_D + \|(I-P)\nabla^2 f(x)P\| \right).$$

$\square$

**Proposition 8.** *Under the assumptions of propositions 4 to 6, setting $\gamma = \frac{\|D\alpha\|}{2}$ gives*

$$\|\nabla f(x_+)\| \leq \frac{L+M}{2}\|D\alpha\|^2 + \|D\alpha\| \left( \frac{\|\varepsilon\|}{\|D\|} \left( \frac{L + M\kappa_D}{2} \right) \kappa_D + \|(I-P)\nabla^2 f(x)P\| \right). \quad (41)$$

*Proof.* Using propositions 4 and 5, setting $B = \tilde{H} - M\gamma I$ and $\gamma = \frac{\|D\alpha\|}{2}$ gives

$$\|\nabla f(x_+)\|$$

$$\leq \frac{L\|D\alpha\|^2}{2} + \|D\alpha\| \left( \frac{L}{2}\|D_\dagger\|\|\varepsilon\| + \|(I-P)\nabla^2 f(x)P\| + M \left\| D_\dagger^T D_\dagger \frac{\|D\|\|\varepsilon\|}{2} - \frac{\|D\alpha\|}{2}P \right\| \right)$$

Moreover,

$$\left\| D_\dagger^T D_\dagger \frac{\|D\|\|\varepsilon\|}{2} - \frac{\|D\alpha\|}{2}P \right\| \leq \frac{\|D_\dagger\|^2\|D\|\|\varepsilon\|}{2} + \frac{\|D\alpha\|}{2}$$

All together, and by definition of $\kappa_D$ (33),

$$\|\nabla f(x_+)\| \leq \frac{L+M}{2}\|D\alpha\|^2 + \|D\alpha\| \left( \frac{\|\varepsilon\|}{\|D\|} \left( \frac{L + M\kappa_D}{2} \right) \kappa_D + \|(I-P)\nabla^2 f(x)P\| \right)$$

$\square$

**Proposition 9.** *Let Assumption 1 and Requirements 1b to 3 hold. Then, $\forall y \in \mathbb{R}^d$, algorithm 1 ensures*

$$f(x_+) \leq f(y) + \frac{M+L}{6}\|y-x\|^3 + \frac{\|y-x\|^2}{2} \left( \|\nabla^2 f(x) - P\nabla^2 f(x)P\| + \delta \frac{L\kappa + M\kappa^2}{2} \right)$$

*Proof.* The output of algorithm 1 ensures that

$$f(x_+) \leq$$

$$\min_\alpha f(x) + \nabla f(x)^T D\alpha + \frac{1}{2}(D\alpha)^T \nabla^2 f(x)D\alpha + \frac{1}{2}\alpha^T \left( H - D^T\nabla^2 f(x)D \right)\alpha + \frac{M}{6}\|D\alpha\|^3$$

However, by the definition of $H$ (10),

$$\frac{1}{2}\alpha^T \left( H - D^T\nabla^2 f(x)D \right)\alpha$$

$$\leq \frac{1}{2} \left( \alpha^T \left( \frac{G^T D + D^T G}{2} - D^T\nabla^2 f(x)D \right)\alpha + \|\alpha\|^2 \frac{M\|D\|\|\varepsilon\|}{2} \right)$$

$$\leq \frac{1}{2} \left( \alpha^T \left( \frac{G^T D + D^T G}{2} - D^T\nabla^2 f(x)D \right)\alpha + \|D^\dagger\|^2\|D\alpha\| \frac{M\|D\|\|\varepsilon\|}{2} \right)$$

$$= \frac{1}{2} \left( (D\alpha)^T \left( G - \nabla^2 f(x)D \right)\alpha + \|D^\dagger\|^2\|D\alpha\| \frac{M\|D\|\|\varepsilon\|}{2} \right).$$

The last equality comes from the fact that

$$\alpha^T \left(D^T G\right) \alpha = \alpha^T \left(\frac{D^T G + G^T D}{2} + \frac{D^T G - G^T D}{2}\right) \alpha = \alpha^T \left(\frac{D^T G + G^T D}{2}\right) \alpha.$$

Now, using (8) with $w = D\alpha$ gives

$$\frac{1}{2}\alpha^T \left(H - D^T \nabla^2 f(x)D\right) \alpha \leq \frac{L\|D\alpha\|}{4} \sum_{i=1}^N |\alpha_i|\varepsilon_i + \|D^\dagger\|^2 \|D\alpha\|\frac{M\|D\|\|\varepsilon\|}{4}.$$

Finally, since

$$\sum_{i=1}^N |\alpha_i|\varepsilon_i \leq \|\alpha\|\|\varepsilon\| \leq \|D^\dagger\|\|D\alpha\|\|\varepsilon\|,$$

the inequality becomes

$$\frac{1}{2}\alpha^T \left(H - D^T \nabla^2 f(x)D\right) \alpha \leq \frac{\|D\alpha\|^2}{4} \left(L\|D^\dagger\|\|\varepsilon\| + M\|D^\dagger\|^2\|D\|\|\varepsilon\|\right)$$
$$= \frac{\|D\alpha\|^2}{4}\frac{\|\varepsilon\|}{\|D\|} \left(L\kappa_D + M\kappa_D^2\right).$$

All together,

$$f(x_+)$$
$$\leq \min_\alpha f(x) + \nabla f(x)^T D\alpha + \frac{1}{2}(D\alpha)^T \nabla^2 f(x)D\alpha + \frac{1}{2}\alpha^T \left(H - D^T \nabla^2 f(x)D\right) \alpha + \frac{M}{6}\|D\alpha\|^3$$
$$\leq \min_\alpha f(x) + \nabla f(x)^T D\alpha + \frac{1}{2}(D\alpha)^T \nabla^2 f(x)D\alpha + \frac{\|D\alpha\|^2}{4}\frac{\|\varepsilon\|}{\|D\|} \left(L\kappa_D + M\kappa_D^2\right) + \frac{M}{6}\|D\alpha\|^3$$

Now, by Requirement 3, for all $y$, one can find $\alpha$ such that

$$D\alpha = P(y - x) = DD^\dagger(y - x).$$

Indeed, multiplying both sides by $D^\dagger$ gives

$$\alpha = D^\dagger(y - x).$$

Therefore, the minimum can be written as a function of $y$ instead of $\alpha$,

$$f(x_+) \leq \min_{y \in \mathbb{R}^d} \ f(x) + \nabla f(x)^T P(y - x) + \frac{1}{2}(P(y - x))^T \nabla^2 f(x)P(y - x)$$
$$+ \frac{\|P(y - x)\|^2}{4}\frac{\|\varepsilon\|}{\|D\|} \left(L\kappa_D + M\kappa_D^2\right) + \frac{M}{6}\|P(y - x)\|^3. \tag{42}$$

Since $P\nabla f(x) = \nabla f(x)$ by Requirement 1b, and using the crude bound $\|P(y - x)\| \leq \|y - x\|$,

$$f(x_+) \leq \min_{y \in \mathbb{R}^d} \ f(x) + \nabla f(x)^T(y - x) + \frac{1}{2}(y - x)^T \nabla^2 f(x)(y - x)$$
$$+ \frac{1}{2}(y - x)\left[\nabla^2 f(x) - P\nabla^2 f(x)P\right](y - x)$$
$$+ \frac{\|y - x\|^2}{4}\frac{\|\varepsilon\|}{\|D\|} \left(L\kappa_D + M\kappa_D^2\right) + \frac{M}{6}\|y - x\|^3.$$

Using the lower bound (6),

$$f(x) + \nabla f(x)^T(y - x) + \frac{1}{2}(y - x)^T \nabla^2 f(x)(y - x) - \frac{L}{6}\|y - x\|^3 \leq f(y),$$

the crude bound $(y - x)\left[\nabla^2 f(x) - P\nabla^2 f(x)P\right](y - x) \leq \|\nabla^2 f(x) - P\nabla^2 f(x)P\|\|y - x\|^2$, and Requirements 2 and 3 lead to the desired result,

$$f(x_+) \leq f(y) + \frac{M + L}{6}\|y - x\|^3 + \frac{\|y - x\|^2}{2} \left(\|\nabla^2 f(x) - P\nabla^2 f(x)P\| + \delta\frac{L\kappa + M\kappa^2}{2}\right)$$

$\square$

**Proposition 10.** *Let Assumption 1 and Requirements 1a, 2 and 3 hold. Then, $\forall y \in \mathbb{R}^d$, algorithm 1 ensures*

$$\mathbb{E}f(x_+) \le \left(1 - \frac{N}{d}\right)f(x) + \frac{N}{d}f(y) + \frac{N}{d}\frac{(M+L)}{6}\|y - x\|^3$$

$$+ \frac{N}{d}\frac{\|y - x\|^2}{2}\left(\delta\frac{L\kappa + M\kappa^2}{2} + \frac{(d - N)}{d}\|\nabla^2 f(x)\|\right)$$

*Proof.* The proof is the same as for proposition 9, until equation (42),

$$f(x_+) \le \min_{y \in \mathbb{R}^d}\ f(x) + \nabla f(x)^T P(y - x) + \frac{1}{2}(P(y - x))^T \nabla^2 f(x)P(y - x)$$

$$+ \frac{\|P(y - x)\|^2}{4}\frac{\|\varepsilon\|}{\|D\|}\left(L\kappa_D + M\kappa_D^2\right) + \frac{M}{6}\|P(y - x)\|^3.$$

With Requirement 1a, the following relations hold (see [35, lemma 5.7])

$$\mathbb{E}[\|P(y - x)\|^2] = (y - x)^T \mathbb{E}[P](y - x) = \frac{N}{d}\|y - x\|^2, \tag{43}$$

$$\mathbb{E}[\|P(y - x)\|^3] \le \mathbb{E}[\|P(y - x)\|^2]\|y - x\| = \frac{N}{d}\|y - x\|^2, \tag{44}$$

$$\mathbb{E}[(y - x)^T P\nabla^2 f(x)P(y - x)] \le \frac{N^2}{d^2}(y - x)\nabla^2 f(x)(y - x) + \frac{N(d - N)}{d^2}\|\nabla^2 f(x)\|\|y - x\|^2 \tag{45}$$

Hence, removing the minimum and taking the expectation of (42) gives

$$\mathbb{E}f(x_+) \le f(x) + \frac{N}{d}\nabla f(x)^T(y - x)$$

$$+ \frac{1}{2}\left(\frac{N^2}{d^2}(y - x)\nabla^2 f(x)(y - x) + \frac{N(d - N)}{d^2}\|\nabla^2 f(x)\|\|y - x\|^2\right)$$

$$+ \frac{N}{d}\frac{\|y - x\|^2}{4}\frac{\|\varepsilon\|}{\|D\|}\left(L\kappa_D + M\kappa_D^2\right) + \frac{N}{d}\frac{M}{6}\|y - x\|^3.$$

Using the lower bound from (6)

$$\frac{1}{2}(y - x)\nabla^2 f(x)(y - x) \le f(y) + \frac{L}{6}\|y - x\|^3 - f(x) - \nabla f(x)(y - x)$$

in the inequality over the expectation gives

$$\mathbb{E}f(x_+) \le f(x) + \frac{N}{d}\nabla f(x)^T(y - x)$$

$$+ \frac{N^2}{d^2}\left(f(y) + \frac{L}{6}\|y - x\|^3 - f(x) - \nabla f(x)(y - x)\right)$$

$$+ \frac{1}{2}\frac{N(d - N)}{d^2}\|\nabla^2 f(x)\|\|y - x\|^2$$

$$+ \frac{N}{d}\frac{\|y - x\|^2}{4}\frac{\|\varepsilon\|}{\|D\|}\left(L\kappa_D + M\kappa_D^2\right) + \frac{N}{d}\frac{M}{6}\|y - x\|^3.$$

After simplification,

$$\mathbb{E}f(x_+) \le \left(1 - \frac{N^2}{d^2}\right)f(x) + \frac{N^2}{d^2}f(y) + \frac{N}{d}\left(1 - \frac{N}{d}\right)\nabla f(x)^T(y - x)$$

$$+ \frac{1}{2}\frac{N(d - N)}{d^2}\|\nabla^2 f(x)\|\|y - x\|^2$$

$$+ \frac{N}{d}\frac{\|y - x\|^2}{4}\frac{\|\varepsilon\|}{\|D\|}\left(L\kappa_D + M\kappa_D^2\right) + \left(\frac{N^2 L}{6d^2} + \frac{NM}{6d}\right)\|y - x\|^3.$$

To simplify the expression, since $N \leq d$,

$$\left(\frac{N^2 L}{6d^2} + \frac{NM}{6d}\right) \|y - x\|^3 \leq \frac{N(M + L)}{6d} \|y - x\|^3.$$

Finally, since the function is convex,

$$\frac{N}{d}\left(1 - \frac{N}{d}\right)\nabla f(x)^T(y - x) \leq \frac{N}{d}\left(1 - \frac{N}{d}\right)(f(y) - f(x)).$$

From this last relation, Requirement 2 and Requirement 3 comes the desired result,

$$\mathbb{E}f(x_+) \leq \left(1 - \frac{N}{d}\right)f(x) + \frac{N}{d}f(y) + \frac{N(M + L)}{6d}\|y - x\|^3$$
$$+ \frac{\|y - x\|^2}{2}\left(\frac{N}{d}\delta\frac{L\kappa + M\kappa^2}{2} + \frac{N(d - N)}{d^2}\|\nabla^2 f(x)\|\right)$$

$\square$

**Proposition 11.** *Under the assumptions of propositions 4 to 6, setting*

$$\gamma \geq \frac{1}{4}\frac{\|\varepsilon\|}{\|D\|}(1 + \kappa_D^2)$$

*gives*

$$\left\|M\left(\gamma + \frac{\|D\alpha\|}{2}\right)D\alpha + \nabla f(x_+)\right\| \tag{46}$$
$$\leq \|D\alpha\|\left(\frac{L}{2}\|D\alpha\| + \frac{L}{2}\frac{\|\varepsilon\|}{\|D\|}\kappa_D + \|(I - P)\nabla^2 f(x)P\| + M\left(\gamma - \frac{\|\varepsilon\|}{2\|D\|}\right)\right) \tag{47}$$

*Proof.* Using propositions 4 to 6, setting $B = \tilde{H} - M\gamma P$ gives

$$\left\|M\left(\gamma + \frac{\|D\alpha\|}{2}\right)D\alpha + \nabla f(x_+)\right\|$$
$$\leq \frac{L}{2}\|D\alpha\|^2 + \|D\alpha\|\left(\frac{L}{2}\|D_\dagger\|\|\varepsilon\| + \|(I - P)\nabla^2 f(x)P\| + M\left\|D_\dagger^T D_\dagger\frac{\|D\|\|\varepsilon\|}{2} - I\gamma\right\|\right)$$

It remains to bound the last term,

$$\left\|D_\dagger^T D_\dagger\frac{\|D\|\|\varepsilon\|}{2} - P\gamma\right\| = \left\|D(D^T D)^{-\frac{1}{2}}\left((D^T D)^{-1}\frac{\|D\|\|\varepsilon\|}{2} - \gamma\right)(D^T D)^{-\frac{1}{2}}D^T\right\|.$$

Since the smallest and largest eigenvalue of $(D^T D)^{-1}$ are $\frac{1}{\sigma_{\max}^2(D)}, \frac{1}{\sigma_{\min}^2(D)}$ the norm can be explicitly bounded as follow:

$$\left\|D_\dagger^T D_\dagger\frac{\|D\|\|\varepsilon\|}{2} - P\gamma\right\| \leq \max\left\{\frac{\|D\|\|\varepsilon\|}{2\sigma_{\min}^2(D)} - \gamma \; ; \; \gamma - \frac{\|D\|\|\varepsilon\|}{2\sigma_{\max}^2(D)}\right\}$$

Setting $\gamma$ such that the maximum is attained at the right-hand-side, i.e.,

$$\gamma \geq \frac{\sigma_{\min}^{-2}(D) + \sigma_{\max}^{-2}(D)}{4}\|D\|\|\varepsilon\| = \frac{\kappa_D^2 + 1}{4}\frac{\|\varepsilon\|}{\|D\|},$$

simplifies the bound into

$$\left\|D_\dagger^T D_\dagger\frac{\|D\|\|\varepsilon\|}{2} - P\gamma\right\| \leq \gamma - \frac{\|\varepsilon\|}{2\|D\|}.$$

The last step consist in replacing $\|D_\dagger\|$ by $\frac{\kappa_D}{\|D\|}$. $\square$

 **F.2   Missing proofs from Section 2**

782  **Theorem 1.** *Let the function $f$ satisfy Assumption 1. Let $x_+$ be defined as in (4) and the matrices*
783  *$D$, $G$ be defined as in (3) and vector $\varepsilon$ as in (7). Then, for all $w \in \mathbb{R}^d$ and $\alpha \in \mathbb{R}^N$,*

$$-\tfrac{L\|w\|}{2} \sum_{i=1}^N |\alpha_i|\varepsilon_i \leq w^T(\nabla^2 f(x)D - G)\alpha \leq \tfrac{L\|w\|}{2} \sum_{i=1}^N |\alpha_i|\varepsilon_i, \tag{8}$$

$$\|w^T(\nabla^2 f(x)D - G)\| \leq \tfrac{L\|w\|}{2}\|\varepsilon\|. \tag{9}$$

784  *Proof.* Using Cauchy-Schwartz with (5) gives that, for all $v$,

$$v^T\left(\nabla f(y) - \nabla f(z) - \nabla^2 f(z)(y - z)\right) \leq \frac{L\|v\|}{2}\|y - z\|^2.$$

785  Let $v = v_i$, $y = y_i$, and $z = z_i$. By the definition of $Y$, $Z$, $D$, $G$ in (3),

$$v_i^T\left(r_i - \nabla^2 f(z_i)d_i\right) \leq \frac{L\|v_i\|}{2}\|d_i\|^2.$$

786  Introducing $\nabla^2 f(x)$ gives

$$v_i^T\left(r_i - \nabla^2 f(z_i)d_i\right) = v_i^T\left(r_i - \nabla^2 f(x)d_i\right) + v_i^T(\nabla^2 f(z_i) - \nabla^2 f(x))d_i.$$

787  Since the Hessian is $L$-Lipchitz-continuous Assumption 1, $(\nabla^2 f(z_i) - \nabla^2 f(x))d_i \leq L\|d_i\|\|z_i - x\|$.
788  Therefore, by the definition of $\varepsilon_i$,

$$v_i^T\left(r_i - \nabla^2 f(x)d_i\right) \leq \frac{L\|v_i\|\varepsilon_i}{2}. \tag{48}$$

789  Let $v_i = \text{sign}(\alpha_i)w$. Summing all inequalities multiplied by $|\alpha_i|$ gives the first desired result:

$$w^T\left(G - \nabla^2 f(x)D\right)\alpha \leq \frac{L\|w\|\sum_{i=1}^N \varepsilon_i|\alpha_i|}{2}.$$

790  The second result is rather straightforward, since (48) with $v_i = w$ gives

$$w^T\left(r_i - \nabla^2 f(x)d_i\right) \leq \frac{L\|w\|\varepsilon_i}{2}.$$

791  Therefore,

$$\sqrt{\sum_{i=1}^N \left(w^T\left(r_i - \nabla^2 f(x)d_i\right)\right)^2} \leq \|w\|\sqrt{\sum_{i=1}^N \left(r_i - \nabla^2 f(x)d_i\right)^2} \leq \|w\|\sqrt{\sum_{i=1}^N L\varepsilon_i^2} \leq \frac{L\|w\|\|\varepsilon\|}{2}.$$

792                                                                                                          $\square$

793  **Theorem 2.** *Let the function $f$ satisfy Assumption 1. Let $x_+$ be defined as in (4), the matrices $D$, $G$*
794  *be defined as in (3) and $\varepsilon$ be defined as in (7). Then, for all $\alpha \in \mathbb{R}^N$,*

$$f(x_+) \leq f(x) + \nabla f(x)^T D\alpha + \frac{\alpha^T H\alpha}{2} + \frac{L\|D\alpha\|^3}{6}, \quad H \stackrel{def}{=} \frac{G^T D + D^T G + IL\|D\|\|\varepsilon\|}{2} \tag{10}$$

$$\|\nabla f(x_+)\| \leq \|\nabla f(x) + G\alpha\| + \frac{L}{2}\left(\sum_{i=1}^N |\alpha_i|\varepsilon_i + \|D\alpha\|^2\right), \tag{11}$$

795  *Proof.* The inequality (11) is a direct consequence of (5) (with $y = x_+$, $z = x$) combined with (9),

$$\|\nabla f(x_+) - \nabla f(x) - \nabla^2 f(x)D\alpha\| \leq \frac{L}{2}\|D\alpha\|^2$$

$$\Leftrightarrow w^T\left(\nabla f(x_+) - \nabla f(x) - \nabla^2 f(x)D\alpha\right) \leq \frac{L\|w\|}{2}\|D\alpha\|^2$$

$$\Leftrightarrow w^T\nabla f(x_+) \leq \frac{L}{2}\|D\alpha\|^2 + w^T\left(\nabla f(x) + \nabla^2 f(x)D\alpha\right)$$

$$\Leftrightarrow w^T\nabla f(x_+) \stackrel{(8)}{\leq} \frac{L\|w\|}{2}\left(\|D\alpha\|^2 + \sum_{i=1}^N |\alpha_i|\varepsilon_i\right) + w^T\left(\nabla f(x) + G\alpha\right)$$

$$\Leftrightarrow w^T\nabla f(x_+) \leq \|w\|\left(\frac{L}{2}\left(\|D\alpha\|^2 + \sum_{i=1}^N |\alpha_i|\varepsilon_i\right) + \|\nabla f(x) + G\alpha\|\right)$$

796    Setting $w = \nabla f(x_+)$ gives (11).

797    The inequality (10) instead comes from (6) combined with (9). Indeed,

$$f(x_+) \leq f(x) + \nabla f(x)D\alpha + \frac{1}{2}(D\alpha)^T \nabla^2 f(x)(D\alpha) + \frac{L}{6}\|D\alpha\|^3$$

$$\overset{(9)}{\leq} f(x) + \nabla f(x)D\alpha + \frac{1}{2}\left((D\alpha)^T G\alpha + \frac{L\|D\alpha\|}{2}\sum_{i=1}^N |\alpha_i|\varepsilon_i\right) + \frac{L}{6}\|D\alpha\|^3$$

798    It remains to use the followings bounds:

$$\sum_{i=1}^N |\alpha_i|\varepsilon_i = \alpha^T(\text{sign}(\alpha) \odot \varepsilon) \leq \|\alpha\|\|\varepsilon\|,$$

$$\|D\alpha\| \leq \|D\|\|\alpha\|.$$

799    All together,

$$f(x_+) \leq f(x) + \nabla f(x)D\alpha + \frac{1}{2}(D\alpha)^T G\alpha + \frac{L}{4}\|\alpha\|^2\|D\|\|\varepsilon\| + \frac{L}{6}\|D\alpha\|^3$$

800    Finally, since $(D\alpha)^T G\alpha$ is a quadratic form, only the symmetric counterpart of $D^T G$ counts. That
801    means, writing $H = \frac{D^T G + G^T D}{2} + I\frac{L}{2}\|D\|\|\varepsilon\|$ gives the desired result,

$$f(x_+) \leq f(x) + \nabla f(x)D\alpha + \frac{\alpha^T H\alpha}{2} + \frac{L}{6}\|D\alpha\|^3.$$

802         $\square$

### F.3   Missing proofs from Section 3

803

804    **Theorem 3.** *Let $f$ satisfy Assumption 1. Then, at each iteration $t \geq 0$, algorithm 3 achieves*

$$f(x_{t+1}) \leq f(x_t) - \frac{M_{t+1}}{12}\|x_{t+1} - x_t\|^3, \quad M_{t+1} < \max\left\{2L \; ; \; \frac{M_0}{2^t}\right\}. \tag{13}$$

805    *Proof.* Using (35), at each iteration, after the while loop, the first-order condition of the subroutine
806    algorithm 1 reads

$$D_t^T \nabla f(x_t) + H_t \alpha_{t+1} + \frac{M_{t+1}}{2}D_t^T D_t \alpha_{t+1}\|D_t \alpha_{t+1}\| = 0. \tag{49}$$

807    The subscript $t$ is dropped for clarity. After multiplying by $\alpha$,

$$\nabla f(x_t)^T D\alpha + \alpha^T H\alpha + \frac{M}{2}\|D\alpha\|^3 = 0.$$

808    In addition, multiplying both times by $\alpha$ the second-order condition (36) gives

$$\alpha^T H\alpha \geq -\frac{M}{2}\|D\alpha\|^3.$$

809    which gives, after replacing it in (49),

$$\nabla f(x_t)^T D\alpha \leq -\frac{M}{2}\|D\alpha\|^3 + \frac{M}{2}\|D\alpha\|^3 = 0. \tag{50}$$

810    Injecting eqs. (49) and (50) into the while condition of algorithm 1 gives the desired result:

$$f(x_+) \leq f(x) + \nabla f(x)^T D\alpha + \frac{1}{2}\alpha^T H\alpha + \frac{M\|D\alpha\|^3}{6}, \tag{51}$$

$$= f(x) - \frac{1}{2}\nabla f(x)^T D\alpha - \frac{M\|D\alpha\|^3}{12}$$

$$\leq f(x) - \frac{M\|D\alpha\|^3}{12}.$$

811    Where (51) is guaranteed if $M > L$. Therefore, in the worst case, $M < 2L$.      $\square$

**Theorem 4.** *Let $f$ satisfy Assumption 1, and assume that $f$ is bounded below by $f^*$. Let Requirements 1b to 3 hold, and $M_t \geq M_{\min}$. Then, algorithm 3 starting at $x_0$ with $M_0$ achieves*

$$\min_{i=1,\ldots,t} \|\nabla f(x_i)\| \leq \max \left\{ \frac{3L}{t^{2/3}} \left( 12 \frac{f(x_0) - f^\star}{M_{\min}} \right)^{2/3} \; ; \; \left( \frac{C_1}{t^{1/3}} \right) \left( 12 \frac{f(x_0) - f^\star}{M_{\min}} \right)^{1/3} \right\},$$

*where* $C_1 = \delta L \left( \frac{\kappa + 2\kappa^2}{2} \right) + \max_{i \in [0,t]} \|(I - P_i) \nabla^2 f(x_i) P_i\|.$

*Proof.* The starting inequality is (41):

$$\|\nabla f(x_+)\| \leq \frac{L + M}{2} \|D\alpha\|^2 + \|D\alpha\| \left( \frac{\|\varepsilon\|}{\|D\|} \left( \frac{L + M\kappa_D}{2} \right) \kappa_D + \|(I - P) \nabla^2 f(x) P\| \right).$$

The result is obtained by decomposing the inequality using a maximum,

$$\|\nabla f(x_+)\|$$

$$\leq \max \left\{ (L + M) \|D\alpha\|^2 \; ; \; 2\|D\alpha\| \left( \frac{\|\varepsilon\|}{\|D\|} \left( \frac{L + M\kappa_D}{2} \right) \kappa_D + \|(I - P) \nabla^2 f(x) P\| \right) \right\}.$$

In the first case,

$$\|D\alpha\| \geq \sqrt{\frac{\|\nabla f(x_+)\|}{L + M}}, \tag{52}$$

while in the second case,

$$\|D\alpha\| \geq \frac{\|\nabla f(x_+)\|}{\frac{\|\varepsilon\|}{\|D\|} \left( \frac{L + M\kappa_D}{2} \right) \kappa_D + \|(I - P) \nabla^2 f(x) P\|}.$$

Let $C_t$ be defined as

$$C_t = \frac{\|\varepsilon_t\|}{\|D_t\|} \left( \frac{L + M_{t+1}\kappa_{D_t}}{2} \right) \kappa_{D_t} + \|(I - P_t) \nabla^2 f(x_t) P_t\|.$$

Then, using Requirements 2 and 3, and since $M < 2L$ by Theorem 3,

$$C_t \leq C = \delta L \left( \frac{1 + 2\kappa}{2} \right) \kappa + \max_t \|(I - P_t) \nabla^2 f(x_t) P_t\|$$

Therefore,

$$\|D\alpha\| \geq \frac{\|\nabla f(x_+)\|}{C}. \tag{53}$$

At each iteration $t$, combining eqs. (52) and (53) into Theorem 3 gives

$$f(x_t) - f(x_{t+1}) \geq \frac{M_{t+1}}{12} \|\underbrace{x_{t+1} - x_t}_{=D_t\alpha_t}\|^3 \geq \frac{M_{t+1}}{12} \min \left\{ \left( \frac{\|\nabla f(x_+)\|}{L + M_{t+1}} \right)^{3/2} \; ; \; \left( \frac{\|\nabla f(x_+)\|}{C} \right)^3 \right\}$$

Therefore,

$$\begin{aligned}
f(x_0) - f^\star &\geq f(x_0) - f(x_t) \\
&= \sum_{i=0}^{t-1} f(x_i) - f(x_{i+1}) \\
&\geq \sum_{i=0}^{t-1} \left( \frac{M_{i+1}}{12} \|x_{i+1} - x_i\|^3 \right) \\
&\geq \sum_{i=0}^{t-1} \min_t \frac{M_{i+1}}{12} \left\{ \left( \frac{\|\nabla f(x_{i+1})\|}{L + M_{i+1}} \right)^{3/2} \; ; \; \left( \frac{\|\nabla f(x_{i+1})\|}{C} \right)^3 \right\} \\
&\geq t \min_{i \in [0,t-1]} \frac{M_{i+1}}{12} \min \left\{ \left( \frac{\|\nabla f(x_{i+1})\|}{L + M_{i+1}} \right)^{3/2} \; ; \; \left( \frac{\|\nabla f(x_{i+1})\|}{C} \right)^3 \right\} \\
&\geq t \frac{M_{\min}}{12} \min \left\{ \min_{i \in [1,t]} \left( \frac{\|\nabla f(x_i)\|}{3L} \right)^{3/2} \; ; \; \min_{i \in [1,t]} \left( \frac{\|\nabla f(x_i)\|}{C} \right)^3 \right\}
\end{aligned}$$

823    After analyzing separately each case of the minimum, either

$$\left(\frac{\min_{i\in[1,t]}\|\nabla f(x_i)\|}{3L}\right)^{3/2} \le 12\frac{f(x_0)-f^\star}{tM_{\min}} \quad \text{or} \quad \left(\frac{\min_{i\in[1,t]}\|\nabla f(x_{t+1})\|}{C}\right)^{3} \le 12\frac{f(x_0)-f^\star}{tM_{\min}}.$$

824    It remains to simplify to obtain the desired result,

$$\min_{i=1\ldots t}\|\nabla f(x_i)\| \le \max\left\{\frac{3L}{t^{2/3}}\left(12\frac{f(x_0)-f^\star}{M_{\min}}\right)^{2/3} ; \left(\frac{C}{t^{1/3}}\right)\left(12\frac{f(x_0)-f^\star}{M_{\min}}\right)^{1/3}\right\}.$$

825    $\qquad\qquad\qquad\qquad\qquad\qquad\qquad\qquad\qquad\qquad\qquad\qquad\qquad\qquad\qquad$ □

826    **Theorem 5.** *Assume $f$ satisfy Assumptions 1 to 3. Let Requirements 1b to 3 hold. Then, algorithm 3*
827    *starting at $x_0$ with $M_0$ achieves, for $t \ge 1$,*

$$(f(x_t)-f^\star) \le 6\frac{f(x_t)-f^\star}{t(t+1)(t+2)} + \frac{1}{(t+1)(t+2)}\frac{L(3R)^3}{2} + \frac{1}{t+2}\frac{C_2(3R)^2}{4},$$

*where* $\quad C_2 \stackrel{def}{=} \delta L\frac{\kappa+2\kappa^2}{2} + \max_{i\in[0,t]}\|\nabla^2 f(x_i) - P_i\nabla^2 f(x_i)P_i\|.$

828    *Proof.* Starting from the inequality in proposition 9,

$$f(x_{t+1}) \le f(y) + \frac{M_{t+1}+L}{6}\|y-x_t\|^3 + \frac{\|y-x_t\|^2}{2}C_2^{(t)},$$

829    where

$$C_2^{(t)} = \|\nabla^2 f(x_t) - P_t\nabla^2 f(x_t)P_t\| + \delta\frac{L\kappa+M_{t+1}\kappa^2}{2},$$

830    and setting $y = (1-\beta_t)x_t + \beta_t x^\star$ and $f(x^\star) = f^\star$ gives

$$f(x_{t+1}) - f^\star \le f((1-\beta_t)x_t + \beta_t x^\star) - f^\star + \frac{M_{t+1}+L}{6}\beta_t^3\|x_t-x^\star\|^3 + \frac{\beta_t^2\|x_t-x^\star\|^2}{2}C_2^{(t)}.$$

831    Because the function is star-convex,

$$f(x_{t+1}) - f^\star \le (1-\beta_t)(f(x_t)-f^\star) + \frac{M_{t+1}+L}{6}\beta_t^3\|x_t-x^\star\|^3 + \frac{\beta_t^2\|x_t-x^\star\|^2}{2}C_2^{(t)}.$$

832    Since algorithm 1 ensure a decrease in the function value, the iterate $x_t$ satisfies

$$x_t \in \{x : f(x \le f(x_0))\},$$

833    and therefore, $\|x_t - x^\star\| \le R$ by Assumption 2. In addition, $M < 2L$ by Theorem 3. The inequality
834    now becomes

$$(f(x_{t+1}) - f^\star) \le (1-\beta_t)(f(x_t)-f^\star) + \beta_t^3\frac{LR^3}{2} + \beta_t^2\frac{R^2C_2^{(t)}}{2}. \qquad (54)$$

835    Finally, since $M < 2L$, the scalar $C_2^t$ is bounded over time by $C_2$:

$$C_2^{(t)} \le C_2 \stackrel{def}{=} \delta L\frac{\kappa+2\kappa^2}{2} + \max_t\|\nabla^2 f(x_t) - P_t\nabla^2 f(x_t)P_t\|.$$

836    Now, let

837      • $B_t = \frac{t(t+1)(t+2)}{6}$,

838      • $b_t : B_t = B_{t-1} + b_t$, hence $b_t = \frac{t(t+1)}{2}$, and

839      • $\beta_t = \frac{b_{t+1}}{B_{t+1}}$.

840 Therefore, for $t \geq 1$,

$$1 = \frac{B_t}{B_t} = \frac{B_{t-1}}{B_t} + \frac{b_t}{B_t} = \frac{B_{t-1}}{B_t} + \beta_{t-1} \quad \Rightarrow \quad 1 - \beta_{t-1} = \frac{B_{t-1}}{B_t}.$$

841 Injecting those relations in (54) gives

$$(f(x_{t+1}) - f^\star) \leq \frac{B_t}{B_{t+1}}(f(x_t) - f^\star) + \left(\frac{b_{t+1}}{B_{t+1}}\right)^3 \frac{LR^3}{2} + \left(\frac{b_{t+1}}{B_{t+1}}\right)^2 \frac{R^2 C_2}{2},$$

842 hence the recursion

$$B_{t+1}(f(x_{t+1}) - f^\star) \leq B_t(f(x_t) - f^\star) + \frac{b_{t+1}^3}{B_{t+1}^2}\frac{LR^3}{2} + \frac{b_{t+1}^2}{B_{t+1}}\frac{R^2 C_2}{2}$$

$$\leq B_0(f(x_t) - f^\star) + \sum_{i=0}^{t} \frac{b_{i+1}^3}{B_{i+1}^2}\frac{LR^3}{2} + \sum_{i=0}^{t} \frac{b_{i+1}^2}{B_{i+1}}\frac{R^2 C_2}{2}.$$

843

$$(f(x_{t+1}) - f^\star) \leq \frac{B_0}{B_{t+1}}(f(x_t) - f^\star) + \frac{\sum_{i=0}^{t} \frac{b_{i+1}^3}{B_{i+1}^2}}{B_{t+1}}\frac{LR^3}{2} + \frac{\sum_{i=0}^{t} \frac{b_{i+1}^2}{B_{i+1}}}{B_{t+1}}\frac{R^2 C_2}{2}.$$

844 Therefore, the rate reads By the definition of $b_t$ and $B_t$,

$$\frac{b_{i+1}^3}{B_{i+1}^2} = \frac{36}{8}\frac{(i+1)^3(i+2)^3}{(i+1)^2(i+2)^2(i+3)^2} = \frac{9}{2}\frac{(i+1)(i+2)}{(i+3)^2} \leq \frac{9}{2},$$

$$\frac{b_{i+1}^2}{B_{i+1}} = \frac{6}{4}\frac{(i+1)^2(i+2)^2}{(i+1)(i+2)(i+3)} = \frac{3}{2}\frac{(i+2)}{(i+3)}(i+1) \leq \frac{3}{2}(i+1).$$

845 Hence,

$$\frac{\sum_{i=0}^{t}\frac{b_{i+1}^3}{B_{i+1}^2}}{B_{t+1}} \leq \frac{\frac{9}{2}(t+1)}{\frac{(t+1)(t+2)(t+3)}{6}} \leq \frac{27}{(t+2)(t+3)},$$

$$\frac{\sum_{i=0}^{t}\frac{b_{i+1}^2}{B_{i+1}}}{B_{t+1}} \leq \frac{\sum_{i=0}^{t}\frac{3}{2}(i+1)}{\frac{(t+1)(t+2)(t+3)}{6}} = \frac{\frac{3}{4}(t+2)(t+1)}{\frac{(t+1)(t+2)(t+3)}{6}} = \frac{9}{2(t+3)}.$$

846 Shifting from $t+1$ tp $t$ gives the desired result,

$$(f(x_t) - f^\star) \leq 6\frac{f(x_t) - f^\star}{t(t+1)(t+2)} + \frac{1}{(t+1)(t+2)}\frac{L(3R)^3}{2} + \frac{1}{t+2}\frac{C_2(3R)^2}{4}.$$

847 $\qquad\qquad\qquad\qquad\qquad\qquad\qquad\qquad\qquad\qquad\qquad\qquad\qquad\qquad\qquad\qquad\qquad\quad\square$

848 **Theorem 6.** *Assume $f$ satisfy Assumptions 1, 2 and 4. Let Requirements 1a, 2 and 3 hold. Then, in*
849 *expectation over the matrices $D_i$, algorithm 3 starting at $x_0$ with $M_0$ achieves, for $t \geq 1$,*

$$\mathbb{E}_{D_t}[f(x_t) - f^\star] \leq \frac{1}{1 + \frac{1}{4}\left[\frac{N}{d}t\right]^3}(f(x_0) - f^\star) + \frac{1}{\left[\frac{N}{d}t\right]^2}\frac{L(3R)^3}{2} + \frac{1}{\left[\frac{N}{d}t\right]}\frac{C_3(3R)^2}{2},$$

$$\text{where} \quad C_3 \stackrel{def}{=} \delta L\frac{\kappa + 2\kappa^2}{2} + \frac{(d-N)}{d}\max_{i \in [0,t]}\|\nabla^2 f(x_i)\|.$$

850 *Proof.* The proof technique is similar to [35]. Starting from proposition 10 with $x = x_t$,

$$\mathbb{E}f(x_{t+1}) \leq \left(1 - \frac{N}{d}\right)f(x_t) + \frac{N}{d}f(y) + \frac{N}{d}\frac{(M_{t+1} + L)}{6}\|y - x_t\|^3$$

$$+ \frac{N}{d}\frac{\|y - x_t\|^2}{2}\left(\delta\frac{L\kappa + M_{t+1}\kappa^2}{2} + \frac{(d-N)}{d}\|\nabla^2 f(x_t)\|\right),$$

851 where the expectation is taken with $D_0, \ldots, D_{t-1}$ fixed. Using the inequality $M_{t+1} \leq 2L$ gives

$$\mathbb{E}f(x_{t+1}) \leq \left(1 - \frac{N}{d}\right)f(x_t) + \frac{N}{d}\left(f(y) + \frac{\|y - x_t\|^2}{2}C_3 + \frac{L}{2}\|y - x_t\|^3\right)$$

852 where
$$C_3 \overset{\text{def}}{=} \left( \delta L \frac{\kappa + 2\kappa^2}{2} + \frac{(d-N)}{d} \max_{i \in [0,t]} \|\nabla^2 f(x_i)\| \right).$$

853 Let $y = \beta_t x^\star + (1 - \beta_t)x_t$, $\beta_t \in [0,1]$. After using Assumption 4 and Assumption 2,

$$\mathbb{E}f(x_{t+1}) \leq \left(1 - \frac{N}{d}\right) f(x_t) + \frac{N}{d} \left( f\left(\beta_t x^\star + (1-\beta_t)x_t\right) + \beta_t^2 \frac{C_3 R^2}{2} + \beta_t^3 \frac{LR^3}{2} \right)$$

$$\leq \left(1 - \frac{N}{d}\right) f(x_t) + \frac{N}{d} \left( \beta_t f(x^\star) + (1-\beta_t)f(x_t) + \beta_t^2 \frac{C_3 R^2}{2} + \beta_t^3 \frac{LR^3}{2} \right)$$

$$= \left(1 - \frac{N}{d}\right) f(x_t) + \frac{N}{d} \left( \beta_t f(x^\star) + (1-\beta_t)f(x_t) + \beta_t^2 \frac{C_3 R^2}{2} + \beta_t^3 \frac{LR^3}{2} \right),$$

$$= \left(1 - \beta_t \frac{N}{d}\right) f(x_t) + \frac{N}{d} \left( \beta_t f(x^\star) + \beta_t^2 \frac{C_3 R^2}{2} + \beta_t^3 \frac{LR^3}{2} \right).$$

854 Hence, the recursion

$$(\mathbb{E}f(x_{t+1}) - f^\star) \leq \left(1 - \beta_t \frac{N}{d}\right)(f(x_t) - f^\star) + \frac{N}{d} \left( \beta_t^2 \frac{C_3 R^2}{2} + \beta_t^3 \frac{LR^3}{2} \right).$$

855 Now, define

$$b_t = t^2,$$

$$B_t = B_0 + \sum_{i=0}^{t} b_i, \quad B_0 = \frac{4}{3} \left(\frac{d}{N}\right)^3$$

$$\beta_t = \frac{d}{N} \frac{b_{t+1}}{B_{t+1}} \quad \Rightarrow \quad 1 - \frac{N}{d}\beta_t = \frac{B_t}{B_{t+1}}.$$

856 Replacing those relations in the recursion gives

$$B_{t+1}\left(\mathbb{E}f(x_{t+1}) - f^\star\right)$$

$$\leq B_t(f(x_t) - f^\star) + \frac{N}{dB_{t+1}} \left( \left(\frac{d}{N}\frac{b_{t+1}}{B_{t+1}}\right)^2 \frac{C_3 R^2}{2} + \left(\frac{d}{N}\frac{b_{t+1}}{B_{t+1}}\right)^3 \frac{LR^3}{2} \right)$$

$$= B_t(f(x_t) - f^\star) + \frac{d}{N}\frac{b_{t+1}^2}{B_{t+1}}\frac{C_3 R^2}{2} + \frac{d^2}{N^2}\frac{b_{t+1}^3}{B_{t+1}^2}\frac{LR^3}{2}$$

857 Expanding the inequality gives

$$B_{t+1}\left(\mathbb{E}f(x_{t+1}) - f^\star\right) \leq B_0(f(x_0) - f^\star) + \frac{d}{N}\sum_{t=0}^{t+1}\frac{b_{i+1}^2}{B_{i+1}}\frac{C_3 R^2}{2} + \frac{d^2}{N^2}\sum_{t=0}^{t+1}\frac{b_{i+1}^3}{B_{i+1}^2}\frac{LR^3}{2}$$

858 Since

$$B_t = B_0 + \sum_{i=1}^{t} \geq B_0 + \int_0^t x^2 \, \mathrm{d}x = B_0 + \frac{t^3}{3}$$

$$\sum_{i=0}^{t}\frac{b_t^2}{B_t} \leq \sum_{i=0}^{t}\frac{i^4}{B_0 + i^3/3} \leq 3t^2,$$

$$\sum_{i=0}^{t}\frac{b_t^3}{B_t^2} \leq \sum_{i=0}^{t}\frac{i^6}{(B_0 + i^3/3)^2} \leq 9t,$$

859 the bound becomes

$$B_{t+1}\left(\mathbb{E}f(x_{t+1}) - f^\star\right) \leq B_0(f(x_0) - f^\star) + \frac{d}{N}3t^2\frac{C_3 R^2}{2} + \frac{d^2}{N^2}9t\frac{LR^3}{2}$$

Dividing both sides by $B_{t+1}$ gives

$$\mathbb{E}f(x_{t+1}) - f^\star \leq \frac{B_0}{B_0 + \frac{(t+1)^3}{3}}(f(x_0) - f^\star) + \frac{d}{N}\frac{3(t+1)^2}{B_0 + \frac{(t+1)^3}{3}}\frac{C_3R^2}{2} + \frac{d^2}{N^2}\frac{9(t+1)}{B_0 + \frac{(t+1)^3}{3}}\frac{LR^3}{2}.$$

After the following simplifications,

$$\frac{B_0}{B_0 + (t+1)^3/3} = \frac{1}{1 + \frac{(t+1)^3}{3B_0}} = \frac{1}{1 + \frac{1}{4}\left(\frac{N}{d}(t+1)\right)^3},$$

$$\frac{3(t+1)^2}{B_0 + (t+1)^3/3} = \frac{3}{B_0}\frac{(t+1)^3}{1 + \frac{(t+1)^3}{3B_0}}\frac{1}{t+1} \leq \frac{3}{B_0}3B_0\frac{1}{t+1} = \frac{9}{t+1},$$

$$\frac{9(t+1)}{B_0 + \frac{(t+1)^3}{3}} = \frac{9}{B_0}\frac{(t+1)^3}{\frac{(t+1)^3}{3B_0}}\frac{1}{(t+1)^2} \leq \frac{9}{B_0}3B_0\frac{1}{(t+1)^2} = \frac{27}{(t+1)^2},$$

the inequality finally becomes (after shifting from $t+1$ to $t$),

$$\mathbb{E}f(x_t) - f^\star \leq \frac{1}{1 + \frac{1}{4}\left[\frac{N}{d}t\right]^3}(f(x_0) - f^\star) + \frac{1}{\left[\frac{N}{d}t\right]^2}\frac{L(3R)^3}{2} + \frac{1}{\left[\frac{N}{d}t\right]}\frac{C_3(3R)^2}{2}.$$

□

## F.4  Missing proofs from Section 4

**Notations**  The following functions define the estimate sequence,

$$\ell_t(x) = \sum_{i=2}^{t} b_{i-1}\left(f(x_i) + \nabla f(x_i)(x - x_i)\right), \tag{55}$$

$$\phi_t(x) = f(x_1) + \ell_t(x) + \frac{\lambda_t^{(1)}}{2}\|x - x_0\|^2 + \frac{\lambda_t^{(2)}}{6}\|x - x_0\|^3 \tag{56}$$

$$\Phi_t(x) = \frac{\phi_t(x)}{B_t}, \tag{57}$$

where $\lambda_t^{(1,2)}$ are non-negative and increasing, and the sequences $b_t$, $B_t$ are

$$B_t = \frac{k(t+1)(t+2)}{6} = \sum_{i=1}^{t} b_i, \tag{58}$$

$$b_t = \frac{(t+1)(t+2)}{2} = B_{t+1} - B_t. \tag{59}$$

$$\tag{60}$$

Moreover, the following quantities will be important later,

$$v_t = \arg\min_x \phi_t(x) = \arg\min_x \Phi_t(x), \tag{61}$$

$$\beta_t = \frac{b_t}{B_{t+1}}, \tag{62}$$

$$y_t = (1 - \beta_t)x_t + \beta_t v_t. \tag{63}$$

### F.4.1  Technical results

**Lemma 1.** *From [44, Lemma 4]. The Bregman divergence of the function $\|x\|^i$ satisfies, for $i \geq 2$,*

$$\|x\|^i - \|y\|^i - \nabla(\|y\|^i)(x - y) \geq \frac{1}{2^{i-2}}\|x - y\|^i.$$

**Proposition 12.** *The function $\phi_t$ is lower-bounded by*

$$\phi_t \geq \underbrace{\phi_t(v_t)}_{=\phi_t^\star} + \frac{\lambda_t^{(1)}}{2}\|x - v_t\|^2 + \frac{\lambda_t^{(2)}}{12}\|x - v_t\|^3 \tag{64}$$

*where $v_t = \arg\min_x \phi_t(x)$.*

*Proof.* The first order condition on $\phi_t$ reads,

$$\ell_t' + \nabla \left( \frac{\lambda_t^{(1)}}{2} \|v_t - x_0\|^2 + \frac{\lambda_t^{(2)}}{6} \|v_t - x_0\|^3 \right) = 0.$$

Multiplying both sides by $(x - v_t)$ gives

$$\ell_t'(x - v_t) + \nabla \left( \frac{\lambda_t^{(1)}}{2} \|v_t - x_0\|^2 + \frac{\lambda_t^{(2)}}{6} \|v_t - x_0\|^3 \right)(x - v_t) = 0.$$

Note that, since $\ell_t$ is an affine function, $\ell_t'(x - v_t) = \ell_t(x) - \ell_t(v_t)$. Hence,

$$\ell_t(x) - \ell_t(v_t) + \nabla \left( \frac{\lambda_t^{(1)}}{2} \|v_t - x_0\|^2 + \frac{\lambda_t^{(2)}}{6} \|v_t - x_0\|^3 \right)(x - v_t) = 0.$$

Finally, adding $\frac{\lambda_t^{(1)}}{2} \|x - x_0\|^2 + \frac{\lambda_t^{(2)}}{6} \|x - x_0\|^3$ on both sides and after reorganizing the terms,

$$\phi_t(x) = \ell_t(v_t) + \frac{\lambda_t^{(1)}}{2} \|x - x_0\|^2 + \frac{\lambda_t^{(2)}}{6} \|x - x_0\|^3 - \nabla \left( \frac{\lambda_t^{(1)}}{2} \|v_t - x_0\|^2 + \frac{\lambda_t^{(2)}}{6} \|v_t - x_0\|^3 \right)(x - v_t). \tag{65}$$

From lemma 1 with $x = x - x_0$, $y = v_t - x_0$, and after reorganizing the terms,

$$\|x - x_0\|^i - \nabla(\|v_t - x_0\|^i)(x - v_t) \geq \frac{1}{2^{i-2}} \|x - v_t\|^i + \|v_t - x_0\|^i.$$

Therefore, using the previous inequality with $i = 2$ and $i = 3$, (65) becomes

$$\phi_t(x) \geq \ell_t(v_t) + \frac{\lambda_t^{(1)}}{2} \|v_t - x_0\|^2 + \frac{\lambda_t^{(2)}}{6} \|v_t - x_0\|^3 + \frac{\lambda_t^{(2)}}{2} \|v_t - x\|^2 + \frac{\lambda_t^{(3)}}{12} \|v_t - x\|^3$$

By definition of $\phi_t^\star = \phi_t(v_t)$,

$$\phi_t(x) \geq \phi_t^\star + \frac{\lambda_t^{(1)}}{2} \|v_t - x\|^2 + \frac{\lambda_t^{(2)}}{12} \|v_t - x\|^3.$$

$\square$

**Proposition 13.** *Under the assumptions of proposition 11 the condition*

$$\frac{\|f(x_+)\|^2}{M \left( \gamma + \frac{\|D\alpha\|}{2} \right)} \leq -\nabla f(x)^T D\alpha$$

*is guaranteed as long as $\gamma$ and $M$ are sufficiently big,*

$$\gamma \geq \frac{1}{2} \frac{\|\varepsilon\|}{\|D\|} \frac{1 + \kappa_D^2}{2},$$

$$M \geq \frac{1}{\frac{\|D\alpha\|}{2} + \frac{\|\varepsilon\|}{2\|D\|}} \left( \frac{L}{2} \left( \|D\alpha\| + \frac{\|\varepsilon\|}{\|D\|} \kappa_D \right) + \|(I - P)\nabla^2 f(x)P\| \right).$$

*Proof.* Elevating to the square the inequality of proposition 11 gives

$$\left( M \left( \gamma + \frac{\|D\alpha\|}{2} \right) \right)^2 \|D\alpha\|^2 + \|\nabla f(x_+)\|^2 + \left( M \left( \gamma + \frac{\|D\alpha\|}{2} \right) \right) \nabla f(x_+)^T D\alpha$$

$$\leq \|D\alpha\|^2 \left( \frac{L}{2} \|D\alpha\| + \frac{L}{2} \frac{\|\varepsilon\|}{\|D\|} \kappa_D + \|(I - P)\nabla^2 f(x)P\| + M \left( \gamma - \frac{\|\varepsilon\|}{2\|D\|} \right) \right)^2.$$

The desired result holds if the following condition is satisfied,

$$\left( M \left( \gamma + \frac{\|D\alpha\|}{2} \right) \right)^2 \|D\alpha\|^2$$

$$\geq \|D\alpha\|^2 \left( \frac{L}{2} \|D\alpha\| + \frac{L}{2} \frac{\|\varepsilon\|}{\|D\|} \kappa_D + \|(I - P)\nabla^2 f(x)P\| + M \left( \gamma - \frac{\|\varepsilon\|}{2\|D\|} \right) \right)^2.$$

After simplification of the squares and $\gamma$,

$$M\frac{\|D\alpha\|}{2} \geq \frac{L}{2}\|D\alpha\| + \frac{L}{2}\frac{\|\varepsilon\|}{\|D\|}\kappa_D + \|(I-P)\nabla^2 f(x)P\| - M\frac{\|\varepsilon\|}{2\|D\|}.$$

Hence, the following condition is sufficient,

$$M \geq \frac{1}{\frac{\|D\alpha\|}{2} + \frac{\|\varepsilon\|}{2\|D\|}}\left(\frac{L}{2}\left(\|D\alpha\| + \frac{\|\varepsilon\|}{\|D\|}\kappa_D\right) + \|(I-P)\nabla^2 f(x)P\|\right).$$

$\square$

**Proposition 14** (Guarantees that algorithm 4 terminates)**.** *Let $f$ satisfies Assumption 1. Then, under Requirements 1b to 3, if $M_0 < M$, the output of algorithm 4 guarantees that*

$$M \leq 2\frac{1}{\frac{\|D\alpha\|}{2} + \frac{\|\varepsilon\|}{2\|D\|}}\left(\frac{L}{2}\left(\|D\alpha\| + \frac{\|\varepsilon\|}{\|D\|}\kappa_D\right) + \|(I-P)\nabla^2 f(x)P\|\right)$$

$$\leq L\kappa_D + \frac{2\|(I-P)\nabla^2 f(x)P\|}{\frac{\|D\alpha\|}{2} + \frac{\|\varepsilon\|}{2\|D\|}}$$

$$M\left(\gamma + \frac{\|D\alpha\|}{2}\right) \leq (1+\kappa_D^2)\left(\frac{L}{2}\left(\|D\alpha\| + \frac{\|\varepsilon\|}{\|D\|}\kappa_D\right) + \|(I-P)\nabla^2 f(x)P\|\right).$$

*Proof.* Assume $\Delta = \infty$, so that the algorithm can only terminates with ExitFlag equals to SmallStep. Either the algorithm terminates at $M = M_0$, or $M_0 < M$. In the second case, algorithm 4 multiplies $M$ by a factor 2 while

$$\frac{\|f(x_+)\|^2}{M\left(\gamma + \frac{\|D\alpha\|}{2}\right)} \geq -\nabla f(x)^T D\alpha,$$

then, by proposition 13, once

$$M \geq \frac{1}{\frac{\|D\alpha\|}{2} + \frac{\|\varepsilon\|}{2\|D\|}}\left(\frac{L}{2}\left(\|D\alpha\| + \frac{\|\varepsilon\|}{\|D\|}\kappa_D\right) + \|(I-P)\nabla^2 f(x)P\|\right)$$

holds, the condition is met and the algorithm terminates. In the worst case, $M$ is at most two times larger than the bound:

$$M \leq 2\frac{1}{\frac{\|D\alpha\|}{2} + \frac{\|\varepsilon\|}{2\|D\|}}\left(\frac{L}{2}\left(\|D\alpha\| + \frac{\|\varepsilon\|}{\|D\|}\kappa_D\right) + \|(I-P)\nabla^2 f(x)P\|\right) \qquad (66)$$

Since, for all $c_1 > 0$, $c_2 > 0$, $c_3 > 1$,

$$\frac{c_1 + c_2 c_3}{c_1 + c_2} = 1 + \frac{c_2(c_3 - 1)}{c_1 + c_2} = 1 + \frac{c_3 - 1}{\frac{c_1}{c_2} + 1} \leq c_3, \qquad (67)$$

the bound becomes

$$M \leq \left(L\kappa_D + \frac{2\|(I-P)\nabla^2 f(x)P\|}{\frac{\|D\alpha\|}{2} + \frac{\|\varepsilon\|}{2\|D\|}}\right).$$

Going back to (66), and after multiplying both sides by $\gamma + \frac{\|D\alpha\|}{2}$ with $\gamma = \frac{1}{2}\frac{\|\varepsilon\|}{\|D\|}\frac{1+\kappa_D^2}{2}$ gives

$$M\left(\gamma + \frac{\|D\alpha\|}{2}\right) \leq 2\frac{\frac{\|D\alpha\|}{2} + \frac{1}{2}\frac{\|\varepsilon\|}{\|D\|}\frac{1+\kappa_D^2}{2}}{\frac{\|D\alpha\|}{2} + \frac{\|\varepsilon\|}{2\|D\|}}\left(\frac{L}{2}\left(\|D\alpha\| + \frac{\|\varepsilon\|}{\|D\|}\kappa_D\right) + \|(I-P)\nabla^2 f(x)P\|\right).$$

Once again, using (67) gives

$$M\left(\gamma + \frac{\|D\alpha\|}{2}\right) \leq (1+\kappa_D^2)\left(\frac{L}{2}\left(\|D\alpha\| + \frac{\|\varepsilon\|}{\|D\|}\kappa_D\right) + \|(I-P)\nabla^2 f(x)P\|\right).$$

$\square$

**Proposition 15.** *Assume $f$ satisfies Assumption 1. Then, under Requirements 1b to 3, if*

$$\|D\alpha\| \geq 2\left(\delta\kappa_D^2 + 2\frac{\|(I-P)\nabla^2 f(x)P\|}{\frac{1}{\sqrt{3}-1}L}\right).$$

*then the condition*

$$\frac{2}{3^{3/4}}\frac{\|\nabla f(x_+)\|^{3/2}}{\sqrt{M}} \leq -\nabla f(x_+)^T D\alpha$$

*is guaranteed as long as $M$ is sufficiently big,*

$$\frac{2}{\sqrt{3}-1}L \leq M.$$

*Proof.* Starting from proposition 7,

$$\left\|\frac{M\|D\alpha\|}{2}D\alpha + \nabla f(x_+)\right\|$$
$$\leq \frac{L}{2}\|D\alpha\|^2 + \|D\alpha\|\left(\frac{\|\varepsilon\|}{\|D\|}\left(\frac{L+M\kappa_D}{2}\right)\kappa_D + \|(I-P)\nabla^2 f(x)P\|\right).$$

Assuming

$$\frac{\frac{L}{\kappa_D}+M}{4}\|D\alpha\| \geq \frac{\|\varepsilon\|}{\|D\|}\left(\frac{L+M\kappa_D}{2}\right)\kappa_D + \|(I-P)\nabla^2 f(x)P\|,$$

or equivalently,

$$\|D\alpha\| \geq 4\left(\frac{\|\varepsilon\|}{2\|D\|}\kappa_D^2 + \frac{\|(I-P)\nabla^2 f(x)P\|}{\frac{L}{\kappa_D}+M}\right), \tag{68}$$

gives the simpler bound

$$\left\|\frac{M\|D\alpha\|}{2}D\alpha + \nabla f(x_+)\right\| \leq \frac{L+\frac{L}{\kappa_D}+M}{4}\|D\alpha\|^2.$$

Elevating both sides to the square give

$$\frac{M^2}{4}\|D\alpha\|^4 + M\|D\alpha\|D\alpha^T\nabla f(x_+) + \|\nabla f(x_+)\|^2 \leq \frac{(L(1+\frac{1}{\kappa_D})+M)^2}{16}\|D\alpha\|^4,$$

hence, and using the fact that $\kappa_D \geq 1$,

$$M\|D\alpha\|D\alpha^T\nabla f(x_+) + \|\nabla f(x_+)\|^2 \leq \frac{(2L+M)^2 - 4M^2}{16}\|D\alpha\|^4.$$

Assuming $\frac{2}{\sqrt{3}-1}L \leq M$,

$$M\|D\alpha\|D\alpha^T\nabla f(x_+) + \|\nabla f(x_+)\|^2 \leq \frac{-M^2}{16}\|D\alpha\|^4.$$

After reorganization, and writing $r = \|D\alpha\|$,

$$\frac{M}{16}r^3 + \frac{\|\nabla f(x_+)\|^2}{Mr} \leq -D\alpha^T\nabla f(x_+).$$

Using

$$\frac{c_1}{r} + c_2 r^3 \geq 4c_2^{1/4}\left(\frac{c_1}{3}\right)^{3/4},$$

the inequality becomes

$$-D\alpha^T\nabla f(x_+) \geq \frac{M^{1/4}}{2}\frac{\|\nabla f(x_+)\|^{3/2}}{M^{3/4}}\frac{4}{3^{3/4}}$$
$$= \frac{2}{3^{3/4}}\frac{\|\nabla f(x_+)\|^{3/2}}{\sqrt{M}}.$$

Finally, the condition on $\|D\alpha\|$ in (68) is made stronger by replacing $M$ with its lower bound, by using Requirements 1b to 3 and because $\kappa_D \geq 1$, i.e.,

$$\|D\alpha\| \geq 4\left(\frac{\delta\kappa_D^2}{2} + \frac{\|(I-P)\nabla^2 f(x)P\|}{\frac{2}{\sqrt{3}-1}L}\right).$$

$\square$

 **Proposition 16.** *If $\lambda_t^{(1)}$ and $\lambda_t^{(2)}$ satisfy*

$$\lambda_t^{(1)} \geq \frac{b_{t+1}^2}{B_t} M_{t+1} \left( \gamma_t + \frac{\|D_t \alpha_t\|}{2} \right), \quad \lambda_t^{(2)} \geq \frac{4}{\sqrt{3}} \frac{b_{t+1}^3}{B_t^2} M_{t+1}$$

 *Then, the function $\phi$ satisfies*

$$B_t f(x_t) \leq \phi_t(x), \qquad \phi_t(x) \leq B_t f(x) + \frac{\lambda_t^{(1)} + \tilde{\lambda}^{(1)}}{2} \|x - x_0\|^2 + \frac{\lambda_t^{(2)} + \tilde{\lambda}^{(2)}}{6} \|x - x_0\|^3,$$

 *where*

$$\tilde{\lambda}^{(1)} = \|\nabla f(x_0) - P_0 \nabla f(x_0) P_0\| + \delta \left( \frac{L\kappa + M_1 \kappa^2}{2} \right), \quad \tilde{\lambda}^{(2)} = M_1 + L.$$

 *Proof.* The result is proven by recursion. At $t = 1$, the condition $B_t f(x_t) \leq \phi_t(x)$ is obviously
 satisfied since

$$f(x_1) \leq \min_v \phi_1(v) = f(x_1).$$

 On the other hand, by proposition 9,

$$f(x_1) \leq \min_x f(x) + \frac{\tilde{\lambda}^{(2)}}{6} \|x - x_0\|^3 + \frac{\tilde{\lambda}^{(1)}}{2} \|x - x_0\|^2$$

$$\leq f(x) + \frac{\tilde{\lambda}^{(2)}}{6} \|x - x_0\|^3 + \frac{\tilde{\lambda}^{(1)}}{2} \|x - x_0\|^2.$$

 Therefore, the second condition holds by definition of $\phi$,

$$\phi_t = f(x_1) + \frac{\lambda_t^{(1)}}{2} \|x - x_0\|^2 + \frac{\lambda_t^{(2)}}{6} \|x - x_0\|^3$$

$$\leq \frac{\lambda_1^{(1)} + \tilde{\lambda}^{(1)}}{2} \|x - x_0\|^2 + \frac{\lambda_1^{(2)} + \tilde{\lambda}^{(2)}}{6} \|x - x_0\|^3.$$

 Now, assume $t > 1$, and $B_t f(x_t) \leq \phi_t(x)$. Hence,

$$\min_x \phi_{t+1}(x)$$

$$= \min_x \ell_t(x) + b_t \left[ f(x_{t+1}) + \nabla f(x_t)(x - x_{t+1}) \right] + \frac{\lambda_{t+1}^{(1)}}{2} \|x - x_0\|^2 + \frac{\lambda_{t+1}^{(2)}}{6} \|x - x_0\|^3$$

$$= \min_x \phi_t(x) + b_t \left[ f(x_{t+1}) + \nabla f(x_{t+1})(x - x_{t+1}) \right]$$

$$+ \frac{\lambda_{t+1}^{(1)} - \lambda_t^{(1)}}{2} \|x - x_0\|^2 + \frac{\lambda_{t+1}^{(2)} - \lambda_t^{(2)}}{6} \|x - x_0\|^3$$

$$\geq \min_x \phi_t(x) + b_t \left[ f(x_{t+1}) + \nabla f(x_{t+1})(x - x_{t+1}) \right]$$

$$\overset{(64)}{\geq} \min_x \phi_t^\star + \frac{\lambda_t^{(1)}}{2} \|x - v_t\|^2 + \frac{\lambda_t^{(2)}}{12} \|x - v_t\|^3 + b_t \left[ f(x_{t+1}) + \nabla f(x_{t+1})(x - x_{t+1}) \right]$$

$$\geq \min_x B_t f(x_t) + \frac{\lambda_t^{(1)}}{2} \|x - v_t\|^2 + \frac{\lambda_t^{(2)}}{12} \|x - v_t\|^3 + b_t \left[ f(x_{t+1}) + \nabla f(x_{t+1})(x - x_{t+1}) \right]$$

$$\overset{A.4}{\geq} \min_x B_t f(x_{t+1}) + \nabla f(x_{t+1})(x_t - x_{t+1}) + b_t \left[ f(x_{t+1}) + \nabla f(x_{t+1})(x - x_{t+1}) \right]$$

$$+ \frac{\lambda_t^{(1)}}{2} \|x - v_t\|^2 + \frac{\lambda_t^{(2)}}{12} \|x - v_t\|^3$$

$$= \min_x B_{t+1} f(x_{t+1}) + \nabla f(x_{t+1})(B_t x_t + b_t x - B_{t+1} x_{t+1}) + \frac{\lambda_t^{(1)}}{2} \|x - v_t\|^2 + \frac{\lambda_t^{(2)}}{12} \|x - v_t\|^3$$

$$\overset{(63)}{=} \min_x B_{t+1} f(x_{t+1}) + B_{t+1} \nabla f(x_{t+1})(y_t - x_{t+1})$$

$$+ b_t \nabla f(x_{t+1})(x - v_t) + \frac{\lambda_t^{(1)}}{2} \|x - v_t\|^2 + \frac{\lambda_t^{(2)}}{12} \|x - v_t\|^3$$

924  The inequality is satisfied if either

**(a)**  $0 \leq B_{t+1}\nabla f(x_{t+1})(y_t - x_{t+1}) + b_t\nabla f(x_{t+1})(x - v_t) + \dfrac{\lambda_t^{(2)}}{12}\|x - v_t\|^3,$  or

**(b)**  $0 \leq B_{t+1}\nabla f(x_{t+1})(y_t - x_{t+1}) + b_t\nabla f(x_{t+1})(x - v_t) + \dfrac{\lambda_t^{(1)}}{2}\|x - v_t\|^2.$

925  It remains now to find *sufficient condition* such that one of the previous inequalities hold.

926  Define $x_{t+1}$ to be the output of algorithm 4 starting from $y_t$, hence $y_t - x_{t+1} = -D_t\alpha_t$. The
927  algorithm guarantees that

$$\textbf{(b)} \quad -\nabla f(x_{t+1})^T D_t\alpha_t \geq \frac{\|f(x_{t+1})\|^2}{M_{t+1}\left(\gamma_t + \frac{\|D_t\alpha_t\|}{2}\right)}, \quad \text{or} \tag{69}$$

$$\textbf{(a)} \quad -\nabla f(x_{t+1})^T D_t\alpha_t \geq \frac{2}{3^{3/4}}\frac{\|\nabla f(x_{t+1})\|^{3/2}}{\sqrt{M_{t+1}}} \quad \text{and} \quad \|D\alpha\| \geq \Delta. \tag{70}$$

928  Combining the expressions **(a)** and **(b)** leads to the following sufficient conditions:

$$0 \leq B_{t+1}\frac{2}{3^{3/4}}\frac{\|\nabla f(x_{t+1})\|^{3/2}}{\sqrt{M_{t+1}}} + b_t\nabla f(x_{t+1})(x - v_t) + \frac{\lambda_t^{(2)}}{12}\|x - v_t\|^3, \tag{71}$$

$$0 \leq B_{t+1}\frac{\|f(x_{t+1})\|^2}{M_{t+1}\left(\gamma_t + \frac{\|D_t\alpha_t\|}{2}\right)} + b_t\nabla f(x_{t+1})(x - v_t) + \frac{\lambda_t^{(1)}}{2}\|x - v_t\|^2. \tag{72}$$

929  **Case 1: equation** (71).   Starting from the first order condition of the minimum of (71) over $x$,

$$b_t\nabla f(x_{t+1}) + \frac{\lambda_t^{(2)}}{4}\|x - v_t\|(x - v_t) = 0. \tag{73}$$

930  Multiplying (73) by $(x - v_t)$ gives

$$b_t\nabla f(x_{t+1})(x - v_t) = -\frac{\lambda_t^{(2)}}{4}\|x - v_t\|^3$$

931  Hence, when $x$ satisfies (73),

$$b_t\nabla f(x_{t+1})(x - v_t) + \frac{\lambda_t^{(2)}}{12}\|x - v_t\|^3 = -\frac{\lambda_t^{(2)}}{6}\|x - v_t\|^3. \tag{74}$$

932  Going back to (73), after isolating $x - v_t$,

$$(x - v_t) = -\frac{4b_t}{\lambda_t^{(2)}}\nabla f(x_{t+1})\frac{1}{\|x - v_t\|}$$

933  Therefore, after taking the norm and changing the power,

$$\|x - v_t\|^3 = \left(\frac{4b_t}{\lambda_t^{(2)}}\|\nabla f(x_{t+1})\|\right)^{3/2},$$

$$\Leftrightarrow \frac{\lambda_t^{(2)}}{6}\|x - v_t\|^3 = \frac{\lambda_t^{(2)}}{6}\left(\frac{4b_t}{\lambda_t^{(2)}}\|\nabla f(x_{t+1})\|\right)^{3/2}$$

$$= \frac{4}{3\sqrt{\lambda_t^{(2)}}}\left(b_t\|\nabla f(x_{t+1})\|\right)^{3/2}.$$

934  After using (74) and injecting the minimal value makes the condition (71) stronger:

$$0 \leq B_{t+1}\frac{2}{3^{3/4}}\frac{\|\nabla f(x_{t+1})\|^{3/2}}{\sqrt{M_{t+1}}} - \frac{4}{3\sqrt{\lambda_t^{(2)}}}\left(b_t\|\nabla f(x_{t+1})\|\right)^{3/2}.$$

935  Hence, if $\lambda_t^{(2)}$ satisfies

$$B_{t+1}\frac{2}{3^{3/4}\sqrt{M_{t+1}}} \geq \frac{4}{3\sqrt{\lambda_t^{(2)}}}b_t^{(3/2)} \quad \Leftrightarrow \quad \lambda_t^{(2)} \geq \frac{4}{\sqrt{3}}\frac{b_t^3}{B_{t+1}^2}M_{t+1}, \tag{75}$$

936  then (71) is satisfied.

937 **Case 2: equation** (72). Starting from the first order condition of the minimum of (72) over $x$,

$$b_{t+1}\nabla f(x_{t+1}) + \lambda_t^{(1)}(x - v_t). \tag{76}$$

938 Hence,

$$(x - v_t) = -\frac{b_t \nabla f(x_{t+1})}{\lambda_t^{(1)}}.$$

939 Injecting the value back in (72) gives

$$B_{t+1}\frac{\|f(x_{t+1})\|^2}{M\left(\gamma_t + \frac{\|D_t\alpha_t\|}{2}\right)} - b_t^2 \frac{\|\nabla f(x_{t+1})\|^2}{\lambda_t^{(1)}} + \frac{1}{2}b_t^2 \frac{\|\nabla f(x_{t+1})\|^2}{\lambda_t^{(1)}}.$$

940 Therefore, if the following condition holds,

$$\frac{B_{t+1}}{2M_{t+1}\left(\gamma_t + \frac{\|D_t\alpha_t\|}{2}\right)} \geq \frac{b_t^2}{\lambda_t^{(1)}} \quad \Leftrightarrow \quad \lambda_t^{(1)} \geq \frac{b_t^2}{2B_{t+1}}M_{t+1}\left(\gamma_t + \frac{\|D_t\alpha_t\|}{2}\right),$$

941 then (72) is satisfied.

942 $\qquad\qquad\qquad\qquad\qquad\qquad\qquad\qquad\qquad\qquad\qquad\qquad\qquad\qquad\qquad\qquad\qquad\qquad\qquad$ $\square$

943 **Proposition 17.** *Let $f$ satisfies Assumption 1. Then, under Requirements 1b to 3, using the re-scaling*
944 *technique from algorithm 6*

$$(M_0)_{t+1} \leftarrow M_{t+1}\left(\frac{\|\varepsilon_t\|}{2\|D_t\|} + \frac{\|D_t\alpha_t\|}{2}\right),$$

945 *makes $(M_0)_{t+1}$ bounded as follow:*

$$(M_0)_{t+1} \leq \frac{L}{2}(2\Delta + (2\kappa^2 + \kappa)\delta) + (2\sqrt{3} - 1)\max_{0 \leq i \leq t}\|(I - P_i)\nabla^2 f(x_i)P_i\|. \tag{77}$$

946 *Proof.* By proposition 14, for all $\Delta$, if $M_0 \leq M_{t+1}$,

$$M_{t+1} \leq L\kappa_{D_t} + \frac{2\|(I - P_t)\nabla^2 f(x)P_t\|}{\frac{\|D_t\alpha_t\|}{2} + \frac{\|\varepsilon_t\|}{2\|D_t\|}},$$

947 where $(M_0)_t$ is the initial smoothness parameter in algorithm 4. The desired result comes immediately
948 after multiplying by $\left(\frac{\|\varepsilon_t\|}{2\|D_t\|} + \frac{\|D_t\alpha_t\|}{2}\right)$, using Requirements 2 and 3 and because the re-scaling
949 technique requires $M_0 < M_{t+1}$.

950 In addition, if $\|D_t\alpha_t\|$ is sufficiently large, i.e., if

$$\|D_t\alpha_t\| \geq \max\left\{\Delta \; ; \; 2\kappa_D^2\delta + 2\max_{0 \leq i \leq t}\frac{\|(I - P_i)\nabla^2 f(x_i)P_i\|}{\frac{1}{\sqrt{3}-1}L}\right\},$$

951 then by proposition 15 the algorithm terminates when $M_{t+1} \geq \frac{2}{\sqrt{3}-1}L$. For simplicity, consider the
952 stronger condition

$$\|D_t\alpha_t\| \geq \Delta + 2\kappa_D^2\delta + 2\max_{0 \leq i \leq t}\frac{\|(I - P_i)\nabla^2 f(x_i)P_i\|}{\frac{1}{\sqrt{3}-1}L}. \tag{78}$$

953 Hence, $(M_0)_{t+1}$ cannot be larger than

$$(M_0)_{t+1}$$
$$\leq \frac{L}{2}\left(\Delta + 2\kappa^2\delta + 2\max_{0 \leq i \leq t}\frac{\|(I - P_i)\nabla^2 f(x_i)P_i\|}{\frac{1}{\sqrt{3}-1}L} + \delta\kappa\right) + \max_i\|(I - P_i)\nabla^2 f(x_i)P_i\|,$$
$$= \frac{L}{2}(2\Delta + (2\kappa^2 + \kappa)\delta) + (2\sqrt{3} - 1)\max_{0 \leq i \leq t}\|(I - P_i)\nabla^2 f(x_i)P_i\|.$$

954 $\qquad\qquad\qquad\qquad\qquad\qquad\qquad\qquad\qquad\qquad\qquad\qquad\qquad\qquad\qquad\qquad\qquad\qquad\qquad$ $\square$

**Proposition 18.** *Let $f$ satisfies Assumption 1. Then, under Requirements 1b to 3, $\lambda_t^{(1)}$ and $\lambda_t^{(2)}$ in algorithm 6 are bounded by*

$$\lambda_t^{(1)} \le 2 \cdot \frac{b_{t+1}^2}{B_t} \frac{(M_0)_{\max}^2}{L}, \tag{79}$$

$$\lambda_t^{(2)} \le 2 \cdot \frac{4}{\sqrt{3}} \frac{b_{t+1}^3}{B_t^2} \max\left\{1 \; ; \; \frac{2}{\Delta}\right\} (M_0)_{\max}. \tag{80}$$

*where*

$$(M_0)_{\max} \stackrel{def}{=} \frac{L}{2}(2\Delta + (2\kappa^2 + \kappa)\delta) + (2\sqrt{3} - 1) \max_{0 \le i \le t} \|(I - P_i)\nabla^2 f(x_i)P_i\|. \tag{81}$$

*Proof.* Since algorithm 6 doubles $\lambda_t^{(1)}$, $\lambda_t^{(2)}$ until $\phi_t^\star \ge f(x_{t+1})$, then by proposition 16, both $\lambda_t^{(1)}$, $\lambda_t^{(2)}$ achieves at most

$$\lambda_t^{(1)} \le 2 \cdot \frac{b_{t+1}^2}{B_t} M_{t+1}\left(\gamma_t + \frac{\|D_t\alpha_t\|}{2}\right), \quad \lambda_t^{(2)} \le 2 \cdot \frac{4}{\sqrt{3}} \frac{b_{t+1}^3}{B_t^2} M_{t+1}.$$

There are three cases to distinguish.

**First case.** When $(M_0)_t = M_{t+1}$, whatever the value of ExitFlag, by proposition 17, and by construction of algorithm 6, $(M_0)_t$ is bounded as follow:

$$(M_0)_t \le \max\left\{\frac{(M_0)_{t-1}}{2} \; ; \; \frac{L}{2}(2\Delta + (2\kappa^2 + \kappa)\delta) + (2\sqrt{3} - 1) \max_{0 \le i \le t} \|(I - P_i)\nabla^2 f(x_i)P_i\|.\right\}.$$

In the worst case, the maximum is attained in the right hand side. For simplicity, let $(M_0)_{\max}$ be defined as

$$(M_0)_{\max} \stackrel{def}{=} \frac{L}{2}(2\Delta + (2\kappa^2 + \kappa)\delta) + (2\sqrt{3} - 1) \max_{0 \le i \le t} \|(I - P_i)\nabla^2 f(x_i)P_i\|.$$

In this case, $\lambda_1^{(t)}$ and $\lambda_t^{(2)}$ are bounded by

$$\lambda_t^{(1)} \le 2 \cdot \frac{b_{t+1}^2}{B_t}(M_0)_{\max}\left(\gamma_t + \frac{\|D_t\alpha_t\|}{2}\right)$$

$$\lambda_t^{(2)} \le 2 \cdot \frac{4}{\sqrt{3}} \frac{b_{t+1}^3}{B_t^2}(M_0)_{\max}. \tag{82}$$

By Requirements 2 and 3, $\gamma_t$ is bounded by

$$\gamma_t = \frac{1}{2} \frac{\|\varepsilon_t\|}{\|D_t\|} \frac{1 + \kappa_{D_t}^2}{2} \le \frac{1 + \kappa^2}{4}\delta.$$

Moreover, under the condition (78),

$$\|D_t\alpha_t\| \ge \Delta + 2\kappa_D^2\delta + 2 \max_{0 \le i \le t} \frac{\|(I - P_i)\nabla^2 f(x_i)P_i\|}{\frac{1}{\sqrt{3}-1}L},$$

the algorithm algorithm 4 terminates with ExitFlag equals to LargeStep. Hence, to update $\lambda_t^{(1)}$,

$$\|D_t\alpha_t\| \le \Delta + 2\kappa_D^2\delta + 2 \max_{0 \le i \le t} \frac{\|(I - P_i)\nabla^2 f(x_i)P_i\|}{\frac{1}{\sqrt{3}-1}L} \tag{83}$$

Therefore,

$$\begin{aligned}
\gamma_t + \frac{\|D_t\alpha_t\|}{2} &\le \frac{1 + \kappa^2}{4}\delta + \frac{1}{2}\left(\Delta + 2\kappa^2\delta + 2 \max_{0 \le i \le t} \frac{\|(I - P_i)\nabla^2 f(x_i)P_i\|}{\frac{1}{\sqrt{3}-1}L}\right) \\
&\le \frac{1}{L}\left(\frac{L}{2}\left(\frac{1 + 5\kappa^2}{2}\delta + \Delta\right) + (\sqrt{3} - 1) \max_{0 \le i \le t} \|(I - P_i)\nabla^2 f(x_i)P_i\|\right) \\
&\le \frac{(M_0)_{\max}}{L}.
\end{aligned}$$

By consequence,

$$\lambda_t^{(1)} \leq 2 \cdot \frac{b_{t+1}^2}{B_t}(M_0)_{\max}\left(\gamma_t + \frac{\|D_t\alpha_t\|}{2}\right)$$

$$\leq 2 \cdot \frac{b_{t+1}^2}{B_t}\frac{(M_0)_{\max}^2}{L}.$$

**Second case.** When $M_0 \leq M_t$ and `ExitFlag` equals `SmallStep`, only $\lambda_1$ is updated. Therefore,

$$\lambda_t^{(1)} \leq 2 \cdot \frac{b_{t+1}^2}{B_t}M_{t+1}\left(\gamma_t + \frac{\|D_t\alpha_t\|}{2}\right).$$

Since $M_0 \leq M_{t+1}$, by proposition 14, then by Requirements 2 and 3,

$$M_{t+1}\left(\gamma_t + \frac{\|D_t\alpha_t\|}{2}\right) \leq (1 + \kappa_{D_t}^2)\left(\frac{L}{2}\left(\|D_t\alpha_t\| + \frac{\|\varepsilon_t\|}{\|D_t\|}\kappa_{D_t}\right) + \|(I - P_t)\nabla^2 f(x_t)P_t\|\right),$$

$$\leq (1 + \kappa^2)\left(\frac{L}{2}\left(\|D_t\alpha_t\| + \delta\kappa\right) + \max_{0 \leq i \leq t}\|(I - P_i)\nabla^2 f(x_i)P_i\|\right)$$

Because `ExitFlag` is `SmallStep`, $\|D\alpha\|$ is bounded by (83). Hence,

$$M_{t+1}\left(\gamma_t + \frac{\|D_t\alpha_t\|}{2}\right) \leq (1 + \kappa^2)\left[\frac{L}{2}\left(\Delta + 2\kappa_D^2\delta + 2\max_{0 \leq i \leq t}\frac{\|(I - P_i)\nabla^2 f(x_i)P_i\|}{\frac{1}{\sqrt{3}-1}L} + \delta\kappa\right)\right.$$

$$\left. + \max_{0 \leq i \leq t}\|(I - P_i)\nabla^2 f(x_i)P_i\|\right]$$

$$= (1 + \kappa^2)\left[\frac{L}{2}\left(\Delta + (2\kappa_D^2 + \kappa)\delta\right) + \sqrt{3}\max_{0 \leq i \leq t}\|(I - P_i)\nabla^2 f(x_i)P_i\|\right]$$

$$\leq (1 + \kappa^2)(M_0)_{\max}.$$

Since $(1 + \kappa^2) \leq \frac{(M_0)_{\max}}{L}$,

$$M_{t+1}\left(\gamma_t + \frac{\|D_t\alpha_t\|}{2}\right) \leq \frac{(M_0)_{\max}^2}{L}.$$

Hence,

$$\lambda_t^{(1)} \leq 2 \cdot \frac{b_{t+1}^2}{B_t}\frac{(M_0)_{\max}^2}{L}.$$

**Third case.** It remains to bound $\lambda_t^{(2)}$, when $(M_0)_t \leq M_{t+1}$ and `ExitFlag` equals `LargeStep`. In such a case, by proposition 14,

$$M_{t+1} \leq 4\frac{1}{\|D_t\alpha_t\| + \frac{\|\varepsilon_t\|}{\|D_t\|}}\left(\frac{L}{2}\left(\|D_t\alpha_t\| + \frac{\|\varepsilon_t\|}{\|D_t\|}\kappa_{D_t}\right) + \|(I - P_t)\nabla^2 f(x_t)P_t\|\right)$$

Note that, for all $a, b$, the function $\frac{x+a}{x+b}$ is decreasing as long as $b < a$. Hence, since `ExitFlag` equals `LargeStep`, $\|D_t\alpha_t\| \geq \Delta$ and

$$\frac{\|D_t\alpha_t\| + \frac{\|\varepsilon_t\|}{\|D_t\|}\kappa_{D_t}}{\|D_t\alpha_t\| + \frac{\|\varepsilon_t\|}{\|D_t\|}} \leq \frac{\Delta + \frac{\|\varepsilon_t\|}{\|D_t\|}\kappa_{D_t}}{\Delta + \frac{\|\varepsilon_t\|}{\|D_t\|}},$$

and therefore, using Requirements 2 and 3 leads to

$$M_{t+1} \leq 4\frac{1}{\Delta + \frac{\|\varepsilon_t\|}{\|D_t\|}}\left(\frac{L}{2}\left(\Delta + \frac{\|\varepsilon_t\|}{\|D_t\|}\kappa_{D_t}\right) + \|(I - P_t)\nabla^2 f(x_t)P_t\|\right)$$

$$\leq 4\frac{1}{\Delta}\left(\frac{L}{2}\left(\Delta + \frac{\|\varepsilon_t\|}{\|D_t\|}\kappa_{D_t}\right) + \|(I - P_t)\nabla^2 f(x_t)P_t\|\right)$$

$$\leq 4\frac{1}{\Delta}\left(\frac{L}{2}\left(\Delta + \delta\kappa\right) + \max_{0 \leq i \leq t}\|(I - P_i)\nabla^2 f(x_i)P_i\|\right)$$

$$\leq 2\frac{(M_0)_t}{\Delta}.$$

Therefore, after combining this inequality with (82),

$$\lambda_t^{(2)} \le 2 \cdot \frac{4}{\sqrt{3}} \frac{b_{t+1}^3}{B_t^2} \max\left\{1 \; ; \; \frac{2}{\Delta}\right\} (M_0)_{\max}.$$

□

**Theorem 7.** *Assume $f$ satisfy Assumptions 1, 2 and 4. Let Requirements 1b to 3 hold. Then, algorithm 5 starting at $x_0$ with $M_0$ achieves, for all $\Delta > 0$ and for $t \ge 1$,*

$$f(x_t) - f^\star \le \frac{(M_0)_{\max}^2}{L}\left(\frac{3R}{t+3}\right)^2 + \frac{4(M_0)_{\max}}{3\sqrt{3}}\max\left\{1 \; ; \; \frac{2}{\Delta}\right\}\left(\frac{3R}{t+3}\right)^3 + \frac{\frac{\tilde{\lambda}^{(1)}R^2}{2} + \frac{\tilde{\lambda}^{(2)}R^3}{6}}{(t+1)^3}.$$
*where* $\tilde{\lambda}^{(1)} = 0.5 \cdot \delta\left(L\kappa + M_1\kappa^2\right) + \|\nabla f(x_0) - P_0\nabla f(x_0)P_0\|, \qquad \tilde{\lambda}^{(2)} = M_1 + L,$
$$(M_0)_{\max} = \frac{L}{2}(2\Delta + (2\kappa^2 + \kappa)\delta) + (2\sqrt{3} - 1)\max_{0 \le i \le t}\|(I - P_i)\nabla^2 f(x_i)P_i\|.$$

*Proof.* By construction of $\phi_t(x)$, from proposition 16 and Assumption 2,

$$B_t f(x_t) \le \min_x \phi_t(x) \tag{84}$$

$$\le \phi_t(x^\star) \tag{85}$$

$$\le B_t f(x^\star) + \frac{\lambda_t^{(1)} + \tilde{\lambda}^{(1)}}{2}\|x^\star - x_0\|^2 + \frac{\lambda_t^{(2)} + \tilde{\lambda}^{(2)}}{6}\|x^\star - x_0\|^3 \tag{86}$$

$$\le B_t f(x^\star) + \frac{\lambda_t^{(1)} + \tilde{\lambda}^{(1)}}{2}R^2 + \frac{\lambda_t^{(2)} + \tilde{\lambda}^{(2)}}{6}R^3 \tag{87}$$

$$\Rightarrow f(x_t) - f^\star \le \frac{\lambda_t^{(1)} + \tilde{\lambda}^{(1)}}{2B_t}R^2 + \frac{\lambda_t^{(2)} + \tilde{\lambda}^{(2)}}{6B_t}R^3. \tag{88}$$

By proposition 18, the following bounds holds:

$$\lambda_t^{(1)} \le 2 \cdot \frac{b_{t+1}^2}{B_t}\frac{(M_0)_{\max}^2}{L},$$

$$\lambda_t^{(2)} \le 2 \cdot \frac{4}{\sqrt{3}}\frac{b_{t+1}^3}{B_t^2}\max\left\{1 \; ; \; \frac{2}{\Delta}\right\}(M_0)_{\max}.$$

Hence,

$$f(x_t) - f^\star \le \frac{2 \cdot \frac{b_{t+1}^2}{B_t}\frac{(M_0)_{\max}^2}{L} + \tilde{\lambda}^{(1)}}{2B_t}R^2 + \frac{2 \cdot \frac{4}{\sqrt{3}}\frac{b_{t+1}^3}{B_t^2}\max\left\{1 \; ; \; \frac{2}{\Delta}\right\}(M_0)_{\max} + \tilde{\lambda}^{(2)}}{6B_t}R^3.$$

Since $\frac{b_{t+1}}{B_t} = \frac{3}{(t+3)}$,

$$\frac{b_{t+1}^3}{B_t^3} = \frac{3^3}{(t+3)^3}, \tag{89}$$

$$\frac{b_{t+1}^2}{B_t^2} = \frac{3^2}{(t+3)^2}. \tag{90}$$

$$\tag{91}$$

Therefore,

$$f(x_t) - f^\star \le \frac{(M_0)_{\max}^2}{L}\left(\frac{3R}{t+3}\right)^2 + \frac{4(M_0)_{\max}}{3\sqrt{3}}\max\left\{1 \; ; \; \frac{2}{\Delta}\right\}\left(\frac{3R}{t+3}\right)^3 + \frac{\frac{\tilde{\lambda}^{(1)}R^2}{2} + \frac{\tilde{\lambda}^{(2)}R^3}{6}}{(t+1)^3}.$$

□