# OpenReview forum: "Adaptive Quasi-Newton and Anderson Acceleration Framework with Explicit Global Convergence Rates"
_NeurIPS.cc/2023/Conference — Submitted to NeurIPS 2023_

### Official Review · Reviewer_dfNf · 2023-07-04

**Soundness:** 3 good
**Presentation:** 3 good
**Contribution:** 3 good
**Rating:** 7
**Confidence:** 4

**Summary:**

This paper presents first-order iterative methods for solving unconstrained minimization problem. The connection between the proposed methods, quasi-Newton and Anderson accelaration methods are illustrated, which gives an insight of the motivation. Under certain assumptions, the methods exhibit explicit gobal non-asympotic convergence rates adaptively using backtracking strategy. The effiency of the propsed methods is compared to a fine-tuned BFGS algorithm with line search in the numerical experiments.

**Strengths:**

- The problem is well-motivated.

- Under certain assumptions, the proposed adaptive first-order methods achieve explicit global convergence rates that blend those of gradient descent and cubic regularized Newton's method.

- The inputs of the algorithms are carefully discussed. It should be clear for others to implement.

- The algorithm complexity is analyzed and compared in the numerical experiments.

**Weaknesses:**

- Too many assumptions and requirements in the theoretical part. Can those requirements be verified? Seems the requirements are post to let the proof go through.

- The algorithm setting for the numerical experiments in section 6 is a bit confusing. For example, what online techniques are used for 'Iterate Only', 'Accelareted Forward Only', 'Forward Estimate Only' and 'Greedy'?

- The performance of the accelarated algorithm is suboptimal and unstable.

**Questions:**

- Section 1.1 Contributiions:
  - any reference to support the statement in bold: that has remained unanswered for over 50 years.
  - for strongly convex cases, can the author prove the linear or super-linear convergence rate?
- What is the motivation to use the update formula for $H$ in line 3 of Algorithm 1?
- When taking orthogonal directions for $D$, the orthogonality is metric-dependent. For example, in conjugate gradient methods, it is $A$-orthogoal.
- What is the definition of $r_i$ in the equation below line 104?
- In Requirement 1a: where is the randomness of matrix $D_t$? Before that, all presentaion is for determinstic algorithms.
- What is the step size $h$ in the methods below line 236? By the definition of steepest, it is chosen by the exact line search. But a small $h=10^{-9}$ is used. Then it is called gradient descent not steepest gradient method. And what is the reason to choose  such a tiny step size.
- What is the 'Greedy' refers to in line 258?
- Should the right hand side of inquality in Theorem 5 be written as $f(x_0) - f^\star$ in the numerator of the first term?
- Typos: line 7, 'cubic regularized'; line 197, 'Theorem 6'; line 237, 'includes'.

**Limitations:**

- The performance of the accelarated algorithm is suboptimal and unstable.
- Analysis is provided only for Algorithm 1 not Algorithm 2 while the later seems better.

---

> ### Author Rebuttal · Authors · 2023-08-09
>
> # Author's response
>
> We appreciate the reviewer’s positive feedback. We will do our best to answer the reviewer’s concerns.
>
> ## Weaknesses
>
> > **Too many assumptions and requirements in the theoretical part. Can those requirements be verified?**
>
> See **SDP, A)** for more details. Those requirements are not restrictive, as we provide a few strategies to meet them in Section 5. For instance, the orthogonal forward estimates only satisfy those requirements:
> - Requ. 1b, because the last column of $D_t$ is the orthogonal projection of $\nabla f(x_t)$ onto the span of the other columns of $D_{t}$,
> - Requ. 2, because $||D_t|| = h^2$ and $||\epsilon||=O(hR)$,
> - Requ. 3, because the columns of $D_t$ are orthogonal with norm $h$, hence $\kappa = 1$.
>
> ---
>
> > **The algorithm setting for the numerical experiments in section 6 is a bit confusing.**
>
> We are committed to enhancing the clarity of the paper as suggested. However, we kindly request clarification on the specific part that the reviewer found unclear.
>
> We are unsure if we did answer the reviewer’s question correctly here: as explained in Section 5.1, the matrices are updated in an “online” fashion by adding successive new information. However, we forgot to mention we used the *orthogonal* forward estimate for all methods.
>
> We thank the reviewer in advance for the additional clarification.
>
> ---
>
> > **The performance of the accelerated algorithm is suboptimal and unstable.**
>
> We agree with the reviewer that the accelerated algorithm does not perform better than the non-accelerated ones. As explained in the paper, the accelerated method’s performance is suboptimal, possibly because it tightens the theoretical analysis, diminishing its inherent adaptivity.
>
> ## Questions
>
> > **any reference to support the statement in bold: that has remained unanswered for over 50 years.**
>
> See the citation from [50] in the global response.
>
> ---
>
> > **for strongly convex cases, can the author prove the linear or super-linear convergence rate?**
>
> **Linear convergence:** It seems that, numerically, the algorithm converges at a linear rate of convergence on strongly convex functions. Given our rates are not too different from first-order methods, we might show linear convergence as well. However, since the paper is already quite heavy in term of theoretical results, we left this analysis for future work.
>
> **Superlinear convergence:** Given our bounds, we might be able to show a super-linear convergence rate for strongly convex functions as well, but only if we consider a case where $N=d$, which is not suitable for large-scale problems. The study of $N=d$ is left for future work.
>
> ---
>
> > **What is the motivation to use the update formula for $H$ in line 3 of Algorithm 1?**
>
> Very good question! We will add this explanation in the camera-ready: The Hessian satisfies approximately the following property: $\nabla f(x)D\approx G$. Hence, after multiplying by $D$, we have $D^T\nabla f(x)D\approx D^TG$. Since both sides are multiplied by $\alpha$, only the symmetric part counts. It suffices to add regularization that takes into account the error of approximation, and this gives the update.
>
> ---
>
> > **When taking orthogonal directions for $D$, the orthogonality is metric-dependent.**
>
> Very interesting point. The orthogonalization was motivated by theory, to reduce the constant $\kappa$. However, by using another metric, we might improve the performance of the Type-2 method (which uses the matrix $G$).
>
> ---
>
> > **What is the definition of $r_i?$**
>
> The $r_i$ should be $g_i$.
>
> ---
>
> > **In Requirement 1a: where is the randomness of matrix $D$?**
>
> There was no assumption on the matrix $D$ beforehand, hence it can be chosen at random.
>
> ---
>
> > **What is the step size $h$ in the methods below line 236?**
>
> The step size is chosen to be as small as possible to improve the bounds in the algorithm. The smaller the step size is, the better the Hessian is approximated. We tried several strategies (e.g., line search), but as long as the step size is not too big (e.g., takes divergent steps), the methods behave the same.
>
> ---
>
> > **What is the 'Greedy' refers to in line 258?**
>
> The “greedy” strategy is described in line 248: it stores the last $N$ directions, regardless of whether they were generated by the forward estimate or the algorithm’s step. Then, we use QR regularization ($O(N^2d)$) to orthogonalize the directions, and then compute their associated pairs of gradients.
>
> ---
>
> > **the right-hand side of the inequality in Theorem 5**
>
> We corrected the typo.
>
> ## Limitations
>
> > **The performance of the accelerated algorithm is suboptimal and unstable.**
>
> As explained in the paper, the accelerated method’s performance is suboptimal,
> possibly because it tightens the theoretical analysis, diminishing its inherent adaptivity. The “stability” issue probably comes from the fact that the update of parameters depends on whether the algorithm performs small or large steps.
>
> ---
>
> > **Analysis is provided only for Algorithm 1, not Algorithm 2, while the latter seems better.**
>
> The analysis of Algorithm 2 is left for future work: we expect algorithm 2 to shine on variational inequalities or strongly convex functions. The main reason is that Algorithm 2 does not rely on the existence of an objective function $f$, it focuses only on minimizing the gradient norm.
>
> ---
>
> We hope we have answered to all the reviewer's concerns.

---

> > ### Comment · Reviewer_dfNf · 2023-08-16
> >
> > Thank you for response. After reading other reviews and the rebuttal, I have decided to maintain my current score as it stands.

---

> > > ### Author Response · Authors · 2023-08-17
> > >
> > > Dear reviewer,
> > >
> > > Thank you again for your time and expertise! We are glad to see that our rebuttal has answered your questions.
> > >
> > > Best,

---

### Official Review · Reviewer_4Qfc · 2023-07-06

**Soundness:** 2 fair
**Presentation:** 2 fair
**Contribution:** 3 good
**Rating:** 4
**Confidence:** 3

**Summary:**

The paper proposes a generic framework for developing new iterative schemes for smooth optimization in the deterministic setting. The derived new methods show some similarity to quasi-Newton methods and Anderson acceleration, while using a backtracking line search for estimating the Lipschitz constant and having step sizes adaptively determined by minimizing an upper bound of the objective function or the gradient norm. The explicit, global and non-asymptotic convergence rates are established for one type of the derived methods. Numerical results are presented for some specific implementations of the framework.

**Strengths:**

The paper proposes a novel framework inspired by the recent development of cubic regularized Newton's method. This framework leads to several optimization methods that can be viewed as new variants of quasi-Newton methods and Anderson acceleration. The theory seems to be solid. The explicit, non-asymptotic convergence rates are established under different assumptions of the optimization problems, justifying the introduced techniques. The content is informative and the main idea is easy to follow. Numerical results are presented to support some claims of the paper.

**Weaknesses:**

The weaknesses are listed as follows:

1. The theory requires the Lipschitz continuity of the Hessian in $\mathcal{R}^d$, which is a strong assumption in practice.

2. The theorems only consider the iteration number towards convergence, which may overestimate the efficiency of the algorithms. Since the number of function evaluations can be large in each iteration, it needs to be clarified whether the claimed convergence rates are still valid when taking these hidden calculations into account.

3. The biggest drawback of the proposed methods is their suitability for solving high-dimensional problems. To well approximate the Hessian which is critical for the convergence, the memory usage needs to be large, which can cause failure of the algorithms when the memory resource is limited. What's worse, the algorithms are very complicated. They use a backtracking line search to estimate the Lipschitz constant and solve a minimization problem to determine the stepsizes in each inner step of the line search. It is unknown how many times the minimization problem needs to be solved during each iteration. It is likely that the computational cost of each iteration is much higher than the classical quasi-Newton methods or Anderson acceleration. However, there is no discussion about this issue.

4. The algorithms were only tested for solving small-scale logistic regression problems. Some numerical results do not support the theory. For instance, the accelerated method does not exhibit any acceleration in practice. In many cases, the simple BFGS with the default setting still achieves the best performance, while the proposed methods are less efficient due to higher costs.

5. Since many additional calculations are hidden in the subroutines of the algorithms, it is more convincing to report comparisons of the considered algorithms with respect to running time. However, these results are missing.

There are also some other concerns about the paper:

1. In Section 2, the paper claims that quasi-Newton methods and Anderson acceleration share the common property that the iterates are combinations of previous iterates and the current gradient. This claim may be true for quadratic optimization but seems to be wrong in general cases. There is no justification for this property throughout the paper. The presented framework is more like a generalized heavy-ball or Nesterov-like method, or can be viewed as an adaptive version of the subspace Newton method. Its connection with quasi-Newton methods and Anderson acceleration is not clear. It is misleading if such a connection is not valid.

2. Although the presented framework does not use the traditional line-search or trust-region technique to guarantee global convergence, it uses a more costly backtracking line-search strategy to estimate the Lipschitz constant. Such complexity can impair the significance of this work.

3. Section 1.2 says that Anderson acceleration does not generalize well outside quadratic minimization. It is not true since Anderson acceleration is well-known for its usefulness in accelerating fixed-point iterations in scientific computing.

4. Some descriptions of the classical methods are not standard. The formulas of BFGS and Anderson acceleration in Section 2 are quite strange. The formula of BFGS below Line 97 seems to be wrong. The formula of Anderson acceleration below Line 104 is incorrect since $d_0$ and $g_0$ are undefined. The formula below Line 513 is not the Anderson acceleration.

5. Many notations are not defined clearly, e.g., $P_i$ in Equation (2), $r_i$ below Line 104, and $R^\dagger$ in Line 600.

6. The paper discusses many possible methods derived from the framework, but the pros and cons of each method lack clear clarifications. For example, the orthogonalized greedy method seems to outperform BFGS in some cases, but it also doubles the memory usage. So the comparison in the experiments may be unfair.

7. The proofs need to be reorganized to make them easy to follow. It is better to give more examples of the introduced notations and assumptions.

**Questions:**

1. How to derive (2), and what is the explicit form of $P_i$?

2. In Section 5.1, how to guarantee the linear independence of the columns of $D$? When $P_{t-1}\nabla f(x_t) = \nabla f(x_t)$, the orthogonal option leads to $x_{t+1/2} = x_t$. Then how to define the iteration, and is the convergence theory still valid in this case?

3. Section B.2.1 claims that the SR-1 method solves (21). It seems to be wrong, since (21) corresponds to the Powell-symmetric-Broyden method. The author(s) may provide some references or calculations to support this claim.

**Limitations:**

Some limitations have been mentioned in the main paper, but there is no discussion about the total computation cost and memory cost.

---

> ### Author Rebuttal · Authors · 2023-08-06
>
> # Author's response
>
> We thank the reviewer for their thoughtful evaluation of our paper. While we genuinely appreciate their feedback, we must respectfully address some concerns regarding inaccurate statements made in the review.
>
> ### A) Scalability for large-scale problems.
>
> > **The algorithm might not be suitable for solving high-dimensional problems (point 1.3 and 1.4). To well approximate the Hessian, which is critical for the convergence, the memory usage needs to be large, which can cause failure**
>
> We politely disagree with the reviewer's claim:
>
> - While better approximating the Hessian, a larger memory is not critical for convergence. For limited-memory QN methods, small N tends to perform better in practice. This is due to keeping too old secant equations and the diminishing return of adding more information.
>
> - Moreover, all operations scale linearly w.r.t the dimension (see attached pdf file); therefore, our algorithm is suitable for large-scale problems. In fact, **both memory and per-iteration complexity are the same as Anderson Acceleration**: $O(N^3+N^2d)$. For large $N$,  the complexity can be improved to $O(N^2+Nd)$ with online updates for H and an iterative solver for the cubic subproblem.
>
> - We also kindly indicate to the reviewer that those concerns are not specific to our contribution - in fact, **as for all QN methods, scaling the memory with $d$ will inevitably lead to complexities of at least $O(d^2)$**.
>
> ### B) Line-search and the total number of iteration
> > **The backtracking line search is complicated/costly. How often must the subproblem be solved (points 1.2, 1.3, 2.2)?**
>
> The backtracking line search technique is known to be simple yet effective in estimating the Lipchitz parameter, preferable both in terms of implementation and running time than Wolfe line search or trust region methods. More precisely, with this technique, it is well known that after $T$ Iterations, **the total number of gradient evaluations cannot exceed $2T + log_2(M_0/L)$** [1, Sec 5.2]. This will be made clear in the paper.
>
> ---
>
> ## Detailed rebuttal
>
> ### Part 1
>
> > **1.1 Lipschitz continuity of $\nabla^2 f(x)$ is a strong assumption**
>
> The Lipschitz continuity assumption is standard in analyzing second-order and quasi-Newton methods (e.g. [1]) and is not considered too restrictive in the literature.
>
> ---
>
> > **1.2 The theorems only consider the iteration number, not the function evaluations.**
>
> The cubic subproblem does not make additional calls to the first-order oracle and is well-studied in the literature, e.g., [1] [2]. Only the line search could make additional calls to the function, but this is bounded by O(2T) (see **B)**).
>
> ---
>
> > **1.3  Not suitable for solving high-dimensional problems**
>
> See **A)**, **B)**, and the attached PDF file.
>
> ---
>
> > **1.4 and 1.5 The algorithms were only tested for solving small-scale problems. It is more convincing to report running time.**
>
> The attached PDF file shows the algorithm works well in high dimensions ($d\approx 10^7$).
>
> Moreover, the running time might not be relevant as, compared to BFGS, our prototype implementation contains additional checks/computations for debugging purposes. Nevertheless, for larger-scale problems, the subroutine computation time is completely negligible compared to computing a new gradient.
>
> ---
>
> ### Part 2
>
> >  **2.1 The statement “QN methods and Anderson acceleration combine previous iterations with the current gradient” is valid only for quadratics.**
>
> Eq. (2) shows that QN and Anderson acceleration generates points in the span of the previous direction and gradients on non-quadratic problems. The reviewers might have been confused by the fact that, on quadratics, the span of previous directions multiplied with the Hessian is the span of previous differences of gradients.
>
> ---
>
> > **2.2 The backtracking line search is costly**
>
> See **B)**
>
> ---
>
> > **2.3 Anderson acceleration generalizes outside quadratics**
>
> Anderson Acceleration is very effective for fixed point iterations but is unstable in many cases, which is why it is combined with line-search, regularization, globalization, or trust region technique ([62, 68, 56]).
>
> ---
>
> > **2.4 The formulas of BFGS and Anderson acceleration are strange**
>
> The BFGS formula is the same but reorganized, highlighting that after multiplying the matrix with a gradient, it combines previous directions and gradients. Anderson acceleration is known under many variants; we presented one of them.
>
> ---
>
> > **2.5 Notations are not defined**
>
> $r_i$ should be $g_i$. $P_i$ should be $I-g_id_i^T/g_i^Td_i$. $R^\dagger$ is a pseudo-inverse of $R$.
>
> ---
>
> > **2.6 The pros and cons of each method need clarification. The comparison in the experiments may be unfair.**
>
> We already added a section in the appendices explaining each method's pros and cons in Section 5.
>
> The experiments are as fair as possible: all methods use the same memory (including the greedy method). The number of oracle calls is explicitly shown as all our methods use a different LS technique (compared to BFGS).
>
> ---
>
> > **2.7 Unclear proof organization**
>
> We will improve the clarity of the proof by including, for instance, a table of notation.
>
> ### Questions
>
> > **1) How to derive (2), and what is the explicit form of $P_i$?**
>
> See answer to Reviewer FnQv, Question 10.
>
> ---
>
> > **2) How to guarantee the linear independence of the columns of $D$?**
>
> Either ignoring the hessian update or removing the oldest vector to ensure linear independence works: In both cases, all results still hold since Requ. 1b is met. Note that we never encountered such a case in all of our experiments.
>
> ---
>
> > **3) Equ. (21) corresponds to the Powell-symmetric-Broyden method.**
>
> We corrected the mistake.
>
> ### References
>
> [1] Nesterov et al., *Cubic regularization of Newton’s method*
>
> [2] Zhu et al., *Solving the Adaptive Cubic Regularization Sub-Problem Using the Lanczos Method*

---

> > ### Comment · Reviewer_4Qfc · 2023-08-16
> >
> > Thank you for the response. I have read the rebuttal and decided to keep the score.

---

> > > ### Author Response · Authors · 2023-08-16
> > >
> > > Dear Reviewer,
> > >
> > > We believe we have addressed the main concerns of your review:
> > > - The proposed algorithm scales as well as current QN methods (even in running time) but comes with theoretical guarantees (see **A** in the rebuttal, and the attached PDF file).
> > > - The number of function evaluations is bounded (see **B** in the rebuttal).
> > >
> > > The score of **4** in the NeurIPS guidelines mentions the following:
> > > > Technically solid paper where reasons to reject, e.g., limited evaluation, outweigh reasons to accept, e.g., good evaluation.
> > >
> > > We kindly request the reviewer to elaborate on the specific reasons for considering rejection so that we can improve the paper accordingly.
> > >
> > > Best,
> > > The authors.

---

### Official Review · Reviewer_SDpo · 2023-07-07

**Soundness:** 2 fair
**Presentation:** 2 fair
**Contribution:** 2 fair
**Rating:** 4
**Confidence:** 4

**Summary:**

The authors introduce a generic framework for developing novel quasi-Newton and Anderson/Nonlinear acceleration schemes, offering a global convergence rate in various scenarios, including accelerated convergence on convex functions. They also provide empirical results in the numerical experiments.

**Strengths:**

The structure is clear and the theoretical analysis and proof are correct. The results presented in the numerical experiments section are consistent with the theoretical results.

**Weaknesses:**

There is the following weakness regarding this submission.

1. First, the authors ignores a lot of existing quasi-Newton type methods or accelerated versions that achieve the non-asymptotic global convergence rates of $\mathcal{O}(1/k)$ and $\mathcal{O}(1/k^2)$ such as:

“Practical inexact proximal quasi-Newton method with global complexity analysis”. Katya Scheinberg and Xiaocheng Tang. Mathematical Programming160(2016), pp.495–529.

“Proximal quasi Newton methods for regularized convex optimization with linear and accelerated sublinear convergence rates”. Hiva Ghanbari and Katya Scheinberg. Computational Optimization and Applications69(2018),pp.597–627

“Accelerated Quasi-Newton Proximal Extragradient: Faster Rate for Smooth Convex Optimization”. R. Jiang and A. Mokhtari. https://arxiv.org/abs/2306.02212

Recently there also exists a paper proposing the quasi-Newton type method that could achieve a global explicit superlinear convergence rate:

“Online Learning Guided Curvature Approximation: A Quasi-Newton Method with Global Non-Asymptotic Superlinear Convergence”. R. Jiang, Q. Jin, A. Mokhtari. Conference on Learning Theory (COLT), 2023.

However, this global superlinear convergence rate assumes strong convexity of the objective function. The authors should compare the results in this submission to all these related literature in detail to check if the proposed algorithm could achieve an improvement in the aspect of convergence rate or computational cost per iteration.

2. The authors didn't compare the results of the this paper to the global convergence rates of first-order gradient descent and accelerated gradient descent. The authors claimed that the proposed algorithm could match the results of accelerated gradient descent, but is the constant in the convergence rate of this proposed method better than the constant of accelerated gradient descent? What is the improvement of this algorithm compared with the accelerated gradient descent? Notice that the computational cost per iteration of this QN-type method is worse than the computational cost per iteration of accelerated gradient descent.

3. It's better for the authors to use the notations of $s_t = x_{t + 1} - x_t$ and $y_t = \nabla{f}(x_{t + 1}) - \nabla{f}(x_{t})$. These notations are more commonly-used in the quasi-Newton methods.

4. The notations used from equations (3) to (7) are quietly confusing. What's the exact definition of $y_i$ and $z_i$? Is it $y_i = x_{t - i + 1}$ and $z_i = x_{t - i}$. Also it seems that $D$, $G$ and $\epsilon$ depend on the iterations index $t$. Then it should be $D_t$, $G_t$ and $\epsilon_t$. These notations are messed and make the readers difficult to understand and follow the ideas of the authors.

5. What is the definition or function of the parameters $M_0$, $M_t$ and $M_{min}$ presented in the algorithms? The authors said that the $M_0$ is the initial guess of the smoothness parameter, So $M$ is an estimation of the parameter $L$ in assumption 1? What's the value of $M_0$ in all the numerical experiments presented in this paper? What theoretical conditions should this $M_0$ satisfy?

6. The authors presented some designed requirements in section 3.1. These expressions of requirements are a bit strange for the theoretical results. The authors should either make these requirements as the assumptions of the algorithms or make these requirements as some lemmas/theorems of the theoretical analysis. However, these requirements are too strong or restrictive as assumptions. On the other hand, the authors didn't give any strict mathematical proof to give any conditions to satisfy these requirements. The authors argue that these requirements are not restrictive in the text of section 3.1. But this is not enough for the theoretical analysis of the algorithm. We need strict and clear mathematical proof or empirical results from the numerical experiments. It is not clear how to make these requirements to be satisfied. There is no formal proof.

7. The authors claimed that $N$ could be small around line 149. However, as expressed in the theorem 6, it seems that $N$ should be comparable to the dimension $d$ to reach a good convergence rate. But when $N = \mathcal{O}(d)$, the computational cost of solving the sub-problems could be as expensive as $\mathcal{O}(d^3)$ as presented in line 148. This is as costly as the Newton's method.

8. From line 203 to line 210, the authors argue that some propertied needed to be satisfied to retrieve the convergence rate of Newton’s method with cubic regularization. However, it seems that when these properties are satisfied, the computational complexity is the same as the Newton's method. Then, what is the advantages or improvements of this method compared to Newton’s method with cubic regularization?

9. What are the definitions of $b_i$ and $\lambda_t^{(1, 2)}$ in the equation below line 213? The authors should explain these parameters for the accelerated algorithms.

10. The most significant weakness or drawback of this submission is the lack of formal algorithm for the update of matrices $Y, Z, D, G$. Although the authors presented the online and batch techniques in section 5, these descriptions of the update scheme is just an outline and too introductory. We need a lot of details and formal description of the algorithm for the implementations or updates of matrices $Y, Z, D, G$. Without explicit algorithm like Algorithm 1 in the paper, there are a lot of unclear operations regarding the update of the corresponding matrices. The authors should present a formal algorithm regarding the implementations of these updates in the appendix. There also exists a lot of issues for these algorithms. For example, in the deviations of Iterates, Forward Estimates and Greedy updates after the line 239, it seems that column numbers of matrix $Y$ and $Z$ are $N + 1$ instead of $N$. The dimensions are not consistent. Also to implement the orthogonal iterate in the equation under line 236, it needs the matrix $P_{t - 1}$ defined in equation (12). However, this definition in equation (12) needs the operation of matrix multiplication and matrix inversion, which could be very expensive in high dimension condition. Also, the authors argue that Orthogonal Forward Estimates can ensure that the condition number $\kappa_D = 1$ and the norm of the error vector is small. But this observation is not obvious and need detailed explanations. There is no formal or strict mathematical proof and theoretical analysis to ensure that these updates of matrix $D$ could satisfy the requirements presented in the section 3.1. At last for the batch technique, the authors apply the QR decomposition. However, the computational cost or complexity of QR decomposition could be very high in the high dimension condition and make the implementation very slow in practice.

**Questions:**

Please check the weakness section for questions.

**Limitations:**

Please check the weakness section for limitations.

---

> ### Author Rebuttal · Authors · 2023-08-08
>
> # Author's response
>
> We thank the reviewer for the reviews and detailed feedback.
>
> We begin by answering what we believe to be the most critical points of the review.
>
> ## A) Points 6 and 10: Clarity of algorithm
>
> > **10)** The most significant weakness [..] is the lack of formal algorithm for the update of matrices $Y,Z,D,G$. [...] there are a lot of unclear operations regarding the update of the corresponding matrices.
>
> > **6)** The authors presented some designed requirements in section 3.1. However, these requirements are too strong or restrictive as assumptions [..] the authors didn't give any strict mathematical proof to give any conditions to satisfy these requirements.
>
> ### Author’s response
>
> Based on the reviewer’s feedback, we added a new section in the appendix that presents each update in detail. Moreover, we corrected some inaccuracies in the section (e.g. the memory size of $N+1$ instead of $N$).
>
> Concerning **6)**, the requirements are not too strong as assumption: Section 5 presents explicit updates that satisfy those. We will explain in more detail why those updates satisfy the condition in Section 3.
>
> For instance, here is the algorithm for the orthogonal forward estimate.
>
> 1. if t>N: remove the first column of $Z_{t-1}, Y_{t-1}, G_{t-1}, D_{t-1}$
>
> 2. $g_t = \nabla f(x_t) $
>
> 3. $d_t = g_t-D(D^Tg_t)/h^2; d_t = -hd_t/||d_t||$ % compute $-(I-P)g_t$ and normalize knowing $D$ has orthogonal columns of norm $h$
>
> 4. $x_{t+1/2} = x_t+d_t$
>
> 5. $G_t = [G_{t-1} , \nabla f(x_{t+1/2})- \nabla f(x_t)]$
>
> 6. $Z_t = [Z_{t-1} , x_t]$
>
> 7. $D_t = [D_{t-1}, d_t]$
>
> 8. $\epsilon_t = eq (7)$
>
> 9. $x_{t+1}, M_{t+1} \leftarrow  [algorithm 1](f, Gt, Dt, \epsilon_t, x_t, (M_t/2))$
>
> **Complexity.** The matrix $D_t$ is iteratively constructed to be orthogonal, hence $D_t^TD_t =h^2 I$ is a scaled identity matrix, hence the multiplication / inversion is $O(N)$. Therefore, the whole update is $O(Nd)$, whose most complex part is computing $D_t(D_t^Tg_t)$.
>
> **Requirement.** The requirements are not restrictive. In our case,
>
> - *Req. 1b.* is satisfied by construction of the matrix D: the last column is the orthogonal projection of $\nabla f(x_t)$ onto the span of the other columns of $D_t$.
>
> - *Req. 2* is satisfied since $||D|| = \delta$ and $||\epsilon_t|| \leq h^2 + h||z_i-x_t||$ ($i\in[t-N,t]$), which can be bounded by $h^2+hR$ using Assumption 2, hence the ratio is bounded by R (this is a very crude bound).
>
> - *Req. 3* is satisfied because $D^TD = h^2I$, hence $\kappa = ||D^TD||\cdot ||(D^TD)^{-1}|| = 1$.
>
> ---
>
> ## B) Point 1: Comparison with existing work.
>
> > **1)** The authors ignore a lot of existing quasi-Newton type methods or accelerated versions that achieve $O(1/k)$ or $O(1/k^2)$.
>
> ### Author’s response
>
> We clarify how those works differ from our approach.
>
> **[1,2]** In [1,2], the primary focus was not on designing a quasi-Newton method, but rather on utilizing QN methods with prox operators. The authors state that they do not rely on or exploit the accuracy of second-order approximations [1]. They derive all the theory for arbitrary positive definite Hessian estimates, without assumptions about their structure or their proximity to the true Hessian [2]. They also occasionally use a scaled identity as estimate [1,2]. Additionally, the convergence rate is assumed under a non-verifiable assumption [1], and the authors acknowledge difficulties in ensuring the assumption without arming the convergence.
>
> **[3,4]** Papers [3] (appeared 1 month after submission) and [4] show improvement over first-order methods when $T > d log(d)$, but it requires tracking a full dxd matrix, resulting in $O(d^2)$ per iteration complexity. While improving over first-order methods, those algorithms remain suboptimal compared to Cubic Newton and are not suitable for large-scale problems.
>
> **Regime $N\approx d$.** The complete study of the $N=O(d)$ regime is left for future work; but we anticipate that our approach yields similar improvements when $N=d$, with $T=O(d)$.  Indeed, the projector P_i will span the entire space, resulting in smaller constants in $C_1/C_2/C_3/(M_0)_{max}$ in Thm 4/5/6/7. By combining rank update on H and approximated subproblem solutions, the per-iteration complexity should be $O(d^2)$ while maintaining a convergence rate close to that of second-order methods (i.e., $O(1/k^2) / O(1/k^3)$).
>
> [1] *Practical inexact proximal quasi-Newton method with global complexity analysis*.
>
> [2] *Proximal quasi Newton methods for regularized convex optimization with linear and accelerated sublinear convergence rates*.
>
> [3] *Accelerated Quasi-Newton Proximal Extragradient: Faster Rate for Smooth Convex Optimization*.
>
> [4] *Online Learning Guided Curvature Approximation: A Quasi-Newton Method with Global Non-Asymptotic Superlinear Convergence*.
>
> ---
>
> ## Detailed responses
>
> > **2.** The authors didn't compare the results to (accelerated) gradient descent.
>
> See global response
>
> > **3., 4., 5. and 9.** confusing notations
>
> 3. Even if $s_t,y_t$ are mon common in the QN literature, we prefer to keep $d_t$ for directions and $g_t$ for gradients.
>
> 4. The notation $D,G,\epsilon$ does not depends on $t$ in the first part of the paper, but we will make the transition more clear.
>
> 5. In the paper l144, we explain that $M_t$ is the current estimate in the backtracking line search. Thm 4 states $M_{\min} < M_t$ and is only required for convergence. $M_0>0$ is any initial estimate, and can be initialized to any positive value. In the experiments, $M_0=1$.
>
> 9. Equation l213 was a sketch showing the acceleration mechanism. $\lambda^{(1,2)}$ means $\lambda^{(1)}$ and $\lambda^{(2)}$. The $b_i$ are positive parameters that are built iteratively by the accelerated algorithm. To avoid confusion, we will remove this equation.
>
> > **7. and 8.** The $N$ should be comparable to $d$, but this causes $O(d^3)$ complexity. What are the advantages VS cubic Newton?
>
> See **B)**, response **[Reviewer 4Qfc, A)]**, and global response

---

> > ### Comment · Reviewer_SDpo · 2023-08-15
> >
> > Thanks for your clarification. I will keep my score.

---

> > > ### Author Response · Authors · 2023-08-15
> > >
> > > Dear reviewer,
> > >
> > > We believe we have successfully addressed all of the concerns you raised. Notably,
> > >
> > > 1. our paper compares favorably to the previously mentioned works,
> > > 2. the requirements listed in Section 3 are provably satisfied by our algorithm,
> > > 3. the constants in the rates of convergences are not worse than GD or AGD. We revised the paper accordingly based on your feedback.
> > >
> > > We understand that despite our efforts, you have maintained your initial evaluation of the paper. We kindly ask the reviewer to elaborate further on the specific aspects that still raise concerns after the rebuttal phase. Your insights would greatly assist us in addressing any remaining issues.

---

### Official Review · Reviewer_FnQv · 2023-07-25

**Soundness:** 3 good
**Presentation:** 2 fair
**Contribution:** 3 good
**Rating:** 6
**Confidence:** 4

**Summary:**

In the paper, the authors propose a new framework that connects Quasi-Newton methods with Anderson acceleration. By exploiting the Cubic Regularization technique, the authors achieve new competitive convergence rates comparable to first-order methods in the worst case and second-order methods in the best case. The paper describes various problem setups with their respective convergence rates: non-accelerated methods for non-convex, star-convex, and convex functions, as well as an accelerated method for convex functions. The authors also propose different variants of approximation. Experiments are presented for both convex and non-convex functions.

**Strengths:**

I believe the results presented in the paper are both original and significant. The novel connection between Anderson averaging, Quasi-Newton methods, and subspace sketch-Newton-type methods is both new and prospective. The provided proofs seem mostly correct.
Furthermore, introducing online approximation techniques adds value to the research. The thorough comparison of these techniques in practical applications showcases their effectiveness and underscores their relevance.
In conclusion, this paper contributes valuable insights to the field, and its innovative approach holds great potential for further advancements in optimization methods.

**Weaknesses:**

I may guess that the paper was created in a hurry before the conference deadline, which might explain why it contains a lot of small mistakes and misprints. While the mathematical results are sound, The mathematical results are sufficient but the overall presentation and clarity need improvement. A thorough review and revision by the authors are necessary to address these issues. Next, I will present some of the mistakes and misprints that I find.

1) Page 6, line 191, Theorem 5. The brackets on the right side are unnecessary. On the right side, $f(x_t)-f^{\ast}$ should be changed to $f(x_0)-f^{\ast}$. The same is in the Appendix.
2) Page 8, line 239. All $Z_t$ and $Y_t$ contain $N+1$ elements, but they should contain $N$. So, I think the indices should be corrected.
3) Page 36, line 785. $r_i$ was not defined anywhere before, moreover, $r$ is used in other places for different things. It probably should be $G_i$.
4) Page 36, line 787. The last formula in the line is incorrect; the left side is a vector, and the right side is a number.
5) Page 36, line 791. The inequality is incorrect because $(r_i - \nabla^2 f(x) d_i)^2$ is a vector and shouldn’t be squared in such a way.
6) Page 36, line 795. The third transition is incorrect because $\|w\|$ is missing for the $L$ term.
7) Page 37, line 800. I believe it is better to explicitly prove that the term $\alpha^T D^TG\alpha$ could be upper bounded by $\alpha^T (D^T G+G^T D)\alpha/2$; otherwise, it may be confusing.
8) Page 5, line 174. The notation $\varepsilon_t$ as a vector is confusing because a page before $\varepsilon_i$ was a number, the coordinate of a vector $\varepsilon$.
9) Page 4, line 127. $x_{+}$ is defined in Theorem 1 but never used inside. So, I think it can be removed. On the other hand, $x$ is used inside but not really specified.

Next are the moments that caused confusion for the first read.

10) Page 3, Motivation. The transitions between line 97 to formula (2) and then to formula (4) are very confusing. It may help to specify what are the $\alpha_i$ and $P_t$ for that case. For formula (4), the author may say that it is a special case of formula (2) when $H_0=0$. It may help to understand such a transition.
11) Page 4, line 142, formula 10. The norm of $\|D\|$ is not defined, which may cause some confusion because $D$ is a rectangular matrix and could have different norms.
12) Page 7, Algorithm 5. The notation $(M_0)_1$ is very confusing, especially when $M_0$ and $M_1$ also exist. Moving index $t$ of the step to the upper level, like $M^t_0$, may solve both this moment and $\varepsilon_t$ moment, but it is up to the authors how to resolve these issues with the notation.
13) Page 15, line 498, formula (19). As a small comment, I would suggest using the keyword “subspaced” or “sketched” instead of “stochastic” to avoid confusion with stochastic optimization methods such as SGD and others.


**Questions:**

1) It is quite challenging for me to understand how restrictive the Requirements for the methods are, especially Requirement 1b, as it appears to be quite limiting. Could you provide some comments on this and highlight specific examples where it is applicable and the reasoning woy?

2) In Sections 5 and 6, I am curious about the magnitude of $\kappa$ both in practice and in theory. For instance, when $\kappa = 10^9$ for the "Iterates only" approach, it seems to impose a very stringent condition for convergence because of the term $\kappa^2$ in $C_2$ and $C_3$. What are the typical values of $\kappa$ in practice for the "Iterates only" approach? Does such a large value of $\kappa$ indicate that the only applicable regime is the "Forward Estimates Only"?

3) Additionally, I am interested in the practical and theoretical values of $h$ in Sections 5 and 6. My main concern revolves around computational errors and the stability of the gradient difference $\nabla f(x_{t+1/2}) - \nabla f(x_{t})$. If $h=10^{-9}$, then $\nabla f(x_{t+1/2}) - \nabla f(x_{t})$ should be almost equal to zero close to the solution, even in double precision. This fact may destroy practical convergence of the method.

**Limitations:**

Yes

---

> ### Author Rebuttal · Authors · 2023-08-05
>
> # Author's response
> We thank the reviewer for the insightful comments. All the reported typos in points 1 to 9 are corrected, and we are reviewing the paper in more detail to correct the remaining ones.
>
> We also thank the reviewer for finding the results presented in the paper both original and significant, and that the paper contributes valuable insights to the field, and its innovative approach holds great potential for further advancements in optimization methods.
>
> We start by answering the question from the reviewer.
>
> ---
> ## Reviewer's question 1
> > It is quite challenging for me to understand how restrictive the Requirements for the methods are, especially Requirement 1b, as it appears to be quite limiting. Could you provide some comments on this and highlight specific examples where it is applicable and the reasoning woy?
>
> See also **SDpo A)**. The requirements are not restrictive. Section 5 presents several strategies that satisfy those; in particular, the simple forward estimate ensures that requirement 1.b is always satisfied, or using random batches of directions ensures that requirement 1.a is always satisfied.
>
> ---
>
> ## Reviewer's question 2
> > In Sections 5 and 6, I am curious about the magnitude of $\kappa$ both in practice and in theory. For instance, when $\kappa=10^9$ for the "Iterates only" approach, impose a very stringent condition for convergence because of the term $\kappa^2$ in $C_2$ and $C_3$. What are the typical values of $\kappa$ in practice for the "Iterates only" approach? Does such a large value of $\kappa$ indicate that the only applicable regime is the "Forward Estimates Only"?
>
> As mentioned by the reviewer, $\kappa$ can be huge in certain scenarios, for instance in the "iterate only" strategy. This phenomenon is quite well-known when trying to estimate the Hessian with finite difference without controlling the directions stored in memory. **One of the contributions of this paper is proposing the orthogonal forward estimate strategy that explicitly controls this term.** This is probably why the forward estimates only performs better in practice.
>
> The "iterate" and "greedy" strategies were introduced due to their simplicity and natural characteristics, and because the "iterate only" strategy is commonly used in standard QN techniques. It seems, however, that despite having a large condition number, the "iterate only" and "greedy" strategies do not perform as poorly as predicted by the theory.
>
> ---
>
> ## Reviewer's question 3
>
> > Additionally, I am interested in the practical and theoretical values of $h$ in Sections 5 and 6. My main concern revolves around computational errors and the stability of the gradient difference $\nabla f(x_{t+1/2})-\nabla f(x_t)$. If $h=10^{-9}$, then $\nabla f(x_{t+1/2})-\nabla f(x_t)$ should be almost equal to zero close to the solution, even in double precision. This fact may destroy practical convergence of the method.
>
> While it is theoretically possible to compute an optimal bound for the value of h by balancing the term delta in one of the theorems, this may not be feasible in practice as it might require too much information. This is why we used an arbitrarily small value, like 1e-9. As long as h is not way too big (i.e., larger than a gradient step), this does not affect the convergence speed. Moreover, $h=10^{-9} does not seem to be affected by numerical errors.
>
> Note that all elements of $\nabla f(x_t)$ are close to $0$ when $x_t$ is close to the solution; hence, it can be well approximated by double precision as it goes as low as $10^{-300}$.
>
> ---
>
> ## Reviewer's point 10-13
>
> ### Question 10
> > Clarity of the motivation paragraph.
>
> More detail in Page 3, Motivation. Given the feedback from the other review, we will remove this part and move it to the appendices. This will give more space to explain in more detail the special structure of quasi-Newton methods.
>
> In short, the projectors $Pi$ are defined as
> $$P_i = I - \frac{g_i d_i^T}{g_i^T d_i},$$
> and the coefficients alpha_i are defined as
> $$ \alpha_i = \left(d_i^T \frac{d_i^T g_i + g_i^T H_{i-1} d_i}{(g_{i}^T d_{i})^2} - \frac{g_i^T H_{i-1}}{g_{i}^T d_{i}}\right) \left(\prod_{j=i+1}^t P_j\right) \nabla f(x_t).$$
> We will also include additional explanation about $H_0 = 0$, as the reviewer suggested.
>
> ### Question 11
> > Which matrix norm?
>
> The norm is the L2 operator norm, i.e., the largest singular value. We will clarify.
>
> ### Question 12
> > Notation (M_0)_t confusing.
>
> We changed the notation accordingly as suggested by the reviewer.
>
>
> ### Question 13
> > “Stochastic” could be a confusing term.
>
> We changed the name to "random subspace".
>
> ---
>
> We hope we have successfully addressed all the reviewer's comments.

---

> > ### Comment · Reviewer_FnQv · 2023-08-14
> >
> > Thank you for the answers and consideration of the review's feedback! I raised my score to 6.
> >
> > Also, a small question, which does not affect the score, is how to compute in Python the gradient difference with double precision as it goes as low as $10^{-300}$? Does one have to use some additional libraries or classical double in Numpy and Pytorch is enough? Do you know whether Pytorch's or Jax's autograd is capable to compute gradients with such precision automatically? Thank you!

---

> > > ### Author Response · Authors · 2023-08-14
> > >
> > > Dear reviewer,
> > >
> > > We are glad to hear that we have answered your concerns! We thank the reviewer again for the detailed feedback that improved our paper.
> > >
> > > ---
> > >
> > > ### Double-precision floating points
> > >
> > > > How to compute in Python the gradient difference with double precision as it goes as low as $10^{-300}$?
> > >
> > > We used Matlab for the experiment with classical double precision. We quickly clarify the $10^{-300}$.
> > >
> > > Double-precision floating numbers are divided into mantissa and exponent; the exponent range is $[10^{-308}, 10^{308}]$, while the mantissa's precision in double-precision floating-point numbers is about 15 to 17 decimal digits. This means we can perform accurate operations up to 15-17 significant digits of the largest number between numbers in the exponent range.
> > >
> > > Hence, having the gradient components too big or too small does not affect the precision, as the exponent handles the scale of each coordinate. However, depending on the gradient's sensitivity of a particular application, $h=10^{-9}$ may be too small if the gradient does not vary sufficiently quickly. Indeed, this may result in subtracting two close numbers; therefore, the mantissa cannot represent their difference with sufficient precision.
> > >
> > > In our application, however, this value of $h$ worked fine and did not cause any problems. We also implemented a backtracking strategy to find $h$ such that the forward estimate is a descent direction. This strategy did not improve the convergence speed while introducing additional computation time.
> > >
> > > ---
> > >
> > > Thank you again for reviewing our paper!

---

### Author Rebuttal · Authors · 2023-08-09

# Author rebuttal

We thank all the reviewers for their insight and precise comments, and we appreciate seeing that the reviewers see the paper as innovative and hold great potential for further advancements in optimization methods (FnQv), providing a solid theory (4Qfc).

Since some comments in the reviews may stem from a misconception regarding the capabilities of QN methods, in this rebuttal, we quickly recontextualize the current state of the art, recall our contributions, and compare them to first and second order methods.

## Quasi-Newton method: current state of the art

Despite existing since more than 50 years, it is only recently that the explicit rates of QN methods have been studied:

> Although many other works on quasi-Newton methods have appeared [...], all of them still contain only asymptotic results.
>
> [50] Rodomanov & Nesterov (2021). *Greedy quasi-Newton methods with explicit superlinear convergence.*

The only known theoretical benefits of QN (compared to gradient/Newton’s method) is their ability to converge superlinearly when their memory $N$ is set to $N=d$, reducing the per-iteration complexity from $O(d^3)$ (Newton’s method) to $O(d^2)$.

These rates have been recently discovered by [50] (and refined in follow-up papers), whose focus is the study of **local** explicit superlinear rates of convergence of QN methods. See our answer to Reviewer SDpo for more details.

However, this contrasts with the practical usage of QN methods - the most popular and efficient algorithm is the limited-memory version of BFGS, where the memory parameter is much smaller than the dimension.

This is why many works tried to derive convergence bounds for limited memory QN methods, but unfortunately, often **1)** made unverifiable assumptions or 2) do not rely on how close the Hessian is approximated, and therefore, fail to capture the efficiency of QN methods.

This means **the question of global convergence rate remained open**, hence our contribution.

## Contributions

In this paper, we attempt to solve the complicated problem of proposing a QN method with global convergence guarantees.

We provide a **simple, adaptive method that competes with l-BFGS, with known global rates of convergence that interpolate the one of first and second order methods**:

1. The method is globally convergent with explicit rates in multiple scenarios, even for non-convex functions (Sec 3.2)

2. The method provably interpolates first-order and second-order rates (Sec 3.2)

3. Is adaptive to the problem’s constants (Alg. 3 & 5)

4. Have a bound on the accuracy of the Hessian estimation (Thm 1)

5. For a fixed memory, the cost per iteration remains linear in the dimension (see Solving the subproblem, sec 2.2)

6. Have verifiable and reasonable assumptions (sec 3.1 and sec. 5 for strategies that satisfies the requirements)

7. Is competitive with BFGS (Section 6)

Satisfying those points is a challenging problem. We are currently unaware of any line of work that met those conditions simultaneously. Moreover, this is a huge improvement over classical QN methods whose rates are currently unknown.

## Comparison with (Accelerated) Gradient and Cubic Newton

Prior to addressing the complexity analysis, we would like to reiterate the potential achievements attainable with QN methods.

### Expectations regarding QN methods

Some concerns are about comparing the theoretical convergence rate of our QN method with the one of accelerated gradient descent, while other comments suggest that to surpass accelerated gradient descent, the memory $N$ should be of the order $O(d)$, leading to at least a per-iteration complexity of $O(d^2)$.

We indicate to the reviewers that **those concerns apply to all QN methods**, not only ours:

- **Case $N\ll d$**. When $N<d$, QN methods are first order methods. Hence, they are subject to the same lower bounds on convex function: $O(1/T^2)$. Under the same assumptions, it is impossible to theoretically compete with the accelerated gradient method (given we have a bound on the convergence rate).

- **Case $N=O(d)$**. In this case, QN methods enjoy a better convergence rate than first-order methods, but the complexity increases to at least $O(d^2)$. This comes from the matrix-vector multiplication $H\nabla f(x)$. However, the convergence of QN methods in such a regime is worse than the one of second order methods.

### Our paper

Our algorithm presents the same advantages of classical QN methods over first and second order methods.


- **Case $N\ll d$**. We do recover the rate and constant of first-order methods. For instance, using the orthogonal forward estimate strategy, the constant $C_2$ in Thm 5 corresponds to the one of gradient descent in the worst case: indeed, $\|\nabla^2 f(x_i) - P_i\nabla^2 f(x_i)P_i\|\leq LR$  ($LR$ is the bound on the smoothness constant in our setting). Since $\delta<O(R)$ (this is a crude bound) and $\kappa=1$, the leading term in the bound is $O([LR]\cdot R^2/T)$. However, the constant $C_2$ may decrease as $N$ increases, since the projector $P_i$ becomes closer to identity. The same holds for the accelerated method. Such a comparison was not possible before, as there were no bounds of convergence for QN methods.

- **Case $N=O(d)$**. We recall to the reviewers that *this regime was not the main focus of the paper* as the per-iteration complexity of $O(d^2)$ is not suitable for large-scale problems. Nevertheless, for large $N$, we can improve the per-iteration complexity to $O(Nd+N^2)$, by using low-rank updates on the matrix $H$ and by utilizing low-rank updates on the matrix $H$ and employing an iterative method to solve the cubic subproblem (see e.g. Gao et al., *Approximate Secular Equations for the Cubic Regularization Subproblem*, Neurips 2022). Hence, in the worst case where $N=d$, the computation time is $O(d^2)$, which is the complexity of full memory QN methods like BFGS, and we enjoy a complexity close to the second-order methods.

---

### Decision · Program_Chairs · 2023-09-21

**Decision:**

Reject

**Comment:**

Dear authors,

Thank you for submitting your paper on the adaptive Quasi-Newton + Anderson acceleration framework, which also derived an explicit global convergence rate. The paper received positive feedback. Reviewers appreciated the concept of utilizing Quasi-Newton directions in conjunction with Anderson acceleration. The algorithm you proposed offers adept adaptations to a problem's unknown constants and showcased commendable performance across various test problems, as demonstrated in your paper.

However, reviewers also pointed out several weaknesses that preclude the paper from being accepted in its current form. While you have addressed many of these concerns, at least partially, during the discussion period, some reviewers expressed the need to see a revised version of the paper before confidently recommending its acceptance. We strongly believe that by integrating this feedback into the next version, the paper stands a high chance of acceptance in the subsequent cycle.

Best regards,
AC